# In mitosis integrins reduce adhesion to extracellular matrix and strengthen adhesion to adjacent cells

Maximilian Huber[1], Javier Casares-Arias [1], Reinhard Fässler [2],
Daniel J. Müller [1] ✉ & Nico Strohmeyer [1] ✉

To enter mitosis, most adherent animal cells reduce adhesion, which is followed by cell rounding. How mitotic cells regulate adhesion to neighboring cells and extracellular matrix (ECM) proteins is poorly understood. Here we report that, similar to interphase, mitotic cells can employ integrins to initiate adhesion to the ECM in a kindlin- and talin-dependent manner. However, unlike interphase cells, we find that mitotic cells cannot engage newly bound integrins to actomyosin via talin or vinculin to reinforce adhesion. We show that the missing actin connection of newly bound integrins leads to transient ECM-binding and prevents cell spreading during mitosis. Furthermore, β1 integrins strengthen the adhesion of mitotic cells to adjacent cells, which is supported by vinculin, kindlin, and talin1. We conclude that this dual role of integrins in mitosis weakens the cell-ECM adhesion and strengthens the cell-cell adhesion to prevent delamination of the rounding and dividing cell.

Most adherent animal cells undergo drastic morphological changes upon entry into mitosis, beginning with the disassembly of cell adhesion to the extracellular matrix (ECM) and progressing with the formation of a spherical shape that provides the perfect cellular geometry to assemble the mitotic spindle and partition the chromosomes to both daughter cells[1,2]. Mitotic cell rounding is complex and includes cell body retraction after remodeling cell adhesion[3,4], water influx, and the assembly of contractile actomyosin beneath the plasma membrane to counteract the increasing osmotic pressure[5,6]. Although failure to reduce cell-ECM adhesion at mitotic entry results in multinucleated cells[7–9], residual adhesion during mitosis plays an essential role for accurate spindle positioning and subsequent cell division[3,10]. While solitary cells such as fibroblasts or chondrocytes finely tune their adhesion to the surrounding ECM, epithelial cells integrate their intercellular adhesion with reduced adhesion to basement membrane proteins to maintain tissue integrity and prevent cell delamination[11]. With the separation of the midbody and the completion of cell division, the two daughter cells reassemble their native cell–cell and cell–ECM adhesion, spread and resume the polarized shape[12].

Cell–ECM adhesion is mediated primarily by the integrin superfamily, which comprises 24 α/β heterodimeric type I transmembrane proteins. While integrins are present in all cells, the cell type defines which integrins are expressed[13]. Integrin binding to ECM ligands is controlled by reversible conformational changes that shift integrins between inactive and active states[14,15]. Upon ligand binding integrins cluster into adhesion sites and assemble at their cytoplasmic domains hundreds of signaling and adapter proteins, which are collectively called adhesome and connect integrins to the contractile actomyosin cytoskeleton[16,17]. Thereby, integrin activation, clustering, association with the actin cytoskeleton, and adhesome assembly are regulated by two essential adapter proteins, talin and kindlin[18]. The disassembly of adhesion sites at mitotic entry is not fully understood but has been reported to include the reduction of Rap1A-mediated integrin activation, the phosphorylation of certain adhesome proteins, and the transfer of RhoA from adhesion sites to the newly assembled cortical actomyosin[2].

Cell–cell adhesion sites, such as adherens junctions (AJ), are maintained during mitosis. They are mediated by members of the cadherin superfamily that consists of more than 100 members,

[1]Department of Biosystems Science and Engineering, Eidgenössische Technische Hochschule (ETH) Zurich, 4058 Basel, Switzerland. [2]Department of Molecular Medicine, Max Planck Institute of Biochemistry, 82152 Martinsried, Germany. ✉e-mail: daniel.mueller@bsse.ethz.ch; nico.strohmeyer@bsse.ethz.ch

including E-, N-, VE- and P-cadherins[19]. Cadherins predominantly form homophilic interactions[20] and, like integrins, recruit numerous adapter and signaling proteins to their cytoplasmic domain, collectively called cadhesome. The cadhesome anchors cadherins to actomyosin and transduces biochemical and biophysical signals from adjacent cells[21–23]. Cadhesomes share specific proteins with adhesomes, including kindlin and vinculin[21,24]. Cadherin-based adhesion sites contribute to the mitotic process by orienting the mitotic spindle and preventing delamination of epithelial cells[25].

Although the importance of adhesion regulation before, during, and after mitosis is well documented, mechanistic insights into how mitotic cells regulate the initiation of cell–ECM and cell–cell adhesion are limited. Furthermore, the interplay of integrin- and cadherin-mediated adhesion during mitotic entry and progression is not well understood.

Here, we quantify cell–ECM and cell–cell adhesion in interphase and mitotic cells using atomic force microscopy (AFM)-based single-cell force spectroscopy (SCFS) in genetically engineered cell lines to understand how adhesion initiation and strengthening to the ECM and neighboring cells is differentially regulated. Our data shows that in mitotic cells integrins are not linked to the cytoskeleton by talin and vinculin, leading to reduced cell–ECM adhesion strengthening, while β1 integrins and different adhesome proteins, including vinculin, kindlin and talin, support mitotic cadherin-mediated cell–cell adhesion.

## Results

### Mitotic cells poorly strengthen integrin-mediated cell–ECM adhesion

To study integrin-mediated adhesion initiation and strengthening of interphase and mitotic cells to the basement membrane-like Matrigel[26], we combined SCFS with fluorescence microscopy to quantify cell cycle-dependent adhesion forces of HeLa cells. We attached single rounded interphase or mitotic HeLa cells expressing MYH9-GFP and H2B-mCherry to concanavalin A (ConA)-coated cantilevers, approached them to Matrigel- or bovine serum albumin (BSA)-coated supports, and allowed them to initiate and strengthen adhesion for 5 to 360 s before detaching them from the substrate to quantify the adhesion forces (Supplementary Fig. 1a). Throughout the experiments, we monitored the round morphology and cell cycle state by observing the histone H2B distribution (Supplementary Fig. 1b)[6]. The adhesion force of interphase HeLa cells to Matrigel increased with contact time, demonstrating their ability to initiate and strengthen adhesion (Fig. 1a). We quantified the adhesion strengthening rate as the slope of a linear fit through adhesion forces for all contact times (Supplementary Fig. 2a). Negligible adhesion forces of interphase HeLa cells to BSA or in the presence of β1 integrin blocking antibodies (AIIB2) to Matrigel showed that the initiation and strengthening of adhesion occurred predominantly via β1 integrins (Supplementary Fig. 2b, c). HeLa cells that were naturally in mitosis or chemically arrested at prometaphase by S-trityl-L-cysteine (STC; from now on called mitotic[STC] HeLa cells) established similar adhesion forces to Matrigel as interphase HeLa cells at 5 and 20 s contact time (Fig. 1a and Supplementary Fig. 2d). However, they established drastically reduced adhesion forces to Matrigel at longer contact times compared to those of interphase HeLa cells. SCFS with STC-treated interphase HeLa cells excluded that STC affected the initiation and strengthening of adhesion to Matrigel (Supplementary Fig. 2e) and demonstrated that the reduced strengthening is specific to mitosis. Moreover, SCFS with Madin-Darby Canine Kidney (MDCK) or Michigan Cancer Foundation-7 (MCF7) cells adhering to Matrigel as well as fibroblasts adhering to the fibronectin fragment FNIII7-10 (ref. 27) verified the drastically reduced adhesion strengthening of mitotic[STC] cells (Supplementary Fig. 2f–h).

To test whether mitotic cells establish integrin-mediated adhesion to the ECM, we quantified the adhesion forces of mitotic[STC] HeLa cells to Matrigel in the presence of AIIB2 and of mitotic[STC] fibroblasts to

FNIII7-10 lacking the integrin-binding RGD sequence (FNIII7-10ΔRGD; Supplementary Fig. 2i, j)[27]. In both cases, preventing integrin-ligand binding reduced the adhesion force of mitotic[STC] cells to background levels.

Next, we tested whether mitotic cells strengthen and mature integrin adhesion sites with time. To this end, we attached a paxillin-GFP expressing interphase or mitotic[STC] HeLa cell to a ConA-coated cantilever, approached the cell to a Matrigel-coated support, and allowed the cell to adhere to Matrigel for 60 min. During this contact time, we monitored the localization of paxillin-GFP by confocal time-lapse microscopy every 2 min and observed that the interphase HeLa cells matured adhesion sites and steadily increased their spreading area (Fig. 1b–e). However, upon retracting the cantilever after the contact time of 60 min to measure the cell adhesion force, the interphase cells established strong adhesion to Matrigel, which exceeded their adhesion to the cantilever. Consequently, the cells detached from the cantilever during retraction of the cantilever, which made it impossible to quantify their adhesion force to Matrigel. In contrast, cantilever-bound mitotic[STC] HeLa cells adhering to Matrigel for 60 min remained round, failed to assemble paxillin-GFP clusters, and established similar low adhesion forces as observed at 360 s contact time.

Flow cytometry revealed that HeLa cells express the integrin subunits α1, α2, α3, α5, α6, αV, and β1 on their surface (Supplementary Fig. 3). To investigate the cell cycle-dependent adhesion strengthening by different integrins, we quantified the adhesion forces of interphase and mitotic[STC] HeLa cells to collagen I, collagen IV, fibronectin, vitronectin, and laminin at 120 s contact time (Fig. 1f). Although interphase HeLa cells established considerable adhesion forces to each purified ECM protein, mitotic[STC] HeLa cells showed reduced adhesion forces to laminin, vitronectin, and fibronectin, and negligible adhesion forces to collagen I and collagen IV.

### Mitotic cells increase adhesion strengthening to adjacent cells

To characterize cell–cell adhesion initiation and strengthening in mitosis, we brought cantilever-bound interphase or mitotic[STC] HeLa cells, which mainly express N- and VE-cadherins (Supplementary Fig. 3), in contact with interphase or mitotic[STC] HeLa cells seeded for ≥30 min on Matrigel and quantified their adhesion forces. At contact times ranging from 5 to 360 s, two interphase HeLa cells initiated and steadily strengthened their cell–cell adhesion (Fig. 1g and Supplementary Fig. 4a). Mitotic[STC] HeLa cells established similar adhesion forces to interphase HeLa cells at contact times <120 s and higher adhesion forces at longer contact times (≥ 120 s) compared to adhesion forces measured between two interphase HeLa cells (Fig. 1g). The adhesion forces established between two mitotic[STC] HeLa cells were higher at contact times ≤120 s and similar at contact times >120 s as those between two interphase HeLa cells (Fig. 1g and Supplementary Fig. 4a). Incubation of interphase HeLa cells with STC did not affect adhesion forces to another interphase HeLa cell nor between mitotic and interphase HeLa cells (Supplementary Fig. 4b, c). Furthermore, similar to HeLa cells, mitotic[STC] MDCK or MCF7 cells established higher adhesion forces, which they also strengthened faster to other interphase or mitotic cells (Supplementary Fig. 4d, e). The results demonstrate that interphase and mitotic cells initiate cell–cell adhesion to interphase cells similarly while mitotic cells strengthen their adhesion faster.

### Integrin-ECM binding diminishes and cell–cell binding increases in mitosis

To test whether the impaired adhesion strengthening of mitotic HeLa cells to the ECM is caused by changes in integrin surface expression, we compared αV, β1, α6 and β4 integrin surface levels of interphase and mitotic[STC] HeLa cells by flow cytometry (Fig. 2a). Mitotic[STC] HeLa cells exhibited higher surface levels for all integrin

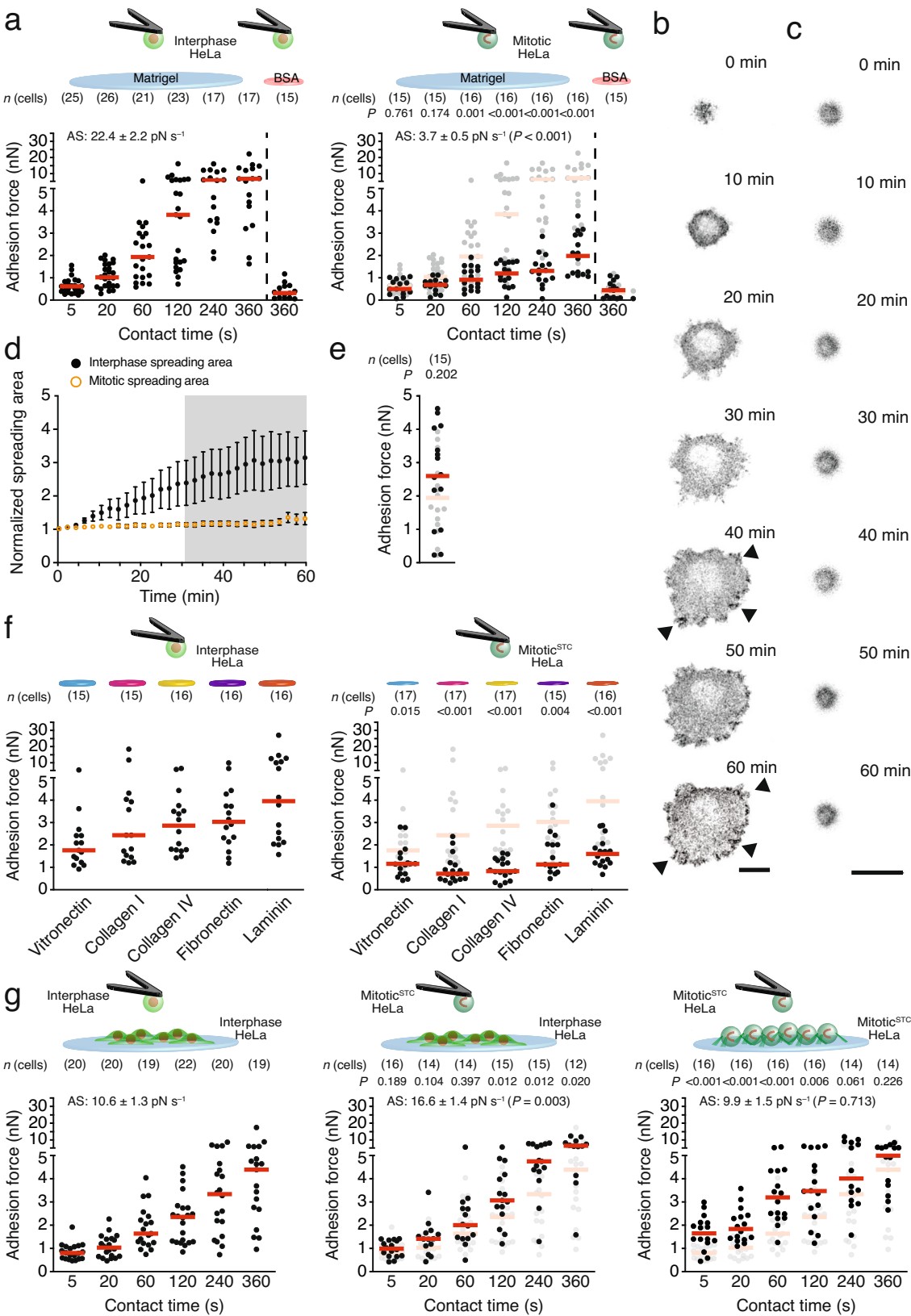

subunits tested than interphase HeLa cells. However, flow cytometry with antibodies that identify extended β1 integrin conformations (9EG7)[28] showed lower fluorescence signals on mitotic[STC] than on interphase HeLa cells (Fig. 2b). Consistent with this finding, SCFS at single molecule sensitivity revealed a reduced binding probability of mitotic[STC] HeLa cells to Matrigel (40.5 ± 3.2%,

mean ± SEM; $n = 10$) compared to interphase HeLa cells (52.9 ± 4.7%; $n = 10$; Fig. 2c and Supplementary Fig. 1c).

Next, we characterized cadherin surface expression and found that mitotic[STC] HeLa cells also displayed elevated surface levels of N-cadherins compared to interphase HeLa cells (Fig. 2d). Interestingly, the binding probability of mitotic[STC] HeLa cells to interphase HeLa was

**Fig. 1 | Mitotic cells considerably reduce adhesion strengthening to the ECM and increase adhesion to neighboring cells. a** Adhesion forces of interphase (left) or mitotic (right) HeLa cells to Matrigel or BSA after at given contact times. Dots represent adhesion forces of single cells, red bars median values and $n$(cells) the number of independent cells tested in at least three independent experiments. AS-values give the adhesion strengthening rate as the slope (±SE) of a linear fit through adhesion forces for all contact times with the $P$-value comparing the AS-value to the reference data set (Supplementary Fig. 2a). Adhesion forces of interphase HeLa cells to Matrigel are given as reference in gray for comparison with mitotic cells. **b, c** Representative time-series of confocal microscopy images of a paxillin-GFP-expressing interphase (**b**) or a mitotic[STC] (**c**) HeLa cells ($n = 7$) adhering to Matrigel during SCFS. Arrows show paxillin-GFP clusters. Scale bars, 20 μm. **d** Contact time-dependent and normalized spreading area (±SEM) of paxillin-GFP-expressing interphase and mitotic[STC] HeLa cells ($n = 7$ independent experiments). Gray area indicates significant differences in spreading area between interphase and

mitotic[STC] HeLa cells ($P$ values Supplementary Table 1). **e** Adhesion forces of mitotic[STC] HeLa cells to Matrigel after 60 min. Adhesion forces after 360 s to Matrigel are given as gray reference. Data representation as described for **a**. **f** Adhesion forces of interphase (left) or mitotic[STC] (right) HeLa cells to given purified ECM proteins at 120 s contact time. Data representation as in **a**. Adhesion forces of interphase HeLa cells to respective ECM proteins are given as gray references. **g** Adhesion forces between two interphase (left), an interphase and a mitotic[STC] (middle), or two mitotic[STC] (right) HeLa cells at given contact times. $P$ values comparing AS-values of displayed and reference data sets (Supplementary Fig. 4a). Adhesion forces between two interphase HeLa cells are given as reference in gray. Data representation as in **a**. "Mitotic[STC]" indicates that mitotic cells were enriched by STC ("Methods"). $P$ values comparing given data with reference data (**a, d–g**) were calculated using two-tailed Mann–Whitney tests and $P$ values comparing AS-values were calculated by a two-tailed extra sum of squares $F$-test.

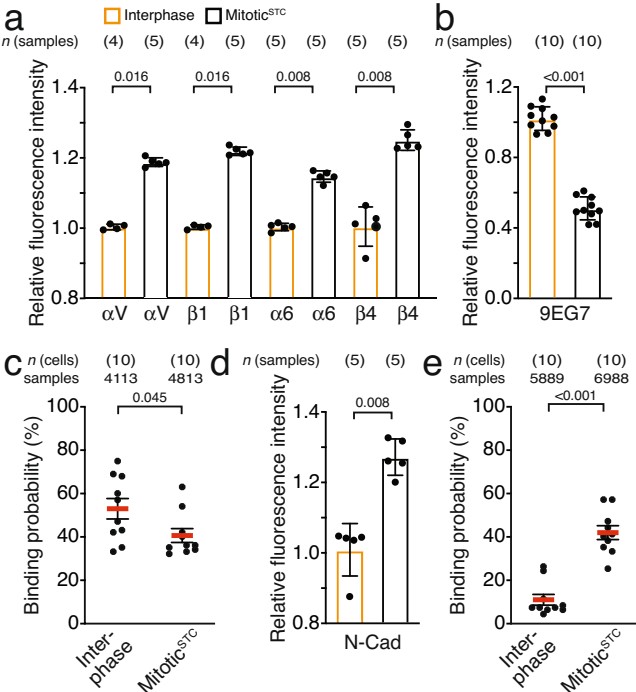

**Fig. 2 | Ligand binding of integrin and cadherin is regulated throughout the cell cycle. a** Interphase and mitotic[STC] HeLa cells were fluorescently labeled for integrin subunits αV, β1, α6 and β4 and analyzed by flow cytometry. Dots represent the median fluorescent intensities of 20,000 cells analyzed per sample normalized to the mean of median fluorescent intensity of interphase HeLa cell samples, bars the mean of all medians and error bars show the SEM. $n$(samples) indicate the number of biological independent samples tested. **b** Flow cytometry of interphase and mitotic[STC] HeLa cells labeled for β1 integrins in an extended conformation (clone 9EG7). Data representation as described for **a**. **c** Binding probability of interphase and mitotic[STC] HeLa cells to Matrigel. Dots represent the binding probability of individual HeLa cells, red bars indicate the median binding probability of all tested cells, and the error bars the SEM. $n$(cells) indicates the number of HeLa cells probed and samples the number of force-distance recorded for each condition. **d** Interphase and mitotic[STC] HeLa cells were labeled for N-cadherins and analyzed by flow cytometry. Data representation as described in **a**. **e** Binding probability of interphase or mitotic[STC] HeLa cells to single interphase cells spread on the substrate. Data representation as described in **c**. $P$ values compare indicated conditions and were calculated using two-tailed Mann–Whitney tests.

## Role of integrin-adapters for cell–ECM and cell–cell adhesion in mitosis

To investigate whether integrin-binding adapter proteins modulate the adhesion strengthening of mitotic HeLa cells to Matrigel, we disrupted the genes encoding vinculin (VKO) or talin1/2 (TKO) or kindlin1/2 (KKO) in HeLa cells by CRISPR/Cas9 (Supplementary Fig. 5). The unchanged surface levels of laminin- and collagen-binding integrins after genetic manipulation (Supplementary Fig. 3) allowed us to compare the adhesion forces of VKO, TKO, KKO and wildtype HeLa cells to Matrigel. Although the depletion of vinculin in interphase HeLa cells slightly increased the adhesion force to Matrigel at 5 s contact time and reduced the adhesion force at ≥120 s contact time, it had no effect on the adhesion force of mitotic[STC] HeLa cells to Matrigel for all contact times tested (Fig. 3a and Supplementary Fig. 6a, b). TKO and KKO HeLa cells showed drastically reduced adhesion forces and strengthening to Matrigel irrespective of whether they were in interphase or mitosis (Fig. 3b, c and Supplementary Fig. 6c–h). Re-expression of the deficient adhesome proteins rescued the adhesion defects (Supplementary Fig. 7) demonstrating that the adhesion defects of our engineered cell lines are specific. Further, we confirmed that kindlin and talin are essential for the interphase and mitotic adhesion of fibroblasts to FNIII7-10 (Supplementary Fig. 8a, b).

Next, we asked whether talin, kindlin, or vinculin influence cell–cell adhesion during interphase and/or mitosis. Depletion of these proteins in HeLa cells did not affect the surface expression of cadherin (Supplementary Fig. 3). While vinculin-depleted interphase HeLa cells did not change their adhesion force to interphase wildtype HeLa cells, vinculin depletion reduced the adhesion forces of mitotic[STC] HeLa cells to interphase wildtype HeLa cells for contact times ≥ 240 s (Fig. 4a and Supplementary Fig. 9a, b). Furthermore, loss of vinculin in mitotic[STC] HeLa cells lowered their adhesion forces to mitotic[STC] wildtype HeLa cells for all contact times measured.

The depletion of talin in interphase HeLa cells did not affect the adhesion to wildtype interphase HeLa cells, but reduced the adhesion forces between mitotic[STC] TKO HeLa cells and interphase or mitotic[STC] wildtype HeLa cells (Fig. 4b and Supplementary Fig. 9c, d). In all three combinations (interphase-interphase; mitotic[STC]-interphase; mitotic[STC]-mitotic[STC]) the adhesion forces and strengthening between KKO and wildtype HeLa cells were lower compared to two wildtype HeLa cells at all contact times tested (Fig. 4c and Supplementary Fig. 9e, f). Since re-expression of the deficient adhesome proteins rescued the adhesion defects (Supplementary Fig. 7d–f), we conclude that the defects are specific.

The findings demonstrate an essential role of kindlin and talin for integrin-ECM adhesion in interphase and mitosis and a role of vinculin in strengthening adhesion to the ECM for interphase but not for mitotic HeLa cells. Furthermore, we reveal an important role of kindlin

four times higher compared to the binding probabilities between interphase cells (Fig. 2e), indicating that mitotic cells slightly decrease adhesion initiation to the ECM, but strongly increase adhesion initiation to other cells.

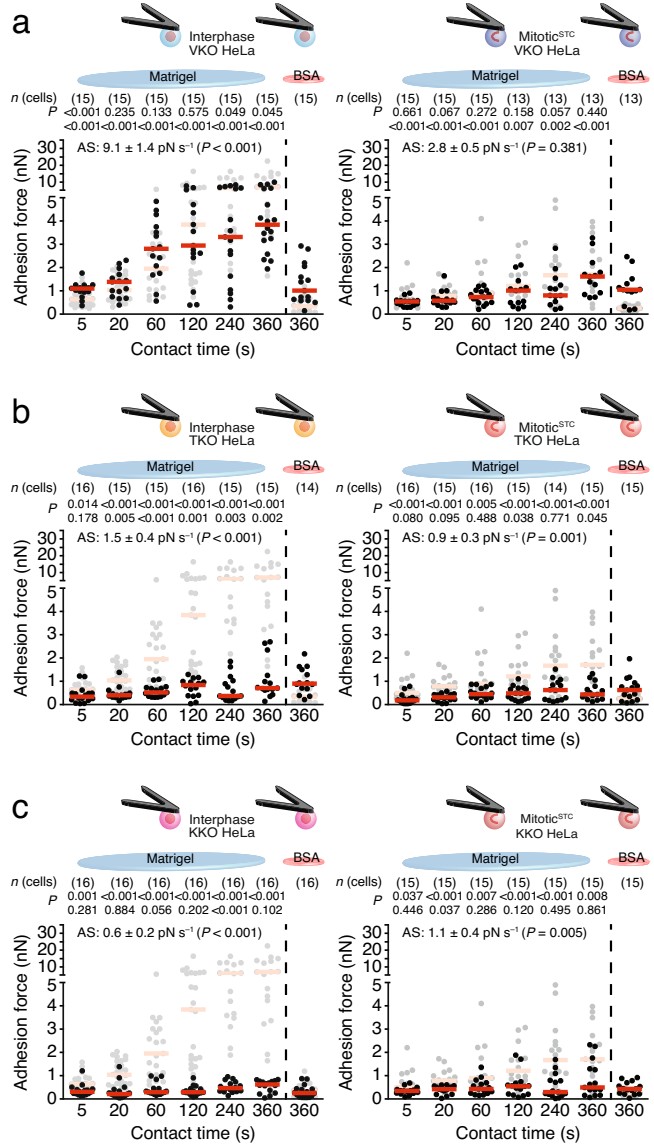

**Fig. 3 | Adhesome proteins differentially regulate cell–ECM in interphase and mitosis.** Adhesion forces to Matrigel or BSA of interphase (left) and mitotic[STC] (right) HeLa cells depleted from vinculin (VKO) (**a**), talin1/2 (TKO) (**b**), or kindlin1/2 (KKO) (**c**) to Matrigel or BSA after given contact times. As reference, cell–ECM adhesion forces of wildtype HeLa cells are given in gray (Fig. 1a). Dots represent adhesion forces between single cantilever-bound and substrate-spread cells, red bars their median values and n(cells) the number of independent cells in at least three independent experiments. AS-values give the adhesion strengthening rate as the slope (±SE) of a linear fit through adhesion forces for all contact times with the P value comparing the AS-value to the reference data (Supplementary Fig. 6). Top row P values compare the displayed data with the reference data. Bottom row P values compare displayed adhesion forces with adhesion forces of wildtype HeLa cells to BSA (left, Supplementary Fig. 2b) or with the adhesion forces of interphase cells of the respective HeLa cell line (right, data taken from left panel). P values comparing adhesion forces were calculated using two-tailed Mann–Whitney tests and P values comparing AS-values were calculated by two-tailed extra sum of squares F-tests.

for cell–cell adhesion in interphase and kindlin, talin, and vinculin in mitosis.

## Rap1A increase mitotic cell–ECM adhesion

Next, we investigated cell–ECM adhesion of mitotic HeLa cells treated with $Mn^{2+}$ or expressing constitutively active Rap1A, which activate integrins and perturbs mitotic cell rounding[8,9,29]. While $Mn^{2+}$-treatment

increased the levels of the β1 integrin-activation-associated 9EG7 epitope on suspended interphase and mitotic[STC] HeLa cells, it enhanced the adhesion forces and strengthening of interphase but not of mitotic[STC] HeLa cells to Matrigel (Fig. 5a, b and Supplementary Fig. 6i, j).

Retroviral transduction of HeLa cells with a constitutively-active mutant of Rap1A (Rap1A-V12; Rap1A-CA HeLa cells) increased the β1 integrin-activation-associated 9EG7 epitope levels on interphase and mitotic[STC] Rap1A-CA HeLa cells to a slightly lower extent than $Mn^{2+}$ (Fig. 5c). In contrast to $Mn^{2+}$-treatment, Rap1A-CA did not increase the adhesion forces of interphase HeLa cells to Matrigel but increased the adhesion force and strengthening of mitotic[STC] HeLa cells to Matrigel (Fig. 5d and Supplementary Fig. 6k–m). However, the adhesion forces of mitotic[STC] Rap1A-CA HeLa cells did not reach those of interphase wildtype HeLa cells to Matrigel.

## Mitotic cells do not connect newly ligated integrins to actomyosin cortex

Activated Rap1A is supposed to release talin auto-inhibition and promote talin binding to integrins and actomyosin[30]. To test how overexpression of the talin1-head domain (THD) affects mitotic cell–ECM adhesion, we retrovirally transduced TKO HeLa cells with THD (TKO + THD). While β1 integrin-activation-associated 9EG7 epitope levels were lower on interphase TKO + THD HeLa cells than on interphase wildtype HeLa cells, the levels were similar between mitotic[STC] TKO + THD and wildtype HeLa cells (Fig. 6a), indicating that in contrast to $Mn^{2+}$ and Rap1-CA, the THD is not sufficient to increase β1 integrin activity in mitotic cells. Interestingly, the adhesion forces of interphase TKO + THD HeLa cells were similar to interphase wildtype HeLa cells at 5 and 20 s contact time, whereas adhesion strengthening at contact times ≥ 60 s was reduced. Adhesion forces of mitotic[STC] TKO + THD HeLa cells to Matrigel were similar to those of mitotic[STC] wildtype HeLa cells at all contact times tested (Fig. 6b and Supplementary Fig. 6n–p). To verify these results, we quantified the adhesion forces of TKO + THD fibroblasts to FNIII7-10 (Supplementary Fig. 8c). Interphase TKO + THD fibroblasts initiated adhesion similar to wildtype fibroblasts but drastically reduced their adhesion force at contact times >120 s. Importantly, the adhesion forces of mitotic[STC] TKO + THD and wildtype fibroblasts were indistinguishable.

Next, we tested whether destabilizing the mitotic actomyosin cortex by inhibiting actin polymerization or RhoA activity allowed mitotic cells to strengthen adhesion (Fig. 6c, d). To this end, we incubated mitotic[STC] HeLa cells with 0.1, 0.5, or 1 mM cytochalasin D and quantified their adhesion force to Matrigel. Although the lowest concentration did not affect their adhesion force, incubating mitotic[STC] HeLa cells with 0.5 mM and 1 mM cytochalasin D decreased adhesion forces at contact times ≥120 s. We also quantified the adhesion force of interphase and mitotic[STC] HeLa cells incubated with Rho inhibitor I for 4 h before and during SCFS to Matrigel. RhoA inhibition decreased the adhesion forces of interphase HeLa cells at contact times ≥120 s, consequently their adhesion strengthening but did not affect mitotic[STC] adhesion forces.

Together, these results indicate that newly ligand-bound integrins in mitotic cells do not engage with actomyosin and that destabilizing the actomyosin cortex does not increase mitotic cell–ECM adhesion.

## Talin1-head rescues mitotic cell–cell adhesion defects caused by talin deficiency

Since talin promotes mitotic cell–cell adhesion, we investigated whether Rap1-CA expression influences cell–cell adhesion and whether THD expression rescues the mitotic cell–cell adhesion defects of TKO HeLa cells (Fig. 7a, b and Supplementary Fig. 9g–j). The cell–cell adhesion forces of HeLa cells in all three setups (interphase-interphase; mitotic[STC]-interphase; mitotic[STC]-mitotic[STC]) were neither affected upon Rap1-CA expression nor between TKO + THD and wildtype

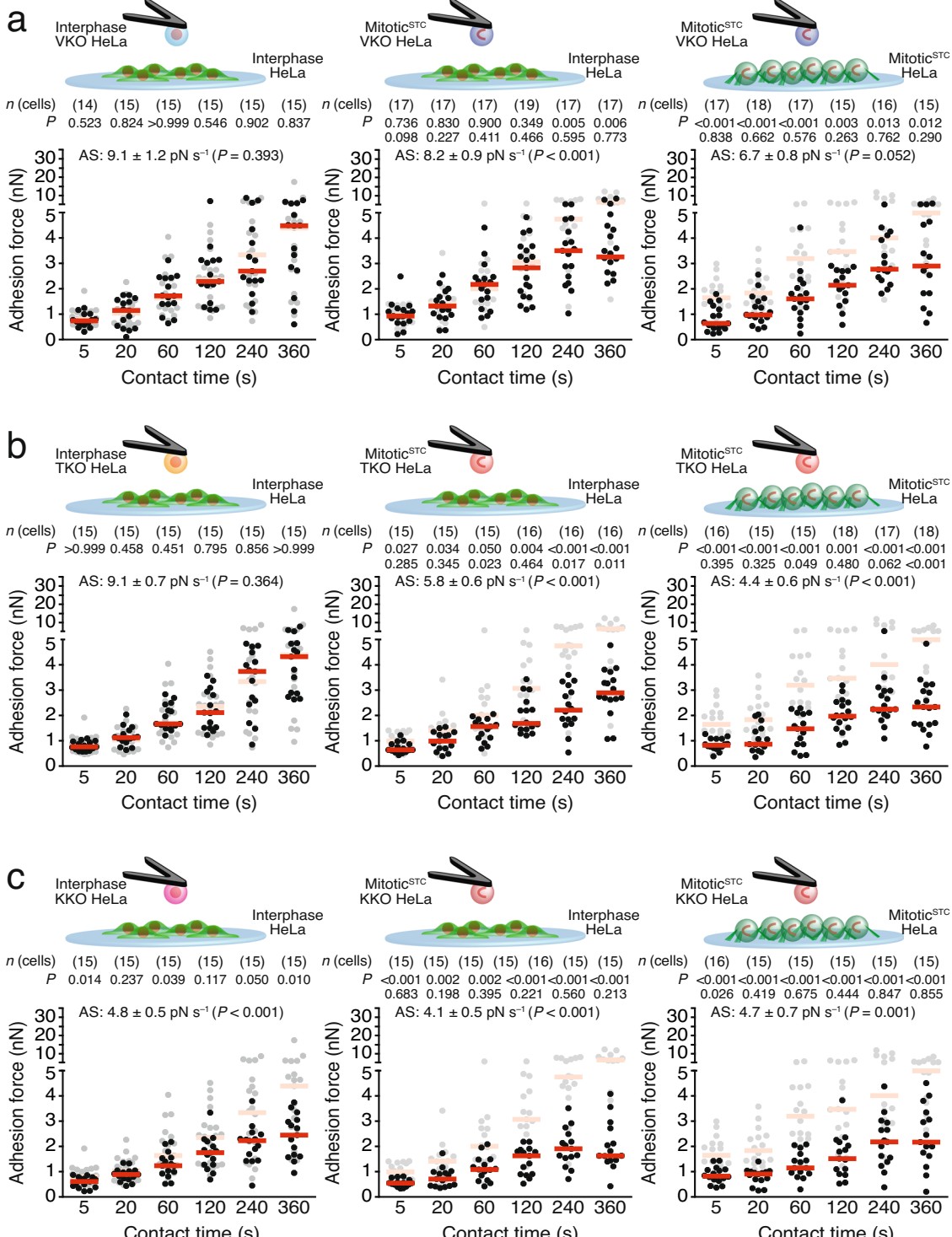

**Fig. 4 | Integrin adhesome proteins differently regulate cell–cell adhesion in interphase and mitosis.** Cell–cell adhesion forces between VKO (**a**), TKO (**b**), or KKO (**c**) HeLa cells and wildtype HeLa cells spread on a Matrigel-coated substrate after given contact times. Panels show adhesion forces of interphase KO and interphase wildtype (left), mitotic[STC] KO and interphase wildtype (middle) or mitotic[STC] KO and mitotic[STC] wildtype HeLa cells (right). AS-values give the adhesion strengthening rate as the slope (±SE) of a linear fit through adhesion forces for all contact times with the *P* value comparing the *AS*-value to the reference data (Supplementary Fig. 9). As reference cell–cell adhesion forces established between two wildtype HeLa cells in the respective condition is given in gray (Fig. 1g). Top row *P* values compare displayed data with reference data. Bottom row *P* values compare displayed data with adhesion forces established between interphase KO HeLa cells and interphase wildtype HeLa cells (data from left panel). Dots represent adhesion forces between single cantilever-bound and substrate-spread cells, red bars their median values and *n*(cells) the number of independent cells in at least three independent experiments. *P* values comparing adhesion forces were calculated using two-tailed Mann–Whitney tests and *P* values comparing AS-values were calculated by two-tailed extra sum of squares *F*-tests.

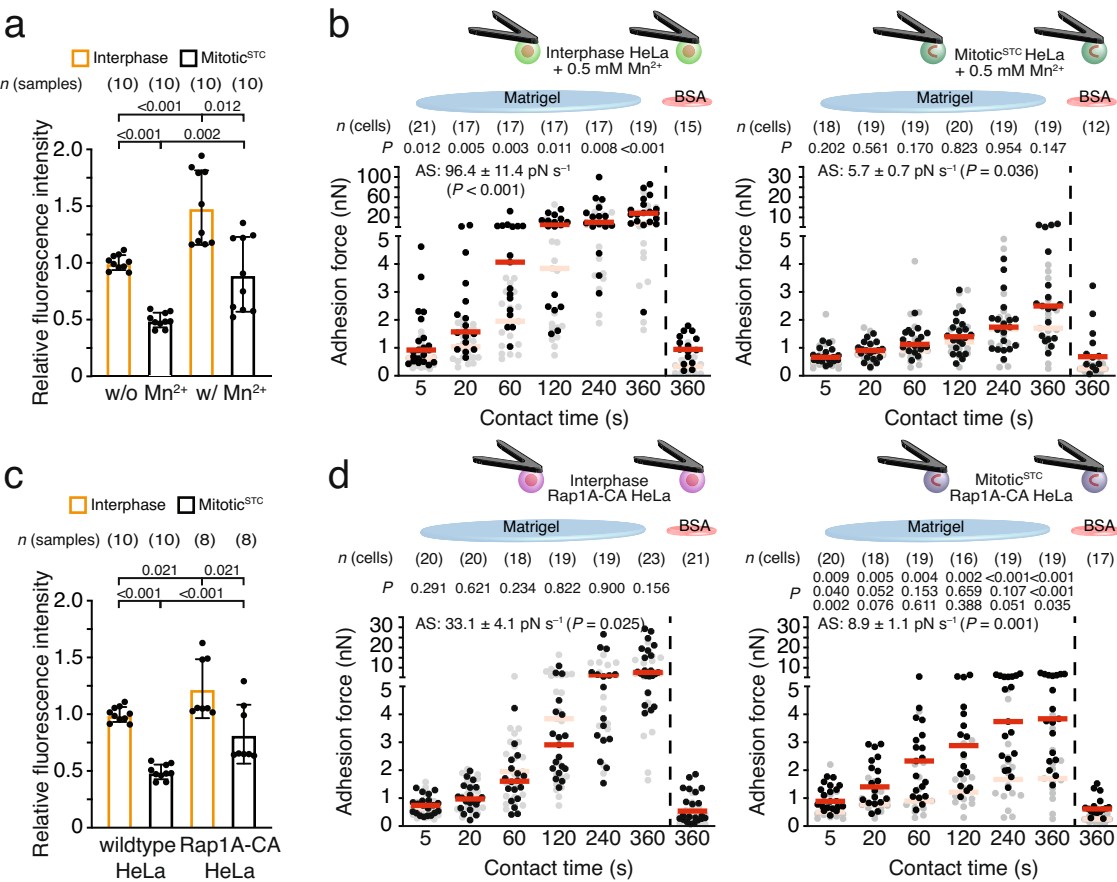

**Fig. 5 | Expression of constitutively-active Rap1A increases mitotic cell–ECM adhesion strengthening. a** Fluorescence intensity of untreated (Fig. 2b) or $Mn^{2+}$-treated HeLa cells labeled for extended β1 integrins (clone 9EG7). Dots represent median fluorescence intensities of 20,000 cells normalized to the mean median fluorescence intensity of untreated interphase HeLa cells, bars the mean of all normalized medians, and error bars their SEM. *n*(samples) indicates the number of biological independent samples tested. **b** Adhesion forces of interphase (left) or mitotic[STC] (right) HeLa cells incubated with 0.5 mM $Mn^{2+}$ to Matrigel or BSA after given contact times. Adhesion forces of untreated interphase or mitotic HeLa cells to Matrigel or BSA are given as reference in gray (Fig. 1a). Dots represent adhesion forces of single cells, red bars their median and *n*(cells) the number of independent cells in at least three independent experiments. AS-values give the adhesion strengthening rate as the slope (±SE) of a linear fit through adhesion forces for all

contact times with the *P* value comparing the AS-value to the reference data. **c** Flow cytometry of wildtype (Fig. 2b) or Rap1A-CA overexpressing HeLa cells labeled for extended β1 integrins (clone 9EG7). Data representation as described for **a**. **d** Cell–ECM adhesion forces of interphase (left) or mitotic[STC] (right) HeLa cells expressing constitutively active Rap1A (Rap1A-CA HeLa cells) to Matrigel or BSA at given contact times. Data representation as described for **b**. For AS-value comparison see Supplementary Fig. 6. Top row *P* values compare displayed data with given reference data. Second row *P* values compare displayed adhesion forces with the adhesion forces of interphase Rap1A-CA HeLa cells (right, data taken from left panel). Bottom row *P* values compare displayed adhesion forces with adhesion forces of interphase wildtype HeLa cells to Matrigel. *P* values compare indicated conditions and were calculated using two-tailed Mann–Whitney tests.

HeLa cells at all contact times analyzed, which together suggests that the THD suffices to mediate cell–cell adhesion of mitotic[STC] HeLa cells.

**Mitotic cells employ integrins to strengthen cell–cell adhesion**

Our finding that mitotic cell–cell adhesion requires kindlin and talin points towards the participation of integrins in mitotic cell–cell adhesion initiation and/or strengthening. To address the possible roles of integrins, we first deprived interphase and mitotic[STC] HeLa cells from $Ca^{2+}$ by EGTA chelation, which curbs homophilic cadherin interactions[31], but maintains integrin function[32], and measured the adhesion forces between two interphase, an interphase and a mitotic[STC], or two mitotic[STC] HeLa cells (Fig. 8a). $Ca^{2+}$-chelation drastically reduced the adhesion forces and strengthening between interphase HeLa cells to non-specific levels. Importantly, the deprivation of $Ca^{2+}$ also reduced the adhesion forces between a mitotic[STC] and an interphase or another mitotic[STC] HeLa cell. However, the adhesion forces were considerably higher than between EGTA-treated interphase HeLa cells. To address whether β1 integrins participate in mitotic cell–cell adhesion, we blocked their ligand binding on interphase or mitotic[STC]

HeLa, MDCK and MCF7 cells with AIIB2, which potently inhibited β1 integrin mediated adhesion of interphase HeLa cells to Matrigel (Supplementary Fig. S2c). Then we attached them to cantilevers and quantified their adhesion forces to untreated interphase or mitotic[STC] HeLa, MDCK, or MCF7 cells seeded on Matrigel (Fig. 8b and Supplementary Fig. 10). Although the adhesion forces between two interphase HeLa cells remained unaffected by β1 integrin blocking, it reduced the adhesion force of mitotic[STC] HeLa cells to untreated interphase or mitotic[STC] HeLa cells at contact times ≥ 120 s or for all contact times tested, respectively. Interestingly, our adhesion force data suggest that integrins and cadherins contribute approximately equally to the adhesion forces established between mitotic[STC] HeLa cells and interphase or mitotic[STC] HeLa cells at tested contact times (Supplementary Fig. 10c). SCFS with AIIB2-treated mitotic[STC] MDCK and MCF7 cells confirmed the involvement of β1 integrins in mitotic cell–cell adhesion initiation and strengthening (Supplementary Fig. 10d, e). Consistent with these findings, β1 integrin blocking did not affect the binding probability between two interphase HeLa cells, while AIIB2-treatment of mitotic[STC] HeLa cells reduced their binding

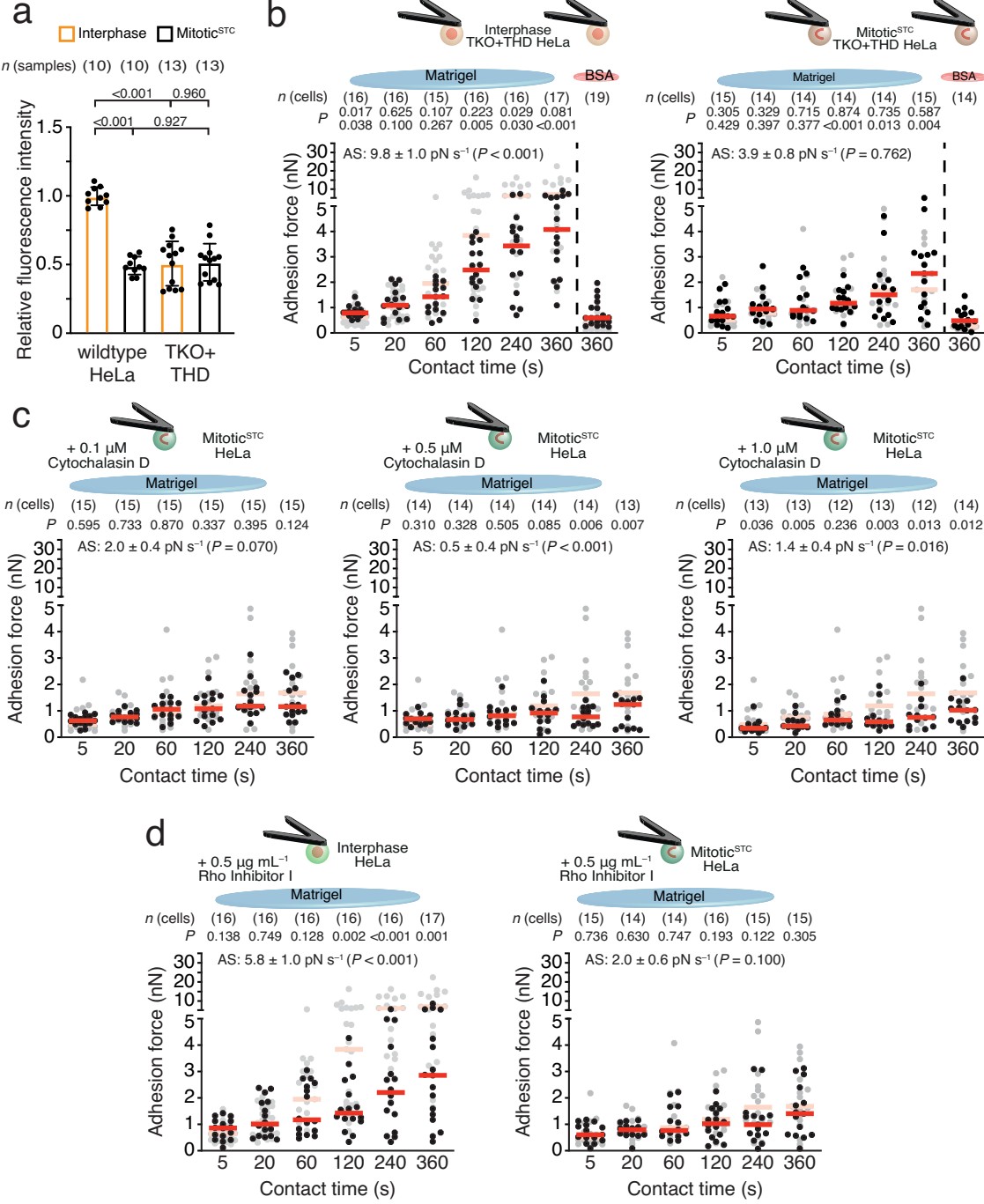

**Fig. 6 | Mitotic cells do not engage integrins to the contractile actomyosin.**
**a** Fluorescence intensity of wildtype (Fig. 2b) or talin1-head domain expressing (TKO + THD) HeLa cells labeled for extended β1 integrins (clone 9EG7). Dots represent median fluorescence intensities of 20,000 cells normalized to the mean median fluorescence intensity of untreated interphase HeLa cells, bars the mean of all normalized medians, and error bars their SEM. *n*(samples) indicates the number of biological independent samples tested. **b** Adhesion forces of interphase (left) or mitotic[STC] (right) TKO + THD HeLa cells to Matrigel or BSA after given contact times. Dots represent adhesion forces of single cells, red bars their median and *n*(cells) the number of independent cells in at least three independent experiments. AS-values give the adhesion strengthening rate as the slope (±SE) of a linear fit through adhesion forces for all contact times with the *P* value comparing the AS-value to the reference data. *P* values compare displayed data with reference data. For AS-value comparison see Supplementary Fig. 6. Top *P* values compare displayed data with given reference data, second row *P* values compare adhesion forces of given data with mitotic[STC] wildtype (left) and with TKO + THD interphase (right) HeLa cells on Matrigel. **c** Adhesion forces of mitotic[STC] HeLa cells in the presents of cytochalasin D in the given concentration. **d** Adhesion forces of interphase or mitotic[STC] HeLa cells in the presence of Rho inhibitor I. **c**, **d** as reference untreated HeLa cells are given. *P* values compare displayed with reference data. *P* values comparing adhesion forces and fluorescence intensities were calculated using two-tailed Mann–Whitney tests and *P* values comparing AS-values were calculated by two-tailed extra sum of squares *F*-tests.

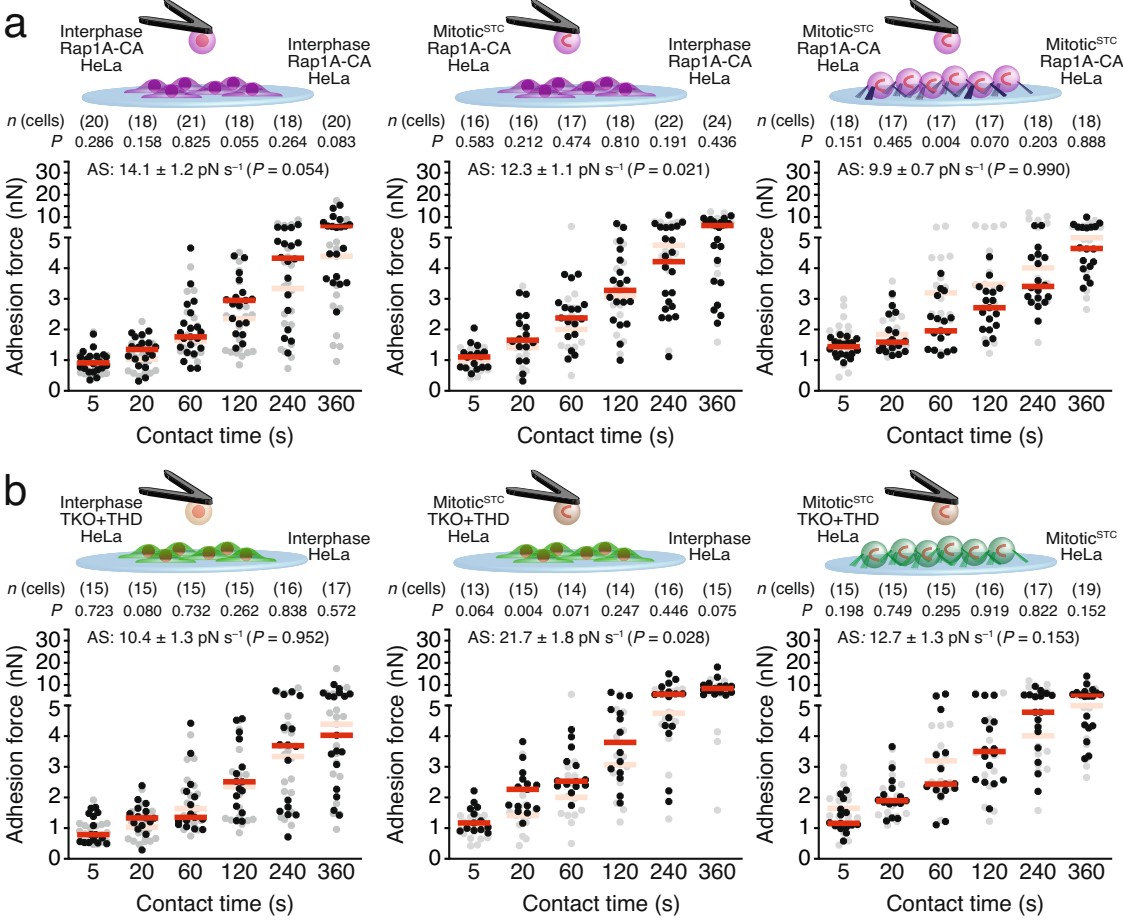

**Fig. 7 | Talin1-head domain expression in mitotic talin-deficient HeLa cells recovers cell–cell adhesion defects. a** Cell–cell adhesion forces established between two interphase (left), an interphase and a mitotic[STC] (middle), or two mitotic[STC] (right) Rap1A-CA. **b** Cell–cell adhesion forces between interphase TKO + THD and interphase wildtype (left), mitotic[STC] TKO + THD and interphase wildtype (middle) or mitotic[STC] TKO + THD and mitotic[STC] wildtype (right) HeLa cells. Dots represent adhesion forces between single cantilever-bound and substrate-spread cells, red bars their medians and n(cells) the number of independent cells in at least three independent experiments. As reference for cell–cell

adhesion forces established between two wildtype HeLa cells in the respective condition are given in gray (Fig. 1g). P values compare adhesion forces with reference data. AS-values give the adhesion strengthening rate as the slope (±SE) of a linear fit through adhesion forces for all contact times with the P value comparing the AS-value to the reference data and were calculated by two-tailed extra sum of squares F-tests (Supplementary Fig. 9). P values comparing adhesion forces and binding probabilities were calculated using two-tailed Mann–Whitney tests.

probability to untreated interphase HeLa cells. However, the binding probability of AIIB2-treated mitotic[STC] HeLa cells to interphase HeLa cells was still higher than that of AIIB2-treated or untreated interphase HeLa cells to untreated interphase HeLa cells (Fig. 8c).

We next evaluated whether integrins in mitotic cells could bind to ECM proteins deposited on the apical side of the cell on the substrate (Supplementary Fig. 11). Therefore, we seeded rounded interphase or mitotic[STC] HeLa cells on ConA for 30 min to mimic cantilever-bound HeLa cells. To mimic substrate bound cells, we seeded interphase HeLa cells on Matrigel for 2 h or for 12 h while arresting them in mitosis using STC. Subsequently, we chemically fixed the cells and stained them for actin, DNA, and collagen I, collagen IV, fibronectin, or laminin. Although we were able to detect some of the tested ECM proteins within the cell, we were unable to detect collagen I, collagen IV, fibronectin, or laminin at the periphery of the cells under any of the conditions. We also tested whether extrinsically activated β1 integrins increased cell–cell adhesion of interphase or mitotic cells (Supplementary Fig. 12a, b). Therefore, we quantified the adhesion forces of interphase or mitotic[STC] HeLa cells to an interphase or mitotic[STC] HeLa cell in the presence of $Mn^{2+}$ or a β1 integrin-activating antibody (12G10)[33]. However, extrinsic β1 integrin activation did not affect the

adhesion forces and strengthening between interphase or mitotic HeLa cells within 360 s.

To further test whether β1 integrins localize at cell–cell adhesion sites between mitotic and interphase cells, we seeded E-cadherin-GFP expressing MDCK cells on Matrigel, grew them to confluency, and stained them for β1 integrins. In interphase MDCK cells, we observed β1 integrins at the basolateral membrane, which colocalized with E-cadherins (Fig. 8d). Similarly, in mitotic MDCK cells, β1 integrins localized along the cell–cell interface and colocalized with E-cadherins. To address whether the ratio between β1 integrins and E-cadherins change at cell–cell contact between two interphase cells or between an interphase cell and a mitotic cell, we performed a ratiometric analysis (Fig. 8d). We found a large variation of ratios between β1 integrins and E-cadherin GFP across different regions of the cell membrane and between different cells independent of whether they connected two interphase cells or an interphase and a mitotic cell. However, the ratiometric maps do not indicate a clear change in the ratios between β1 integrins and E-cadherin GFP in mitotic and non-mitotic cells. To further investigate the influence of integrin-ECM binding on cell–cell adhesion forces, we attached interphase HeLa cells to Matrigel-coated cantilevers for 30 min

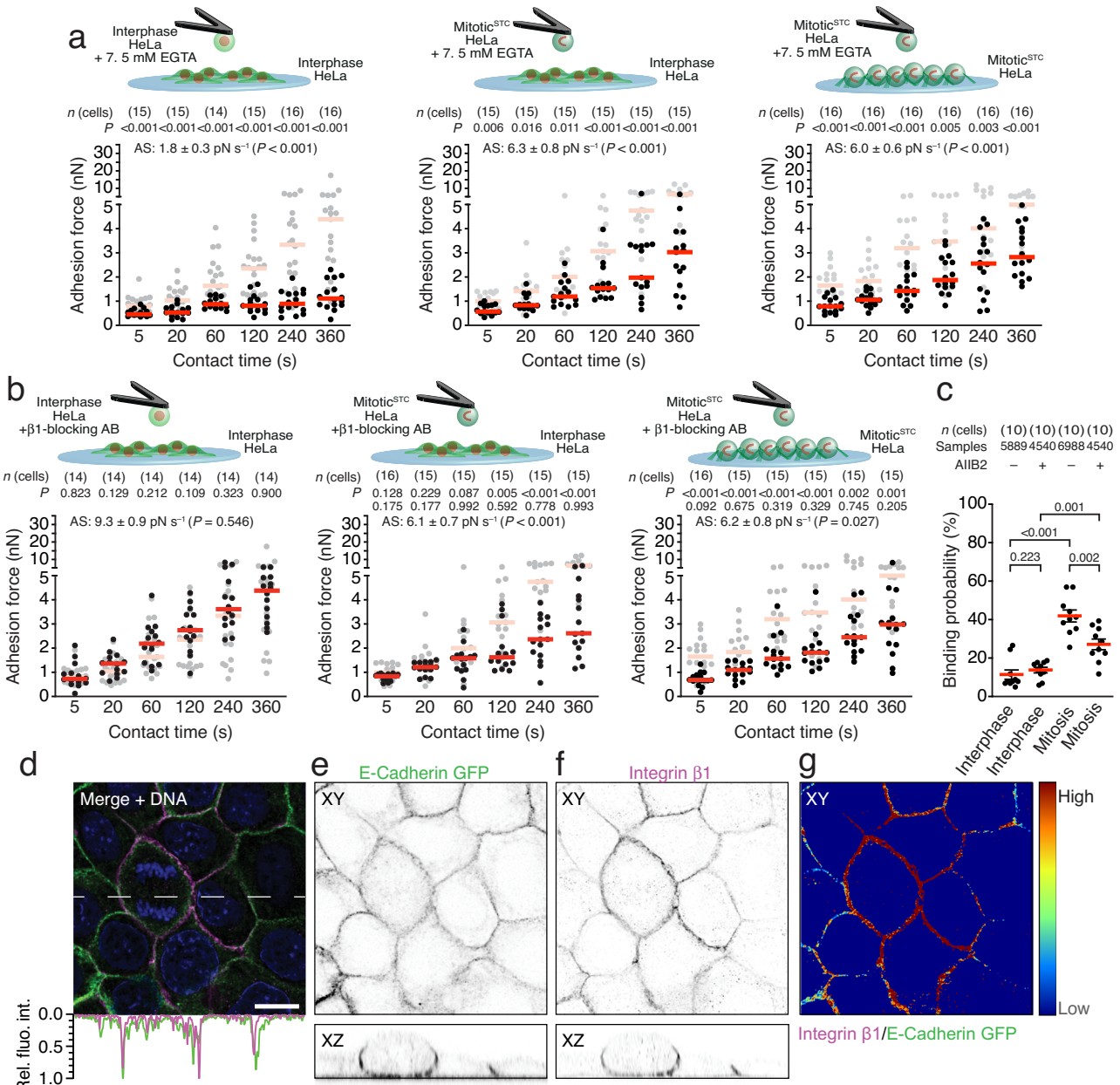

**Fig. 8 | β1 integrins promote mitotic cell–cell adhesion. a** Adhesion forces between two interphase (left), mitotic[STC] and untreated interphase HeLa cells (middle) or two mitotic[STC] HeLa cells (right) the presence of EGTA. **b** Adhesion forces between an AIIB2-treated and untreated interphase (left), AIIB2-treated mitotic[STC] and untreated interphase (middle) or AIIB2-treated and untreated mitotic[STC] (right) HeLa cells. Dots represent adhesion forces between single cantilever-bound and substrate-spread cells, red bars their medians and *n*(cells) the number of independent cells in at least three independent experiments. As reference adhesion forces established between two wildtype HeLa cells in the respective condition are given in gray (Fig. 1g). The second row *P*-values compare displayed adhesion forces with adhesion forces between TKO HeLa cells and wildtype HeLa cells in the same conditions (Fig. 4b). The adhesion strengthening rate (AS) is the slope (±SE) of a linear fit through adhesion forces for all contact times with the *P*-value comparing the AS-value to the reference data set (Supplementary Figs. 9 and 10). **c** Binding probability of untreated (data taken from Fig. 2e) or AIIB2-treated interphase and mitotic[STC] HeLa cells to untreated interphase HeLa cells.

Dots represent the binding probability between individual HeLa cells, red bars indicate the median binding probability. *n*(cells) indicates the number of HeLa cells probed and *samples* the number of force-distance curves recorded for each condition. *P* values on bars compare given conditions. *P* values comparing adhesion forces and binding probabilities were calculated using two-tailed Mann–Whitney tests and *P* values comparing AS-values were calculated by two-tailed extra sum of squares *F*-tests. **d**–**g** Representative immunofluorescence images of MDCK cells seeded on a Matrigel-coated substrate (*n* = 14 images). **d** Merged image of cells expressing E-Cadherin GFP (green; **e**), and stained for β1 integrin (magenta; **f**) and DNA (blue) using β1 integrin-blocking antibodies (AIIB2) and SPY650-DNA, respectively ("Methods"). The dashed line in **d** indicates the section used for intensity plot (**d**, bottom) and orthogonal views (**e**, **f**, bottom). For line profiles, intensities were normalized to the highest intensity for the respective channel. **g** False color image displaying the ratios of intensity signals of β1 integrin and E-cadherin GFP. Scale bar, 10 μm.

and approached them to interphase or mitotic[STC] HeLa cells that were cultured on Matrigel for >12 h (Supplementary Fig. 12c). Interphase HeLa cells attached to Matrigel established similar cell–cell adhesion forces as HeLa cells attached to ConA, indicating that Matrigel-bound integrins do not affect cell–cell adhesion at the contact times tested.

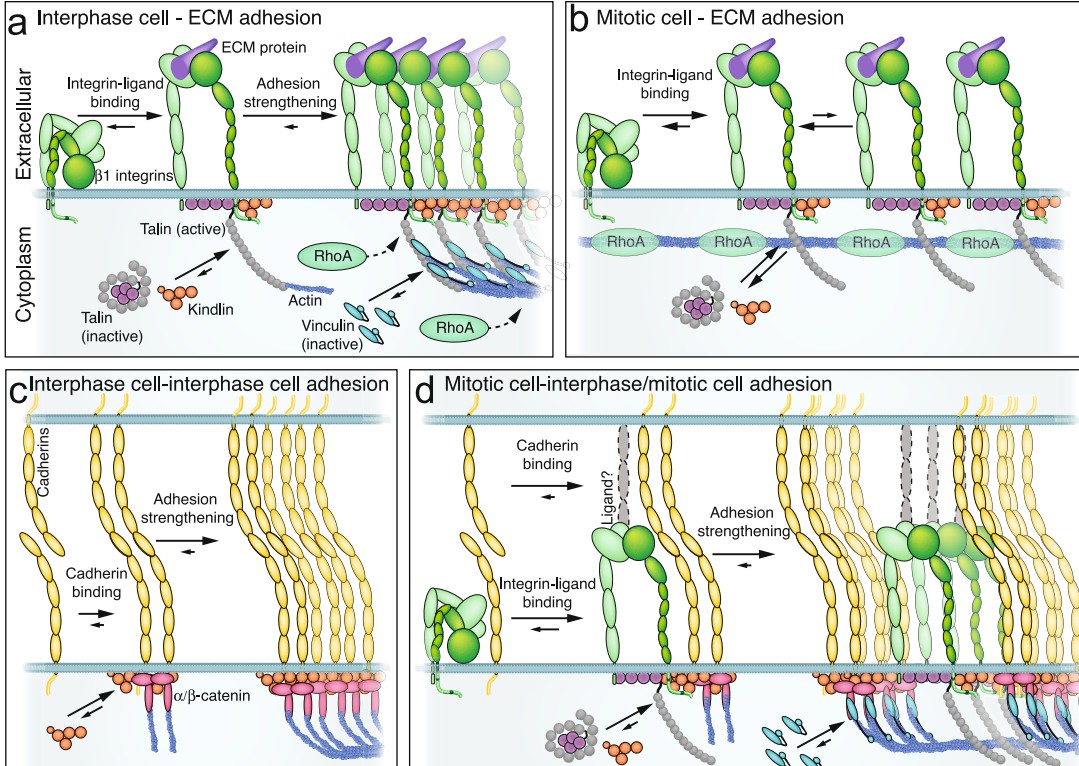

**Fig. 9 | Mitotic cells drastically reduce integrin-mediated cell−ECM adhesion and employ integrins and adhesome proteins to strengthen cell−cell adhesion.** **a** In interphase, integrins rapidly bind ECM ligands and recruit kindlin and talin, which both connect integrins during adhesion strengthening to the contractile actomyosin cortex. During integrin clustering, vinculin is recruited to the adhesion site and reinforces the integrin-actin connection by binding to talin and actin. **b** Mitotic cells initiate integrin-mediated adhesion to the ECM similarly to interphase cells, which depends on kindlin and the talin1-head domain. However, mitotic cells drastically reduce adhesion strengthening and do not connect integrin to actin via the talin-rod domain and vinculin. **c** Interphase cells initiate and strengthen cell−cell adhesion in which they involve kindlin. **d** Mitotic cells increase adhesion strengthening to interphase or mitotic cells, which involves rapid cadherin-cadherin interactions, a faster recruitment of vinculin, as well as the recruitment of β1 integrins, kindlin and the talin1-head domain to cell−cell adhesion sites.

## Discussion

The purpose of our study was to investigate how cells upon entering mitosis initiate cell−ECM and cell−cell adhesion. We reveal a dual role for integrins in mitotic cells: newly ligand-bound integrins are not coupled to actin and hence poorly strengthen adhesion to ECM and β1 integrins reinforce the adhesion of mitotic cells to neighboring cells (Fig. 9). Interphase cells employ kindlin, talin, and vinculin to rapidly initiate and strengthen integrin-mediated cell−ECM adhesion (Fig. 9a). Although kindlin and talin are essential for interphase cell−ECM adhesion[34,35], we found that kindlin and the THD suffice to initiate adhesion, but that cells require the connection of integrins to actin via the talin-rod domain and the reinforcement of the integrin-actin connection by vinculin to strengthen adhesion within seconds[36,37]. Since our experiments were not designed to reveal a potential chronology, we could not distinguish whether talin and vinculin form a complex prior to integrin recruitment[38–40] or whether vinculin is recruited after the mechanosensitive cryptic binding sites of talin are exposed by actomyosin-mediated forces[41].

Our experiments further reveal that the kindlin- and talin-dependent cell−ECM adhesion initiation is indistinguishable in mitotic and interphase cells (Fig. 9b). We observe a ~50% reduction of β1 integrins in an active conformation, which is likely due to the recently reported reduction of kindlin levels to ~20% in mitotic cells[4]. However, mitotic cells only marginally decrease their binding probability and their adhesion forces in the first 20 s contact time remain similar as for interphase cells, indicating that reduced kindlin levels and lower number of active integrins do not majorly affect how cells initiate adhesion. Importantly, adhesion strengthening is drastically reduced

in mitotic cells, which results in transient integrin-ligand binding and hence poor adhesion maturation. The observation that the talin-rod domain and vinculin are not able to strengthen newly formed mitotic cell−ECM adhesions, shows that in mitosis new connections between integrins and cortical actomyosin are not established. Further, destabilizing the stiff and contractile actomyosin cortex of mitotic cells does not increase adhesion strengthening. The inability of mitotic cells to couple integrins to the cortical actomyosin prevents actomyosin-generated force transduction to cell−ECM adhesion sites, which is required to stabilize the adhesion sites, and hence supports mitotic cell rounding[42,43]. Since mitotic cells initiate adhesion to the ECM similar as interphase cells, the inability of integrins to cluster and/or to recruit signaling and actin polymerizing proteins likely maintain mitotic cells round. Further, our results suggest that integrin-mediated adhesion sites that anchor retraction fibers of mitotic cells to the ECM substrate[3,4,10,29] are remodeled focal adhesions. Additionally, since we find that mitotic cells also display drastically lower adhesion forces to vitronectin, we conclude that reticular adhesions that apparently anchor the mitotic cell body to vitronectin[44] are established during cell rounding rather than during mitosis.

Although we observe that β1 integrin activity is reduced during mitosis compared to interphase cells, overexpression of constitutively-active Rap1A and Mn²⁺-treatment, which prevent mitotic cell rounding of adherent cells[7,8], increase β1 integrin activity of suspended interphase and mitotic cells but exert different effects on their cell−ECM adhesion strengthening. While Mn²⁺ increases the adhesion of interphase cells to ECM, it has no effect on mitotic cell−ECM adhesion. In contrast, expression of constitutive-active Rap1A, which releases the

autoinhibition of talin and recruits it to the membrane[30], increases cell–ECM adhesion strengthening in mitosis but not in interphase. Interestingly, THD expression in talin deficient cells does not increase cell–ECM adhesion forces in mitosis, however, adhesion forces of mitotic Rap1A-CA expressing HeLa cells and interphase THD expressing HeLa cells are very similar. This indicates that Rap1A-CA activates talin and recruits it to the membrane, which, however, fails to connect integrins to the cytoskeleton, or that Rap1A-CA triggers integrin-ECM binding in a talin-independent manner. Although our experiments do not provide mechanistic insights into how Rap1-CA promotes integrin mediated ECM binding of mitotic cells, the experiments show that non-regulatable Rap1A majorly affects the adhesion of mitotic cells.

Our experiments show that mitotic cells increase their adhesion strengthening to neighboring cells (Fig. 9c, d), which is partially due to the ~20% increased cell surface level of cadherins and a two-fold increase of cadherin-binding rate. Interestingly, we also found that β1 integrins promote the initiation and strengthening of mitotic adhesion to adjacent interphase or mitotic cells. We did not detect collagens, laminins, or fibronectin on the cell surface of interphase or mitotic cells in our experiments, indicating that integrins involved in cell–cell adhesion of mitotic cells are unlikely to bind to ECM proteins on the cell surface of other interphase or mitotic cells. However, we cannot fully exclude the participation of ECM proteins in mitotic cell–cell adhesion experiments. Whether the contribution of β1 integrins is accomplished through direct binding to E- and/or N-cadherin, as reported for the collagen-binding α2β1 or αEβ7 integrins[45–48], remains to be explored. The observation that extrinsic activation of β1 integrins by $Mn^{2+}$ or by antibodies does not increase mitotic cell–cell adhesion may indicate that β1 integrins function conformation independent or that activation of integrins does not increase their binding dynamics. Although β1 integrins do not contribute to adhesion formation between two interphase cells within the first 360 s, they are localized to cell–cell contacts in confluent MDCK cell monolayers regardless of the cell–cycle state[49,50].

While kindlin promotes cell–cell adhesion in interphase and mitosis, talin promotes cell–cell adhesion strengthening exclusively during mitosis. The apparent similarity of this finding to that of integrin involvement in mitotic cell–cell adhesion suggests that integrins require talin to support mitotic cell–cell adhesion. Vinculin, which has been shown to accumulate exclusively at cell–cell adhesion sites of interphase cells that are directly adjacent to mitotic MDCK cells[51], also contributes to increase adhesion between mitotic and neighboring cells and between the two daughter cells before spreading. Interestingly, although vinculin is required to reinforce and regulate the force transduction of mature cell–cell contacts in interphase cells[52–55], our data shows that vinculin does not participate in establishing adhesion between two interphase cells within the first 360 s of cell–cell contact.

In summary, cells decrease cell–ECM adhesion at the onset of mitosis, which leads to cell rounding and limited need for integrins and adhesome proteins. At the same time, mitotic cells enhance adhesion to adjacent cells by activating cadherin and utilizing non-engaged integrin, kindlin, talin, and vinculin, at cell–cell adhesion sites. This intricate remodeling of cell–ECM and cell–cell adhesion sites ensures mitotic cell rounding and the maintenance of tissue integrity.

## Methods

### Cell culture

Wildtype HeLa (Kyoto) (a kind gift of A. Hyman, MPI Molecular Cell Biology and Genetics, Germany), talin 1/2-depleted (TKO), kindlin 1/2-depleted (KKO), vinculin-depleted (VKO), TKO HeLa cells re-expressing the talin1-head domain (TKO + talin1-head), TKO HeLa cells re-expressing talin1 (TKO + talin1), kindlin2 re-expressing (KKO + kindlin2), vinculin re-expressing (VKO + vinculin), Rap1A-constitutively-active (Rap1A-CA) HeLa cells as well as wild type fibroblasts[56], talin1/2-

depleted (TKO) fibroblasts[35], kindlin 1/2-depleted (KKO) fibroblasts[35] and TKO fibroblasts expressing talin1-head (a kind gift of C. Grashoff, University of Münster), HEK293T cells (kind gift of B. Roska, Institute of Molecular and Clinical Ophthalmology Basel) and MCF7 cells (purchased from ATCC) were cultured in Dulbecco's Modified Eagle Medium (DMEM, 31966047, Thermo Fisher Scientific), supplemented with 10% (vol/vol) fetal bovine serum (FBS, F9665, Sigma Aldrich) and 100 U ml⁻¹ penicillin and 100 μg ml⁻¹ streptomycin (15140122, Thermo Fisher Scientific). VKO + vinculin HeLa cells were cultured with additional selection antibiotics of 100 U ml⁻¹ geneticin (10131027, Thermo Fisher Scientific). Histone-2B and myosin heavy-chain labeled HeLa cells (HeLa MYH9-GFP, H2B-mCherry, kind gift of A. Hyman) were cultured with additional selection antibiotics of 100 U ml⁻¹ geneticin (10131027, Thermo Fisher Scientific) and 0.5 μg ml⁻¹ puromycin dihydrochloride (A1113803, Thermo Fisher Scientific). MDCK (kind gift of B. Roska) and histone-2B-eGFP and actin-mCherry labeled MDCK cells (MDCK H2B-GFP, actin-mCherry)[57] were cultured in minimal essential medium (MEM, 11095080, Thermo Fisher Scientific) supplemented with 5% (vol/vol) FBS and 100 U ml⁻¹ penicillin and 100 μg ml⁻¹ streptomycin (15140122, Thermo Fisher Scientific).

### Cell line engineering

To deplete HeLa (Kyoto) cells or HeLa (Kyoto) cells expressing MYH9-GFP and H2B-mCherry from talin1/2, kindlin1/2 and vinculin, CRISPR/Cas9 was used[58]. Short-guide RNAs (sgRNAs) for respective genes in the human genome (GRCh38/hg38) were identified with the online CRISPR design tool (http://crispor.tefor.net)[59]. sgRNAs were selected for high target score, low off-target score, high CFD and efficiency score (Supplementary Table 2). 5'-phosphorylated forward and reverse DNA sequences encoding sgRNAs were hybridized, digested by Bbs1 (R0539S, New England Biolabs) and ligated into the plasmid pSpCas9-2A-BFP (px458.2)[60], using T4 polymerase (M0203S, New England Biolabs). Plasmids targeting TLN1, FERMT1 or VCL were transiently transfected into HeLa (Kyoto) or HeLa MYH9-GFP H2B-mCherry cell lines using lipofectamine 2000 (11668019, Invitrogen) according to the manufacturer's instruction. 48 h post transfection floating, but viable talin1-KO or kindlin1-KO HeLa cells were transiently transfected with px458.2 plasmids targeting TLN2 or FERMT2, respectively. After two days, single high-BFP-expressing cells were sorted into Matrigel-coated (354234, Corning) 96-well plates using a fluorescence-activated cell sorter (Fortessa, BD Bioscience) and were propagated as described above. Protein depletion in single cell colonies was verified using western blotting. Cells were lysed using SDS sample buffer (125 mM Tris (pH 6.8), 4% (wt/vol) SDS, 20% (vol/vol) glycerol, 0.01% (vol/vol) bromophenol-blue, 10% (vol/vol) β-mercapto ethanol) and the supernatant was loaded onto a 4–12% SDS-PAGE gel (NW04125BOX, Invitrogen) and subsequently transferred onto nitrocellulose paper (Amersham Protran 0.45 μm NC, 10600002, GE Healthcare). Protein expression was analyzed using antibodies against talin1/2 (1:500, ab11188, Abcam), kindlin1 (1:1000, ab68041, Abcam), kindlin2 (1:1000, ab74030, Abcam), and vinculin (1:500, ab18058, Abcam). A GAPDH antibody (1:1000, #2118 S, Cell Signaling) was used as loading control. As secondary antibodies, goat anti-mouse antibody (αMouse IgG HRP-conjugated, 1:2000, #1706516, BioRad) and goat anti-rabbit antibody (αRabbit IgG HRP-conjugated, 1:5000, #1706515, BioRad) were used. Antibody-treated nitro cellulose papers were then analyzed for chemiluminescence using a FUSION PULSE TS imaging system (Vilber).

To re-express adhesome protein depleted HeLa cell lines, VKO HeLa cells were transiently transfected with pEGFP-mouseVinculin (#67935, Addgene) using lipofectamine 2000; TKO HeLa cells were retrovirally transfected with talin-head (pLCPCmod-hTalin1-head-Ypet) or full-length talin1 (pLCPXmod+hTalin−1(wt)-Ypet(short)) and KKO HeLa cells with kindlin2 (pRetroQ-mCherry-mKindlin2) using a standardized protocol[61]. Using the same protocol HeLa (Kyoto) cells were retrovirally transfected with a constitutively active form of Rap1A

(Rap1-V12, pRetroQ-eGFP-hRap1A-V12). For virus production, respective transfer plasmids were transfected together with envelope plasmid pMD2.G (12259, Addgene) and packaging plasmid pUMVC (8449, Addgene) into HEK 293 T cells using lipofectamine 2000. After 14 h post transfection the media was discarded and replaced with 5 ml of fresh culture media. The cell media conditioned with virus was collected after 8 h and 16 h, replaced by fresh cell culture media and stored at 4 °C. Collected media was centrifuged at ~3400 × $g$ for 5 min and filtered using a PVDS 0.22 μm syringe-filter (FBS30PVDF022HS, Filter-Bio). HeLa cell lines were grown in six-well plates to a confluency of ~33%, their media was aspirated and 2 ml of virus containing media was added to each well to transfect with respective retroviruses. For cell propagation virus containing media was used. 48 h post after the first addition of virus containing media, cells were sorted for high expression of the respective fluorescence protein and propagated in virus-free medium.

To visualize paxillin, HeLa (Kyoto) cells were transfected with plasmids encoding for paxillin-GFP (#50529, Addgene) and lifeact-mCherry (#54491, Addgene) using lipofectamine 2000.

### Cantilever and support functionalization

Tipless AFM cantilevers (NP-O, Bruker) were plasma cleaned (PDC-32G, Harrick Plasma) and incubated overnight with 2 mg ml$^{-1}$ concanavalin A (ConA, C2010-100MG, Sigma Aldrich) in PBS or with 2% (vol/vol) Matrigel (354234, Corning) in DMEM at 4 °C overnight. For substrate coating two- and four-segmented PDMS masks were attached onto the glass bottom of a petri dish (FD35, WPI)[62]. PDMS-segmented surfaces were incubated with 25 μl of collagen I (50 μg ml$^{-1}$, PureCol, 5005, Advanced Biomatrix), collagen IV (50 μg ml$^{-1}$, C6745, Sigma-Aldrich), laminin mix (50 μg ml$^{-1}$, L2020, Sigma Aldrich), vitronectin (50 μg ml$^{-1}$, CC080, Millipore), fibronectin (50 μg ml$^{-1}$, 341631, Sigma Aldrich), FNIII7-10 (50 μg ml$^{-1}$)[27], FNIII7-10ΔRGD (50 μg ml$^{-1}$)[27], or 2 % (wt/vol) bovine serum albumin (BSA, Sigma Aldrich) in PBS and incubated at 4 °C overnight. Segments were coated with 2% (vol/vol) Matrigel (354234, Corning) in DMEM by incubation for one hour at room temperature. Subsequently, the dish was washed with PBS and filled with 2 ml of 2% BSA to block any uncoated surfaces.

### SCFS setup and cell preparation

Single cell force spectroscopy (SCFS) was performed using an AFM-based CellHesion200 (JPK Instruments) and a motorized stage (JPK Instruments) or an AFM (NanoWizard II, JPK Instruments) equipped with a CellHesion-module (JPK Instruments) mounted on inverted optical microscopes (both AxioObserver Z1, Zeiss). The CellHesion200 setup was placed in a heat chamber (The Cube, Life Imaging Services) to maintain ambient temperature at 37 °C, the CellHesion-module was equipped with a PetriDish-Heater to maintain SCFS media at 37 °C. SCFS was performed using 200 μm-long, tip-less, V-shaped silicon nitride cantilevers (NP-O, Bruker) with a nominal spring constant of 0.06 N m$^{-1}$. The exact spring constant of each used cantilever was calibrated prior to experiments using the thermal noise method[54]. Cells were grown in 12-well plates to a maximal confluency of ~80%. To enrich for mitotic cells 2 μM (+)-S-trityl-L-cysteine (STC, 164739-5 G, Sigma Aldrich) was added 12 h prior to experiments to cells with appropriate confluency to arrest them in prometaphase. Before experiments interphase cells were washed with PBS and detached with 200 μl 0.25% (wt/vol) trypsin/EDTA (25200072, Thermo Fisher Scientific) for 2 min at 37 °C. Mitotic cells were harvested by mitotic cell shake off. Detached cells were suspended in SCFS medium (DMEM; 12800017, Thermo Fisher Scientific) supplemented with 20 mM HEPES (A3724, Applichem), and 10% (vol/vol) FBS. Cells were pelleted and resuspended in 200 μl SCFS media, for experiments with mitotic cells additionally 2 μM STC was added to resuspended cells. After detachment, cells were allowed to recover for 30 min from trypsin/EDTA treatment in SCFS medium at 37 °C (ref. 63). Substrate-coated glass

bottom petri dishes were washed with PBS to remove excess protein and filled with 3 ml SCFS medium. For experiments with mitotic cells 2 μM STC was added to SCFS medium. For manganese-activated experiments, cells were incubated with 0.5 mM MnCl$_2$ in SCFS medium 30 min before and throughout SCFS. To attach single cells to the apex of a ConA- or Matrigel-coated cantilever, suspended single cells were pipetted into BSA-coated areas of petri dishes. The cantilever was lowered onto a single cell with 10 μm s$^{-1}$ until detecting a contact force of 5 nN. After 5 s contact time the cantilever was retracted at 10 μm s$^{-1}$ by >90 μm to fully separate cell and substrate. Cells were incubated for 10 min on the cantilever to ensure firm binding. Cells of similar size and morphology were attached to cantilevers to minimize variation and cell morphology was monitored throughout experiments using optical microscopy to ensure a round morphology cantilever bound cells. Where expressed, histone-2B-mCherry labeling was used to assess the cell cycle state, i.e., interphase or mitosis, prior and throughout SCFS.

### Cell–ECM SCFS

Cell–ECM adhesion forces were quantified by approaching cells to substrate-coated PDMS supports with 5 μm s$^{-1}$ until a contact force of 1 nN was reached. The cantilever was maintained at constant height for contact times of 5, 20, 60, 120, 240, or 360 s. The order of contact times was randomized for each cell to exclude memory effects of the cells on experimental sequences. Thereafter, the cantilever-bound cell was retracted from the substrate at 5 μm s$^{-1}$ for 100 μm until cell and substrate were fully separated. After the experimental cycle, cells were allowed to recover from adhesion measurement for the time of the contact time before measuring the adhesion force for a different contact time. The area on the substrate was altered after every adhesion force measurement cycle. Adhesion forces of cantilever bound cells were quantified for all contact times unless morphological changes, such as cell spreading or cell division, were detected. The order of contact times was randomized. Adhesion forces were determined after the retraction force-distance curves were drift- and baseline corrected using the JPK data analysis software (JPK Instruments). Adhesion force strengthening was determined as the slope of linear fits to all adhesion forces and contact times (PRISM).

### Long-term cell–ECM SCFS combined with confocal microscopy

To conduct long-term cell–ECM SCFS a CellHesion 200 equipped with a PetriDish Heater (all JPK) were mounted onto an inverted AxioObserver Z1 equipped with a confocal microscopy module (LSM700, all Zeiss) and a ×63 water-immersion objective (LCI Plan-Neofluar ×63/1.3 Imm Corr DIC M27, Zeiss) and ZEN acquisition software (ZEISS). Interphase and mitotic$^{STC}$ HeLa (Kyoto) cells over-expressing paxillin-GFP and (described in "Cell line engineering") were cultured, prepared and attached to a ConA-coated cantilever as described above. Cantilever-attached cells were approached to Matrigel-coated PDMS-supports with 5 μm s$^{-1}$ until a contact force of 1 nN was reached, thereafter the cantilever was maintained at constant height for 60 min. During the contact time paxillin-GFP localization at the cell–ECM interphase was monitored every 2 min using confocal microscopy. Laser intensities and gains were optimized for each cell used prior to experiments. After 60 min, the cantilever was retracted to quantify cell adhesion forces. After every experiment, the cantilever and the cantilever-bound cell were exchanged. To determine the spreading area of HeLa cells, the paxillin-GFP fluorescence signal was analyzed using Fiji (Version 2.1.0/1.52c)[64]. Thereto, contrast and brightness were adjusted and a Gaussian blur with $\sigma = 2$ was applied to each image of the time lapse. A pixel-intensity threshold was applied to differentiate whether a pixel depicted a part of the contact area. Pixels within the contact area that did not meet the threshold criteria were manually added by the "fill holes" option. The spreading area was determined using the "analyze particles" option.

## Cell–cell SCFS

To conduct cell–cell adhesion experiments on interphase cells, suspended cells were pipetted into the experimental dish onto a Matrigel-coated area. Cells were allowed to spread for at least 30 min. To perform cell–cell adhesion experiments on mitotic cells, experimental dishes were prepared before experiments and cells seeded onto Matrigel coated areas. Cells were allowed to spread, subsequently 2 µM STC was added 12 h prior to experiments to arrest cells in prometaphase. Single interphase or STC arrested cells, prepared as described above, were attached to cantilevers. For SCFS with HeLa cells attached to Matrigel-coated cantilevers, single, suspended interphase HeLa cells were attached to and incubated on the cantilevers for 30 min prior to experiments. HeLa cells were seeded onto Matrigel >20 h prior to experiments and mitotic cells were enriched by 2 µM STC for 12 h. Cell–cell adhesion experiments were performed between the cantilever-bound cell and single cells attached on the substrate and were performed as described for cell–ECM SCFS. After each adhesion force measurement, a different substrate-attached cell was selected. Adhesion forces and strengthening were determined as described above.

## SCFS with antibodies, chemicals, or perturbants

For SCFS in presence of a $\beta1$ integrin blocking antibody (clone AIIB2, DSHB) or $\beta1$ integrin activating antibody (clone 12G10, ab30394, abcam) cells were incubated with a 1:100 (vol/vol) dilution (in SCFS medium) for at least 30 min prior to the experiments in SCFS medium. For cell–ECM SCFS AIIB2 was present throughout the experiments at a 1:1000 (vol/vol) dilution in SCFS medium. For cell–cell adhesion experiments 12G10 was present throughout the experiments at a 1:1000 (vol/vol) dilution in SCFS medium. For SCFS in the presence of $Mn^{2+}$ or EGTA, suspended cells were incubated with 0.5 mM $MnCl_2$ or 7.5 mM EGTA in SCFS medium for at least 30 min and the chemicals were present throughout the experiments. For inhibition of RhoA cells were incubated with 0.5 µg ml$^{-1}$ Rho inhibitor I (# CT04, Cytoskeleton) in the culture flask for 4 h prior to their detachment. The inhibitor was present during the recovery of the detachment and during SCFS. For cytochalasin D, suspended cells were incubated with 0.1, 0.5, or 1 µM cytochalasin D (C8273, Sigma Aldrich) in SCFS medium for at least 30 min prior to the experiments. The inhibitor was present throughout the experiments.

## Adhesion probability assay

To determine the binding probability of cantilever-bound cells to their substrate with single molecule sensitivity, adhesion probability assays were performed using an AFM (NanoWizard II) mounted on an inverted optical microscope (AxioObserver Z1). SCFS media was maintained at 37 °C using a PetriDish Heater (JPK Instruments). Cells and supports were prepared as described above. Cells were attached onto the tip of a ConA-coated cantilever as described above. For cell–ECM binding probability quantification cells were approached to Matrigel-coated supports with 3 µm s$^{-1}$ until a contact force of 150 pN was recorded. Immediately after reaching the contact force, the cantilever-bound cell was retracted with 3 µm s$^{-1}$ resulting of a contact time between cell and substrate of ~120 ms. After the cell was retracted from the substrate for 10 µm, the cantilever-bound cell was allowed to recover for 0.5 s, the contact area on the substrate was altered and another experimental cycle was initiated. This experimental cycle was repeated for a maximum of 1 h (in total up to 425 experimental cycles) for each cantilever-bound cell. For cell–cell binding probability, substrate cells were prepared as described in cell–cell SCFS and experimental parameters were as described for cell–ECM binding probability. For cell–cell binding probability experiments the contact time resulted to be ~150 ms. The same cell on the substrate was used for all force-distance curves acquired for a single cantilever bound cell. Binding probabilities were calculated as the ratio of collected force-distance curves showing a single unbinding event and all force-distance curves recorded (Supplementary Fig. 1c).

## Flow cytometry assays

Interphase cells were detached from culture flasks using 0.25% (wt/vol) trypsin/EDTA for 2 min at 37 °C. Cells were resuspended in culture medium, allowed to recover from trypsin treatment for 30 min. To perform flow cytometry with mitotic cells, 2 µM STC was added to cultured cells 12 h prior to and was present throughout experiments. Mitotic cells were harvested by through washing with medium. For flow cytometry experiments in the presence of $Mn^{2+}$, cells were incubated with 0.5 mM $MnCl_2$ for 30 min in Dulbecco's Modified Eagle Medium (DMEM, 31966047, Thermo Fisher Scientific), supplemented with 100 U ml$^{-1}$ penicillin and 100 µg ml$^{-1}$ streptomycin (15140122, Thermo Fisher Scientific) and with 0.5 mM $MnCl_2$ present throughout the experiment. -125,000 cells per sample were pelleted and washed twice with ice-cold flow cytometry buffer (PBS supplemented with 2 mM EDTA and, 0.1% BSA (wt/vol)). Flow cytometry experiments with antibody against extended $\beta1$ integrins (clone 9EG7) were performed with an adjusted flow cytometry buffer (PBS supplemented with 1 mM $CaCl_2$, 1 mM $MgCl_2$ and 3% BSA (wt/vol)). Antibodies against CD51 (integrin $\alpha V$, 327909, BioLegend, PE-conjugated), CD49a (integrin $\alpha1$, 328303, BioLegend, PE-conjugated), CD49b (integrin $\alpha2$, 359307, BioLegend, PE-conjugated), CD49c (integrin $\alpha3$, 343803, BioLegend, PE-conjugated), CD49d (integrin $\alpha4$, 304303, BioLegend, PE-conjugated), CD49e (integrin $\alpha5$, 328009, BioLegend, PE-conjugated), CD49f (integrin $\alpha6$, 313607, BioLegend, Alexa488-conjugated), CD29 (integrin $\beta1$, 303015, BioLegend, Alexa488-conjugated), CD18 (integrin $\beta2$, 366305, BioLegend, FITC-conjugated), CD61 (integrin $\beta3$, 336403, BioLegend, FITC-conjugated), CD104 (integrin $\beta4$, 327807, BioLegend, PE-conjugated), CD29 clone 9EG7 (553715, BD Bioscience, un-conjugated), CDH1 (E-Cadherin, 147306, BioLegend, Alexa594-conjugated), CDH2 (N-Cadherin, 350806, BioLegend, PE-conjugated), CDH5 (VE-Cadherin, 336403, BioLegend, PE-conjugated) were diluted 1:10 in ice-cold flow cytometry buffer. Pelleted cells were resuspended in 50 µl flow cytometry buffer containing the respective antibodies and incubated for 1 h on ice for all pre-conjugated antibodies. Cells incubated with the CD 29 clone 9EG7 antibody were incubated on ice for 30 min with the primary antibody, washed twice with flow cytometry buffer and incubated with 50 µl of 1:10 PE-conjugated IgG2a antibody (407507, BioLegend). Following antibody incubation cells were washed twice with ice-cold flow cytometry buffer and finally resuspended in 250 µl flow cytometry buffer supplemented with 0.25 µg DAPI (422801, BioLegend) and kept on ice. Fluorescence intensities of single cells were analyzed using a flow cytometer (Fortessa, BD Bioscience). Laser intensities were optimized for each experiment and maintained constant for conditions that were compared. Flow cytometry data was gated according to forward and side scatter to exclude debris and doublets, as well as dead cells indicated by the DAPI signal (Supplementary Fig. 13). Median fluorescence intensities were determined using the Fortessa software (BD FACSDiva 8.1).

## Confocal laser scanning microscopy

To detect ECM proteins on the cell surface of mitotic$^{STC}$ or interphase HeLa cells, µ-slides 8 well chambered coverslips (Ibidi) were functionalized with 2 mg ml$^{-1}$ ConA (C2010-100MG, Sigma Aldrich) in PBS or 2% (vol/vol) Matrigel (354234, Corning) in DMEM over night at 4 °C or for 1 h at room temperature, respectively. Rounded interphase or mitotic$^{STC}$ HeLa cells were seeded on ConA for 30 min to mimic cantilever-bound HeLa cells. To mimic substrate bound cells, interphase HeLa cells were seeded on Matrigel for 2 h or for 12 h while arresting them in mitosis using STC. HeLa cells were fixed using 4% paraformaldehyde in PBS (28908, Life Technologies) for 15 min at RT. Fixed cells were blocked with 3% (w/v) BSA in PBS. Subsequently,

samples were incubated with 1:50 (vol/vol) diluted anti-collagen I (ab88147, Abcam, UK), 1:50 (vol/vol) diluted anti-collagen IV (PA128534, ThermoFischer, Switzerland), 1:20 (vol/vol) diluted anti-fibronectin (ab2413, Abcam, UK) or 1:100 (vol/vol) diluted anti-laminin (ab256380, Abcam, UK) antibodies. Secondary anti-rabbit Alexa488 (ab150077, Abcam, UK) and anti-mouse Alexa488 (ab150113, Abcam, UK) were diluted 1:200 (vol/vol) and incubated for 1 h. The samples were washed three times with PBS after each step. Cells were incubated with DAPI (1:1000 diluted) and SirActin (SC001, Spirochrome, Switzerland, 1:1000 diluted) for 1 h at room temperature. All antibodies and staining reagents were diluted in PBS containing 3% BSA.

To localize integrins, µ-slides 8 well chambered coverslips (Ibidi) were functionalized with 2% (vol/vol) Matrigel (354234, Corning) in DMEM by incubation for one hour at room temperature. MDCK cells expressing E-Cadherin GFP were trypsinized and cell suspension in MEM was pipetted into the functionalized wells and cultured for 48 h. Cells were fixed with 4% paraformaldehyde in PBS (28908, Life Technologies) for 15 min at RT, followed by quenching with 0.2 M glycine (Merck) in PBS for 20 min at RT. Fixed MDCK cells were permeabilized with 0.1% Triton X-100 (Merck) for 10 min at RT and blocked with 0.5% BSA (Merck) in PBS (blocking buffer) for 30 min at RT. Samples were incubated with anti-integrin β1 antibody (AIIB2, DSHB) with the dilution of 1:100 (vol/vol) in blocking buffer overnight at 4 °C. Secondary anti-rat AlexaFluor 555 (ab150158, Abcam) was diluted 1:200 (vol/vol) in blocking buffer and incubated for 1 h at RT. The samples were washed three times with PBS after every step. Nuclear staining was carried out by incubating the samples with a 1× solution of Spy650-DNA (SC501, Spirochrome) in PBS for 1 h at RT.

The labeling solution was then substituted by PBS, and the samples were analyzed with a Zeiss LSM980 point scanning confocal microscope (Zeiss) using a C-Apochromat ×40/1.1 W Korr M27 objective (Zeiss). Signals were collected sequentially in Airyscan super-resolution mode. The images were processed with ZEN Blue software (Zeiss), 3D Airyscan processing was performed using default settings, and image analysis was performed with Fiji[64].

For the ratiometric analysis of AIIB2-labeled β1 integrins and E-cadherin GFP, we identified the membrane signals for the ratio analysis by a colocalization analysis of the AIIB2-labeled β1 integrins and E-cadherin GFP channels in Imaris. AIIB2-labeled β1 integrins and E-cadherin GFP signals are exclusively found in cell membranes. We obtained the thresholds for segmentation automatically using the method of Costes et al.[65]. We extracted the intensities from the AIIB2-labeled β1 integrins and E-cadherin GFP channels at the positions of significant colocalization and created a new channel with the AIIB2-labeled β1 integrins/E-cadherin GFP ratios. We applied a small, $3 \times 3 \times 3$ median filter to the ratio channel to suppress local extreme ratios due to noisy pixels. For visualization, all ratios across the analyzed images were pooled and binned using the 25th (ratio = 2.0), 50th (ratio = 4.0) and 75th (ratio = 8.0) percentile and color-coded accordingly. Non-colocalizing pixels were assigned the value of 0. Out-of-focus planes were omitted from the calculation.

## Statistical analysis

Statistical tests, such as indicated in the figure legends, were performed using Prism (GraphPad Software). Data was analyzed using two-tailed unpaired, nonparametric Mann–Whitney $t$ tests. To statistically compare adhesion strengthening under different conditions, a linear regression analysis of the recorded adhesion forces was performed for all contact times (Prism). Thereto, a two-tailed extra sum of squares $F$-test was used to test whether it was superior to fit conditions separately compared to fitting all data combined with a single fit (null hypothesis).

## Reporting summary

Further information on research design is available in the Nature Portfolio Reporting Summary linked to this article.

## Data availability

The data generated in this study have been deposited in the ETH research collection available under https://doi.org/0.3929/ethz-b-000602745. All other relevant data supporting the key findings of this study are available within the article and its Supplementary Information files or from the corresponding author upon reasonable request. Source data are provided with this paper.

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

## Acknowledgements

We thank M.P. Stewart for initiating the project and fruitful discussions, S. Reddy for providing the pSpCas9-2A-BFP (px458.2) plasmid, R. Böttcher for providing the pRetroQ-eGFP-hRap1A-V12 plasmid, C.

Grashoff for providing the pLCPCmod-hTalin1-head-Ypet, pLPCXmod+hTalin–1(wt)-Ypet(short) plasmids and the TKO + talin1 head fibroblasts, P.M. Spoerri and U. Sharma for help with flow cytometry and data analysis, M. Colomer Rosell for initial experiments with MCF7 cells, and A. Ponti for help with the ratiometric analysis. We thank the singe-cell facility of the Department of Biosystems Science and Engineering, ETH Zürich for their support. This work was supported by the Swiss National Science Foundation (SNF; grant no. 31003A_182587/1) and the European Research Council (grant agreement no. 810104 - Point).

## Author contributions

N.S., D.J.M., R.F., and M.H. conceived and designed the study. M.H. created genetically engineered HeLa cell lines and performed most SCFS experiments. N.S. performed cell–ECM SCFS experiments with blocking antibodies, fibroblasts and MCF7 cells. M.H. and N.S. performed flow cytometry experiments and live-cell confocal experiments. J.C.-A. and M.H. designed and performed confocal microscopy of MDCK cells. M.H. and N.S. analyzed SCFS and flow cytometry data. J.C.-A., M.H., and N.S. analyzed confocal microscopy data. N.S., D.J.M., R.F., and M.H. evaluated experimental progress and data. All authors discussed the experiments and wrote the manuscript.

## Competing interests

The authors declare no competing interests.
