## [Peer Review File · Nature Communications]

REVIEWER COMMENTS

Reviewer #1 (Remarks to the Author):

The study by Huber et al uses single cell atomic force microscopy to measure the attachment forces between cell and ECM, and between cells. They use this methodology to investigate cell cycle dependent changes in attachments. Furthermore, they couple this approach to genetic manipulation of adhesion proteins where they reveal that Beta1 integrins strengthen adhesion between adjacent cells in mitotic cells, while ECM interactions are decreased. This work is of wide interest and addresses how cells regulate cell-cell adhesion while reducing cell-ECM adhesion during mitosis to prevent delamination of the dividing cells.

I have the following comments on the work which should be addressed before publication:

(1) Many of the AFM experiments show adhesion forces which are highly variable, this is especially prominent at longer attachment time points (e.g. Fig 1a).

a. Are there multiple cell populations which lead to a large data variance?

b. A larger number of probed cells would reveal if this is the case – I understand this is a technically challenging approach but the data variance suggests more cells should be probed to reveal the “true” spread of the data.

(2) Point 1 could be related to the methodology.

a. If you are taking the same cell attached to the cantilever and then probing for different time points, does the variance in the longer time points represent damaged cells.

b. Can you provide evidence that the cells are viable following the attachment-detachment cycle?

c. What happens if you measure a cell through a single attachment-detachment cycle at each time point?

(3) Adhesion force increases over time with the mitotic cells –

a. Can mitotic cells reach the same adhesion forces at longer time points?

b. In relation to point 2, are mitotic cells less viable/more damaged by the AFM approach therefore fail to attach with the same force at longer time points?

(4) Fig5: Could the authors comment on whether measuring cell-cell interactions in the vertical axis would differ from the forces in the lateral plane?

(5) What is the basis for using a contact force of 1 nN?

(6) Could the authors explain why time points were chosen between 5 and 360 s for measuring attachment?

(7) Could the authors comment on how this work may connect to the recent publications Gough et al JBC 2021 297 100837 which explores an interaction between Talin and CDK1.

Minor comments:

(a) It would be useful to add some numbers to the main text of the manuscript so that the reader can have an idea of the forces involved.

(b) Add molecular weight markers and full blots for the western-blots in the supplementary material.

(c) Many of the figure labels/font size is too small so they should be increased for the reader.

Reviewer #2 (Remarks to the Author):

This manuscript carefully analyzes how the adhesion forces change during cell division and cell rounding. Huber et al., made extensive use of atomic force microscopy-based single-cell force spectroscopy (SCFS) to measure cell-ECM and cell-cell adhesion forces and strengths in interphase and mitosis. Authors conclude that at the onset of mitosis, integrins diminish cell-ECM adhesion and enhance cell-cell adhesion sites. By taking CRISPR/Cas9 based gene editing, they showed that kindlin, talin and vinculin play a role in strengthening cell-cell adhesion during mitosis. The findings of this manuscript are important and will serve to the understanding of cell rounding and how the adhesion components regulate the cell morphology in a cell cycle-dependent manner. The experimental design is very nice to examine the adhesion forces in cell-ECM and cell-cell. However, this reviewer has some concerns about the physiological relevance of these findings. Also, the molecular details of the model are incomplete.

There are two major concerns:

1. This study is important to understand how the cells divide in tissue. However, it is not clear why the HeLa cells were chosen for the analysis. They are not appropriate to study the cell-cell contact instead

of an epithelial non cancer cell line. Although some of the findings are reproduced in MDCK cells, not all which make it complicated.

2. Molecular details of the findings are insufficient. For example, how changing the adhesion forces affect the cell rounding is not explored. The role of Cadherins in cell-cell adhesion during mitosis through integrin is a pure assumption there is no experiment that tests this, other than cellular localization. Perhaps not N or E cadherin but other cadherin members play a role during mitosis.

This reviewer thinks that orthogonal approaches and methodologies should be taken to support the hypothesis that during mitosis integrins lose actomyosin connection which reinforces cell-cell adhesion.

The following conclusions need experimental confirmation using orthogonal approaches in HeLa and also another epithelial cell line.

“Integrins lose actomyosin connection, which curbs cell-ECM adhesion strengthening, and reinforce cell-cell adhesion sites”

“Interestingly, the binding probability of mitotic STC HeLa cells to interphase HeLa was four-fold higher compared to binding probabilities between interphase cells (Fig. 2e), which indicates that mitotic cells decrease integrin and increase cadherin activity.”

Minor Points:

1. Some important explanations are missing in the figure legends. To make the figures self-explanatory a revision of the legends is necessary. For example, the reference dataset is not clearly explained in the legends.

In Figure 1f and later, VN, Col1, fl-FN vs should be explained in the legend.

What is AS-values? abbreviations should be explained in the figure legend or in the text.

2. There are some mistakes in the text, should be corrected:

For example:

In lines 80-81 “we combined SCFS with fluorescence microscopy to quantified cell cycle-dependent adhesion forces of HeLa cells”.

3. According to Lock&Stromblad 2018, integrin b5 is the functional adhesion during mitosis. However, this study ignores integrin b5. Is there a reason for this? How do you interpret Lock, 2018 results? This paper should be mentioned in the discussion.

4. In Supp Figure 3. Although the integrins expression pattern is clear, VE-Cad and N-cad expression are not clear, there is no clear distinction between negative and positive controls. Perhaps it is due to antibodies but this should be figured out.

5. In line 150 9EG7 Antibody needs a reference.

6. In Figure 5e the specificity of the immunostainings should be shown by knockdowns.

Reviewer #3 (Remarks to the Author):

In this manuscript, the authors compared the initiation and strengthening of cell-matrix and cell-cell adhesion using atomic force microscopy. They reported that whereas the initial formation cell-matrix adhesion is intact in mitotic cells, these adhesions fail to undergo mechanical strengthening. The loss of mechanical strengthening could at least partially be rescued by overexpression of a constitutive Rap1a GTPase, which has been shown to increase cell spreading of mitotic cells. As expected, cell-matrix adhesion initiation in mitotic cells relies upon beta 1 integrin, Talin and Kindlin. Furthermore, the authors reported that mitotic cells form stronger cell-cell adhesion with interphase cells, which is in part contributed by beta 1 integrin, Talin and Kindlin. The manuscript thus provides a detailed description of the change of cell adhesion force in mitotic cells. The experiments are carefully designed with good control experiments and solid statistics. However, the work is overall descriptive with limited mechanistic depth. I would recommend this manuscript with a few revisions that potentially strengthen the mechanistic part. These points are ordered according to their importance.

Major points:

1. It is very intuitive that mitotic cells would reduce their cell-ECM adhesion and it is expected that the remaining adhesions still depend on Talin and Kindlin as in interphase cells. Mitotic cells round up by

channeling RhoA activation mediated by Ect2 to cell cortical actin assembly and stiffening of cell cortex. Such stiff cortical actin network may prevent integrin activation via ERM proteins that engage cortical actin and compete with Talin for integrin binding (DOI: 10.1038/nature14323). This manuscript is a good context that the authors also clarify whether Ect2 knockdown by siRNA (Dix et al., 2018, *Developmental Cell* 45, 132–145, the same paper that the authors refer to for Rap1 study), perturbation of F-actin by low dose Cytochalasin, or RhoA inhibition in mitotic cells influence their cell-ECM adhesion. These data would be more important than the Rap1 study which is largely separated from the rest of the paper.

2. As both Talin and Kindlin are required for dynamic regulation of cell-ECM and cell-cell adhesion in mitosis and the authors have both proteins fluorescently tagged and stably transduced into the corresponding KO cells, it is absolutely necessary to provide time lapse z-stack imaging of spontaneous cell division events in epithelial sheet form by HeLa-Kyoto cells. From this experiment the author should analyze the dynamic re-localization (potentially to cell-cell adhesion) and expression of Talin-Ypet and Cherry-Kindlin. The data could be integrated into Figure 5.

3. The increase cell-cell adhesion in mitotic cells contributed by integrin system is very intriguing. It has been reported that artificially activated beta 1 integrins (by P4G11 or 12G10) concentrate at cell-cell junction (DOI: 10.1006/excr.2000.5099) and potentially reinforce cell-cell adhesion (DOI: 10.1091/mbc.E16-12-0852). The authors should test whether integrin-activating antibodies would accelerate or strengthen mitotic cell-cell adhesion. These data should be accompanied in Figure 5.

4. The investigation of Rap1 GTPase is largely detached from the rest of the study. It is known that Rap1 could promote Talin activation by binding to both its rod domain and head domain. As is discussed by the authors, the action of Rap1 could be mediated by Talin or completely independent of Talin. Combining Rap1 binding deficient Talin mutant (R35E, R118E) with constitutive Rap1 could clarify this mechanistic linkage.

Minor points: the cartoon in Figure 6 is too complicated to efficiently convey the key message. I would suggest using a more abstract and simplified graph.

NCOMMS-21-27544-T

In mitosis integrins reduce adhesion to extracellular matrix and strengthen adhesion to adjacent cells

Point-by-point response to the comments of reviewer #1

Reviewer 1: The study by Huber et al uses single cell atomic force microscopy to measure the attachment forces between cell and ECM, and between cells. They use this methodology to investigate cell cycle dependent changes in attachments. Furthermore, they couple this approach to genetic manipulation of adhesion proteins where they reveal that Beta1 integrins strengthen adhesion between adjacent cells in mitotic cells, while ECM interactions are decreased. This work is of wide interest and addresses how cells regulate cell-cell adhesion while reducing cell-ECM adhesion during mitosis to prevent delamination of the dividing cells.

Authors: We thank the reviewer for the encouraging and constructive feedback to our manuscript. Below, we addressed each point raised by the reviewer.

Furthermore, we agree with the reviewers that the data presented in our manuscript was too densely packed. This dense packing of the original figures made it difficult for the reader to clearly discern all information. However, based on the constructive reviewer's comments, we also had to include new experimental data to revise our manuscript. Thus, to display the information for better clarity we have increased the number of main figures from six to nine, decreased the information per figure, and increased the font size of the figures to match the recommendations given in the journal guide lines.

Reviewer 1: Many of the AFM experiments show adhesion forces which are highly variable, this is especially prominent at longer attachment time points (e.g. Fig 1a).

a) Are there multiple cell populations which lead to a large data variance?

Authors: The reviewer is correct, at contact times > 60 s the adhesion force of single cells separates in two populations (**Fig. R1a**). The number of cells exerting higher adhesion forces increases with contact time (**Fig. R1b**). We hypothesize that the reason for the cells to separate in two populations are adhesion regulatory processes, such as myosin II-mediated actin contractility, which strengthens cell adhesion. Since AFM-based SCFS quantifies the adhesion of single cells at high force and time resolution we observe that some cells increase their adhesion force already after 120 s contact time, while other cells need more than 6 min to do so. We and others have observed this phenomenon for multiple cell lines and ECM substrates, including fibroblasts adhering to fibronectin, CHO cells expressing $\alpha 2\beta 1$ integrins adhering collagen type I, and CHO cells adhering to laminin¹⁻⁴. It is noteworthy that the magnitude and dynamics of this adhesion increase depends on the cell type, the integrin type that mediate the adhesion, and the ECM substrate.

Reviewer 1: b) A larger number of probed cells would reveal if this is the case – I understand this is a technically challenging approach but the data variance suggests more cells should be probed to reveal the “true” spread of the data.

Authors: The reviewer is correct, testing many more cells would display the “true” spread of our data. However, it has to be considered that the SCFS experiments presented in this work took about 5 years. To measure about 100 cells instead of 10 – 15 cells per experimental condition would require ~2.5 month per condition. Thus, to increase the number of cells characterized in each SCFS experiment would considerably lie beyond the time requirements of this revision (> 10 years). However, to gain a better understanding of the data spread, we pooled the adhesion forces of interphase TKO + talin1, KKO + kindlin2 and VKO + vinculin HeLa cells adhering to Matrigel (data taken from **Fig. 1a** and **Supplementary Fig. 7a,b,c**) and conducted additional SCFS experiments (**Fig. R1c**). With that we could increase the number of single cells characterized to $n > 90$ for each contact time. Compared to the SCFS data displayed in **Fig. 1a** of the manuscript (here shown in **Fig. R1a,b**), the extended SCFS data set shows that adhesion forces at contact times > 60 s populate a similar distribution (**Fig. R1c-e**). For the extended SCFS data set, the adhesion force distributions at contact times > 120 s could only be reliably fitted by a double Gaussian distribution. Furthermore, in both data sets similar percentages of cells populate the high adhesion force population. To test whether the distribution of the data sets is statistically different, we applied the robust Mann-Whitney t-test (**Fig. R1c**). Despite the large differences of data points, the adhesion forces at all contact times showed no significantly different distribution between SCFS data sets with low and high numbers of cells. Taken together, this suggests that both data distributions are similar and that the number of HeLa cells tested throughout the manuscript is sufficient to display the spread of the data.

Fig. R1. HeLa cells adhering to Matrigel show two adhesion force populations at contact times ≥ 120 s. **a,c**, Adhesion forces of interphase HeLa cells to Matrigel at given contact times. Dots represent adhesion forces of single cells, red bars median values and n (cells) the number of tested cells per condition. The green circled adhesion forces represent the low adhesion forces population and the magenta circled adhesion forces represent the high adhesion force population. Data for **a** is taken from **Fig. 1a** of the original manuscript and data for **c** is taken from **Fig. 1a, Supplementary Fig. 7a-c** and pooled with additional experiments. **b,d**, stacked box plots show the distribution of cells in the low (green) or high (purple) adhesion force populations at different contact times. **e**, distributions of adhesion forces for all contact times with a double gaussian fit (grey line). Green bars represent adhesion forces in the low force population and purple in the high force population. Given are the means \pm SD for the first gaussian distribution and the second. If the second is instable, it was impossible to fit the data with a double Gaussian distribution.

Reviewer 1: Point 1 could be related to the methodology.

a) If you are taking the same cell attached to the cantilever and then probing for different time points, does the variance in the longer time points represent damaged cells?

Authors: We have carefully chosen the settings of our measurements to prevent major distortion or damage of the cells. For example, we only slightly compress cells by applying a relatively small contact force of ≈ 1 nN. During this compression the contact area of the cell and the support is $\approx 70 \mu\text{m}^2$, which results in a contact pressure of $\approx 15 \text{ N m}^{-2}$ (ref. ⁵). To minimize a memory effect of the cells to previous adhesion force measurements, we randomize the order of contact times and allow the cell to recover for at least the time of contact before repeating the forthcoming adhesion force measurement at a different contact time (**Methods**). Further, to minimize the possibility

of contaminating the substrate we changed the contact area of the cell to the substrate after each adhesion force measurement (**Methods**). Intrigued by the reviewer's question, we performed an additional experiment, in which we repeatedly probed adhesion forces of single HeLa cells to Matrigel at 360 s contact time. Between the measurements we allowed the cells to recover for 360 s and changed the contact area of the substrate. While we see adhesion force fluctuations for single HeLa cells between individual adhesion force measurements (**Fig. R2a**), the adhesion force distribution of all HeLa cells showed no statistically significant differences with increasing number of experimental repeats (**Fig. R2b**). Further, the distribution of adhesion forces of all tested HeLa cells in each of the five repeats is similar to the adhesion force distribution we report in **Fig. 1a** of the manuscript (**Fig. R2b**). Importantly, upon increasing the number of SCFS experiment repeats with a single HeLa cell we do not observe a monotonic decrease of adhesion forces. Hence, we conclude that the cells are not damaged after the detachment of from the substrate, even after longer contact times. We have carefully revised the Methods of our Manuscript to describe the experimental randomization of the contact times (see **revised Methods, section 'Cell-ECM SCFS'**).

Fig. R2. HeLa cells establish similar adhesion force after five approach-retract cycles with 360 s contact time. a-b, Adhesion force of interphase HeLa cells repeatedly tested for 360 s contact time for up to five repeats per single HeLa cell. Cells were allowed to recover from the previous adhesion experiments for 360 s before starting the next adhesion measurement on a different position of the substrate. **a,** Dots present adhesion forces of single HeLa cells. The x-axis indicates the number of repeats. C represents the adhesion force of interphase HeLa cells presented in the original manuscript (**Fig. 1a**). *n*(cells) gives the number of cells measured for each SCFS cycle. *P*-values on bars compare indicated cycle with the control data and were calculated using a two-tailed Mann-Whitney test. No significant differences in adhesion forces were found for repeatedly probing the adhesion force of HeLa cells to Matrigel at 360 s contact time (two-tailed Kruskal-Wallis test). **b,** Adhesion forces of single HeLa cells depending on the number of adhesion measurement cycle.

Reviewer 1: b) Can you provide evidence that the cells are viable following the attachment-detachment cycle?

Authors: For the previous question we showed that repeatedly probing the adhesion force of single HeLa cells to Matrigel did not affect the cell adhesion force. This result indicates that the SCFS experiments do not damage the cells. However, to provide further evidence that HeLa cells are viable after adhesion experiments, we examined whether HeLa cells spread on Matrigel after quantifying their adhesion forces at contact times ranging from 5 – 360 s. The adhesion forces of HeLa cells ($n = 5$) tested in these experiments were similar the adhesion forces of the interphase HeLa cells reported in the manuscript (**Fig. R3a** and **Fig. 1a**). Subsequently to adhesion experiments, we approached single cantilever attached HeLa cells to the Matrigel-coated support until reaching a contact force of 1 nN, maintained the cantilever height constant before retracting the cantilever after 60 min and monitored the cell morphology every 2 min for 62 min by differential interference contrast (DIC) microscopy (**Fig. R3b-g**). During the cantilever retraction the cells detached from the cantilever because they established higher adhesion forces to Matrigel. To have an unperturbed image of each cell in the last frame of the timelapse ($t = 62$ min) we laterally moved the cantilever after retraction. We observed the spreading of four out of five tested HeLa cells on Matrigel within 62 min. In one experiment we observed the spontaneous attachment of a floating interphase HeLa cell to Matrigel at the same time as a cantilever bound cell was approached onto Matrigel (**Fig. R3b** and **R3c**). This spontaneously attached interphase HeLa cell showed similar spreading dynamics as the cantilever bound cell that we used for adhesion quantification. The cell that did not spread underneath the cantilever remained attached to the substrate, indicating higher adhesion forces to the substrate than the cantilever and was motile on the substrate. Morphologically this cell appeared bigger after 62 min contact and the images suggests the formation of a metaphase plate in the center of the cell. Hence, this suggests that the HeLa cell transitioned into mitosis within the 60 min contact time. These experiments strengthen our statement that the cell adhesion measurements do not harm cells and cells are viable.

Fig. R3. HeLa cells establish similar adhesion forces after five approach-retract SCFS cycles with 360 s contact time. **a**, Adhesion forces of interphase HeLa cells to Matrigel as measured by SCFS. Dots present adhesion forces of single HeLa cells and red bars their median. n (cells) gives the number of cells measured. The adhesion forces of interphase HeLa cells to Matrigel as reported in **Fig. 1a** are given as reference in the background in grey. P -values compare displayed and reference data using a two-tailed Mann-Whitney test. **b-g**, Time-lapse images of cells spreading on Matrigel. Images were taken every 2 min for 62 min. **b**, After the SCFS experiments displayed in **a** (blue data points) the interphase HeLa cells were approached to Matrigel until reaching a contact force of 1 nN. **c**, A floating interphase HeLa cell coincidentally attached to Matrigel while approaching the cantilever-bound HeLa cell for **Fig. R3b** to Matrigel. **d-g**, Interphase HeLa cells were approached to Matrigel after SCFS experiments (see **Fig. R3a**; **d**, green; **e**, red; **f**, orange; **g**, black) and allowed to interact with Matrigel for 62 min.

Reviewer 1: c) What happens if you measure a cell through a single attachment-detachment cycle at each time point?

Authors: We compared adhesion forces of HeLa cells to Matrigel at 360 s contact time displayed in the original manuscript (**Fig. 1a**) to adhesion forces of the first cycle of the repetitive adhesion force measurements performed for **Fig. R2**, which represent experiments in which probed cells underwent only a single adhesion force quantification. We did not observe significant differences between the adhesion forces (**Fig. R2b**). Since this is the longest contact time, with the largest data spread we conclude that acquiring data with a single adhesion force measurement per HeLa cell would also not affect the results of other contact times (5 s – 240 s). Hence, we conclude that adhesion forces we quantified would not be affected if we would only one contact time per cell instead of all contact times, or until we monitor morphological

changes such as cell spreading. On the other hand, quantifying the adhesion force at a single contact time per cell would increase the time required for these experiments by at least 6-fold and hence would not be feasible considering the large amount of adhesion force experiments performed in the manuscript.

Reviewer 1: Adhesion force increases over time with the mitotic cells –

a) Can mitotic cells reach the same adhesion forces at longer time points?

Authors: To resolve this interesting question, we quantified the adhesion force of mitotic^{STC} HeLa cells expressing paxillin-GFP to Matrigel after 60 min contact time (**Fig. R4** taken from **Fig. 1c-e**). Additionally, we monitored the paxillin-GFP recruitment by confocal microscopy during the contact time and quantified the spreading area. We found that interphase HeLa cells spread and assemble paxillin-positive adhesion clusters within 60 min contact time. The adhesion forces established by interphase cells to Matrigel exceeded the adhesion to the cantilever thereby making adhesion force quantification impossible due to detachment of the cell from the cantilever during its retraction. In contrast, mitotic^{STC} HeLa cells remained rounded and did not assemble optically resolvable paxillin-containing adhesion sites. Furthermore, mitotic^{STC} HeLa cells contacting Matrigel for 60 min showed similar adhesion forces compared to mitotic^{STC} HeLa cells contacting Matrigel for 360 s (**Fig. R4d** and **Fig. 1e**). Thus, the experiments show that mitotic HeLa cells cannot strengthen adhesion to similar levels of interphase HeLa cells even after very long contact times. We have revised the results section “**Mitotic cells poorly strengthen integrin-mediated cell-ECM adhesion**” for clarity.

Fig. R4 (taken from **Fig. 1c-e**). **Mitotic^{STC} HeLa cells do not spread, form no paxillin-containing adhesion sites, and establish low adhesion forces within 60 min.** **a-b**, Representative time-series of confocal microscopy images of an interphase (**a**) and a mitotic^{STC} (**b**) HeLa (Kyoto) cell, transiently transfected with paxillin-GFP (seven cells per condition), and adhering to Matrigel during SCFS for given contact times. Black arrows show paxillin-GFP clusters. Scale bars, 20 μ m. **c**, Contact time-dependent spreading area (\pm SEM) of interphase and mitotic^{STC} HeLa cells transiently transfected with paxillin-GFP (seven cells per condition). The spreading area of each cell is normalized to the contact area at 0 min. The gray area indicates statistically significant differences in the spreading area between interphase and mitotic^{STC} HeLa cells. **d**, Adhesion force of mitotic^{STC} HeLa cells to Matrigel after 60 min. Adhesion forces after 360 s to Matrigel are given in grey.

Reviewer 1: b) In relation to point 2, are mitotic cells less viable/more damaged by the AFM approach therefore fail to attach with the same force at longer time points?

Authors: We have carefully chosen our experimental settings to not harm or damage the cells during our SCFS-based cell adhesion measurements. The experimental settings are very unlikely to harm mitotic HeLa cells for the following reasons. Firstly, while STC arrests cells in prometaphase, it does not harm or damage cells as STC is a commonly used agent to synchronize cells. Hence, the washout of STC results in dividing and viable cells⁶. We also show that STC-arrested and spontaneous mitotic HeLa cells establish the same adhesion forces. Additionally, we show that STC does not affect the adhesion of interphase HeLa cells. Together, this demonstrates that STC does not hamper cell adhesion independent of the cell state. Secondly, we have shown previously that a small compression ($< 2 \mu\text{m}$) of a HeLa cell as applied in our SCFS experiments by bringing the cell and substrate into contact, does not impede the progression of HeLa cells through mitosis⁷. Hence, we conclude that mitotic HeLa cells are viable and undamaged after the compression during the contact time. Thirdly, we have shown that even after recording considerably higher adhesion forces of interphase HeLa cells by SCFS these interphase cells were still viable and spread on Matrigel (**Fig. R2**), from which we conclude that the mechanically more stable mitotic cells⁶ will not be affected by the detachment process. These three points strongly suggest that mitotic cells remain intact and viable during the SCFS experiments.

However, to further demonstrate that mitotic cells are viable and dividing after our SCFS experiments, we quantified the adhesion force of spontaneous mitotic HeLa cells to Matrigel and subsequently monitored whether the cells progressed through mitosis (**Fig. R5**). Every HeLa cell ($n=3$) went through mitosis within 30 min after the SCFS experiments. With this, we are confident that the mitotic HeLa cells, which adhesion forces have been characterized by SCFS, are not less viable or damaged.

Fig. R5. Adhesion force quantification by SCFS does not reduce viability of mitotic HeLa cells. Using SCFS the adhesion forces of cantilever-attached, spontaneous mitotic HeLa cells were quantified at contact times ranging from 5 s to 360 s. Subsequently their progression through mitosis was monitored by time-lapse DIC microscopy. Yellow arrows show single mitotic, or two daughter cells attached to the cantilever. Scale bar, $15 \mu\text{m}$.

Reviewer 1: Fig5: Could the authors comment on whether measuring cell-cell interactions in the vertical axis would differ from the forces in the lateral plane?

Authors: We are not sure whether we understand the reviewer correctly. We assume that the reviewer means that the 'interactions in the vertical' are the interactions at the interface between two adjacent epithelial cells. AFM-based SCFS is limited to measuring forces in the vertical axis (*i.e.* vertical to a substrate or cell membrane) and hence does not allow to address how cell-cell adhesion forces would differ in the lateral plane (*i.e.* parallel to a substrate or cell membrane). However, we are interested how cells initiate and establish adhesion and thus use unpolarized cells. Further, for SCFS we choose cells that either are isolated or connected to only one other cell. Using this widely used approach we make sure that the cell on the substrate has sufficient free cadherins to establish cell-cell adhesion. Hence, we assume that the basic principles of the adhesion formation do not depend on whether we probe it laterally or vertically.

The question of the reviewer could also address how lateral contractile forces induced by cadherin signaling and mediated by actomyosin contractility correlate with the vertical adhesion forces we measure by SCFS. Yet, such a fundamental question in cell-cell and cell-ECM adhesion has not been addressed, neither it can be addressed by the SCFS setup we are using. We are also not aware of any other setup that can simultaneously measure vertical adhesion and lateral contraction forces of living cells to substrates. Hence, to answer this question would require major technological development.

Reviewer 1: What is the basis for using a contact force of 1 nN?

Authors: We have carefully adjusted the contact force to minimize the compression of cell. The relatively small contact force of 1 nN (see also our response further above) ensures that we neither damage the cell nor significantly prestress the actomyosin cortex. At the same time, the contact force is sufficient to keep the cell in contact with the substrate over the entire contact period. Commonly contact forces between 0.75 nN and 2 nN are used to characterize the adhesion forces of mammalian cells by SCFS^{4,5,8-22}. Along the same lines, we have shown in earlier publications that a compression force < 5 nN does not affect the progression of HeLa cells through mitosis and, hence, does not harm mitotic cells⁷.

Reviewer 1: Could the authors explain why time points were chosen between 5 and 360 s for measuring attachment?

Authors: The massive remodeling of adhesion before and during mitotic cell rounding is accompanied by a major adhesion regulatory mechanism that collectively disassembles and re-establishes cell-ECM and cell-cell adhesion. While the disassembly of adhesion can be monitored by light microscopy, we are interested in the quantitative understanding of how mitotic and interphase cells initiate and

strengthen adhesion forces to the ECM and neighboring cells. Hence, the chosen contact times (from milliseconds to minutes) allow us to address how the binding probabilities (**Fig. 2c,e and 8c**) and the adhesion initiation and strengthening (**Fig. 1**) progresses within the first 6 min and how they are differentially regulated in mitotic and interphase cell adhesion. Thereby, we observe binding probability changes of interphase and mitotic cells as well as how integrins and cadherins cooperate/synergize to strengthen the adhesion. We are unable to extend the contact time in our adhesion force experiments much further as interphase cells establish too high forces to Matrigel or to other cells so that the cantilever-bound cells detach from the cantilever during measuring the adhesion forces (e.g., retraction of the cantilever). In our revised manuscript we now better describe the rationale of our experiments and why we have chosen the time points (see revised Manuscript, **Introduction**).

Reviewer 1: Could the authors comment on how this work may connect to the recent publications Gough et al JBC 2021 297 100837 which explores an interaction between Talin and CDK1?

Authors: The reviewer refers to a publication describing a CDK1-dependent phosphorylation of talin at the R7 (S1589), which reduces the mechanical stability of the R7R8 region in the talin rod. The work referred to was published after we submitted our manuscript. In the publication the authors describe that reduction of the interaction between CDK1 and talin by the deletion of R8 in talin results in CDK1-inhibition independent and larger/more integrin adhesion complexes (IACs) in non-synchronized cells. In their experiments using synchronized cells, the authors show that while wild type cells in which talin and CDK1 can interact the overall adhesion area of IACs are reduced in G2, cells in which a talin mutation blocks this interaction do not decrease their adhesion area in the G2. This finding indicates that the activity of CDK1 is essential for the disassembly of adhesion sites before mitosis. In line, in an earlier publication it was shown that CDK1-cyclin A2 promotes IAC formation, while the CDK1-cyclin B1 complex is required to disassemble IACs and cell rounding²³. Together both publications indicate that the complex partner of CDK1 is essential for the cell cycle dependent adhesion regulation. Further, Jones *et al.* describe that an inhibitory phosphorylation of CDK1 by Wee1 triggers the remodeling of IAC before mitosis²³. This indicates a potential reduction of talin phosphorylation at R7 as described by Gough *et al.*²⁴ However, since Gough *et al.* describe the interaction between talin and CDK1 during interphase, the exact molecular mechanism how CDK1 regulates cell adhesion at the transition from interphase to mitosis has not been explored. Hence, any connection between the phosphorylation of talin by CDK1 and our findings without a thorough investigation of the effect of talin and/or CKD1 mutations described by Gough *et al.* would be speculative. Although this is a highly interesting topic, it would be beyond the scope of our manuscript to explore this effect. It would require many more experiments with the different cell lines.

Minor Points

Reviewer 1: It would be useful to add some numbers to the main text of the manuscript

so that the reader can have an idea of the forces involved.

Authors: The reported adhesion forces depend on the experimental details, such as cell type, contact force and retraction speeds. We thus believe that giving explicit adhesion forces would increase the complexity for the reader. Hence, we reported trends in the main manuscript that are more universal for different experimental settings.

Reviewer 1: Add molecular weight markers and full blots for the western-blot in the supplementary material.

Authors: We have added molecular weight markers and full western blots to the supplementary material (revised **Supplementary Fig. 5**).

Reviewer 1: Many of the figure labels/font size is too small so they should be increased for the reader.

Authors: We have split the main figures to increase the font size of the labels and to display the details at better clarity. Now we use font sizes which have been described to be optimal by the journal.

NCOMMS-21-27544-T

In mitosis integrins reduce adhesion to extracellular matrix and strengthen adhesion to adjacent cells

Point-by-point response to the comments of reviewer #2

Reviewer 2: This manuscript carefully analyzes how the adhesion forces change during cell division and cell rounding. Huber et al., made extensive use of atomic force microscopy-based single-cell force spectroscopy (SCFS) to measure cell-ECM and cell-cell adhesion forces and strengths in interphase and mitosis. Authors conclude that at the onset of mitosis, integrins diminish cell-ECM adhesion and enhance cell-cell adhesion sites. By taking CRISPR/Cas9 based gene editing, they showed that kindlin, talin and vinculin play a role in strengthening cell-cell adhesion during mitosis. The findings of this manuscript are important and will serve to the understanding of cell rounding and how the adhesion components regulate the cell morphology in a cell cycle-dependent manner. The experimental design is very nice to examine the adhesion forces in cell-ECM and cell-cell. However, this reviewer has some concerns about the physiological relevance of these findings. Also, the molecular details of the model are incomplete.

Authors: Thank you for your encouraging feedback and for the constructive criticism and comments. To address the reviewers concerns we have included a new cell line (MCF7) to validate our findings for HeLa and MDCK cells and engineered fibroblasts to validate molecular details of the cell-ECM adhesion regulation in interphase and mitosis. In a nutshell, the numerous additional experiments verify our findings and deepen the mechanistic insights into how cells regulate adhesion in interphase and mitosis. These new findings have been included into the revised manuscript. The major findings added the manuscript include:

- We experimentally verify the main findings of the manuscript with MCF7 cells that form E-cadherin mediated cell-cell adhesions.
- We experimentally verify the importance of talin and kindlin for the transient cell-ECM adhesion. The results show that integrins are not connected to the actomyosin cortex in mitotic fibroblasts.
- We experimentally show the differential contribution of cadherins to cell-cell adhesion in interphase and mitosis.
- We show that compared to interphase cells, mitotic HeLa cells strongly reduce adhesion force to vitronectin at contact times ranging from 5 – 360 s. This data suggests that $\alpha V\beta 5$ integrin mediated reticular adhesion attaching the mitotic cell body to vitronectin are formed during cell rounding.
- We show experimentally that destabilizing the actomyosin cortex does not increase the adhesion force of mitotic cells to the ECM.
- We also show experimentally that the artificial activation of integrins by activating antibodies or Mn^{2+} does not increase the adhesion force of mitotic HeLa cells to other interphase or mitotic HeLa cells.

Furthermore, we kindly agree with the reviewer's that the data presented in our manuscript was too densely packed. This dense packing of the original figures made

it difficult for the reader to clearly discern all information. However, based on the constructive reviewer's comments, we also had to include new experimental data to revise our manuscript. Thus, to display the information at better clarity we have increased the number of main figures from six to nine, decreased the information per figure, and increased the font size of the figures to match the recommendations given in the journal guide lines.

Reviewer 2: This study is important to understand how the cells divide in tissue. However, it is not clear why the HeLa cells were chosen for the analysis. They are not appropriate to study the cell-cell contact instead of an epithelial non cancer cell line. Although some of the findings are reproduced in MDCK cells, not all which make it complicated.

Authors: We agree with the reviewer that the insights offered in our manuscript provide important information how cells regulate adhesion to divide in tissue. We understand that from our initial summarizing figure (former **Fig. 6a**) the reader could interpret that our findings are exclusively related to epithelial cells growing in monolayers. However, we did not aim to limit our study to epithelial cells and hence used a variety of cell types, including HeLa cells, MDCK cells and fibroblasts. In this regard, we have revised the summary figure (new **Fig. 9**) and removed the depiction of cells in a monolayer. Driven by the concerns of the reviewer we also included the characterization of an additional MCF7 cell line to the manuscript. While MCF7 cells are also cancerous, they form E-cadherin mediated cell-cell adhesions and show the characteristics of differentiated mammary epithelium. However, it would be extremely time consuming to engineer a talin-depleted (TKO), a TKO re-expressing the talin head domain (THD), a kindlin-depleted (KKO), a vinculin-depleted (VKO) and a RAP1-CA MCF7 cell line and to subsequently characterize these engineered cell lines cell biologically, their adhesion forces of by SCFS for five conditions (interphase and mitotic adhesion to Matrigel; interphase/interphase, mitotic/ interphase, and mitotic/mitotic cell-cell adhesion), their binding probability and their integrin/cadherin levels by flow cytometry. We estimate this additional work to take at least 3 years. We hence consider such considerable amount of additional work to be out of the scope of the revision of the current manuscript. However, to address the reviewer comment at least partly, we reduced the number of experiments and confirmed our major findings of HeLa cells, MDCK cells and fibroblasts also with MCF7 cells. First, we quantified the adhesion force of interphase or mitotic^{STC} MCF7 cells to Matrigel for contact times ranging from 5 s to 360 s. Like the other cell lines, mitotic^{STC} MCF7 cells establish drastically lower cell-ECM adhesion force at contact times ≥ 60 s to Matrigel. We also performed interphase/interphase and mitotic^{STC}/interphase cell-cell adhesion force measurements with MCF7 cells. Compared to interphase MCF7 cells, mitotic^{STC} MCF7 cells strengthen adhesion more rapidly to interphase MCF7 cells, thereby validating the reported results of HeLa and MDCK cells. To test whether, as in HeLa and MDCK cells, $\beta 1$ integrins participate in cell-cell adhesion, we blocked their contribution by incubating mitotic^{STC} MCF7 cells with the $\beta 1$ integrin-blocking antibody AIIB2 prior to the attachment of the cantilever. Indeed, blocking $\beta 1$ integrins on mitotic^{STC} MCF7 cells reduced their adhesion force established to interphase cells at contact times ≥ 60 s. We have included this new experimental data revealed on MCF7 cells into the revised

version of the manuscript to further strengthen our claims that mitotic cells decrease adhesion to the extracellular matrix but increase adhesion to neighboring cells. Please see the revised results sections “**Mitotic cells poorly strengthen integrin-mediated cell-ECM adhesion**”, “**Mitotic cells increase adhesion strengthening to adjacent cells**” and “**Mitotic cells employ integrins to strengthen cell-cell adhesion**” as well as the revised **Supplementary Figs. 2h, 4e and 10d**.

Fig. R6. Mitotic^{STC} MCF7 cells reduce adhesion strengthening to Matrigel but increase adhesion strengthening to mitotic or interphase cells. **a**, Adhesion forces of interphase (left) or mitotic^{STC} (right) MCF7 cells to Matrigel after given contact times. Dots represent adhesion forces of single cells, red bars median values and $n(\text{cells})$ the number of tested cells per condition. AS-values give the adhesion strengthening rate as the slope (\pm SE) of a linear fit through adhesion forces for all contact times with the P -value comparing the AS-value to that of a reference data set. Adhesion forces of interphase MCF7 cells to Matrigel are given as reference in grey for comparison with mitotic cells. P -values compare adhesion forces of interphase and mitotic^{STC} MCF7 cells. **b**, Adhesion forces between two interphase MCF7 cells (left) or an interphase and a mitotic^{STC} MCF7 cell (right) after given contact times. **c**, Cell-cell adhesion forces between AIB2-treated interphase and untreated interphase MCF7 cells (left) or an AIB2-treated mitotic^{STC} and an untreated interphase MCF7 cell (right). **b,c**, Data representation as in **a**. As reference adhesion forces between two interphase MCF7 cells (**b**) or the adhesion forces of unperturbed MCF7 cells in the respective conditions (**c**) are given. P -values compare adhesion forces of given data with reference data. ‘Mitotic^{STC}’ indicates that mitotic cells were enriched by 2 μ M STC for 12 h prior to and incubated with STC throughout the experiments. P -values were calculated using two-tailed Mann-Whitney tests and P -values comparing AS-values were calculated by a two-tailed extra sum of squares F-test.

To confirm that mitotic cells initiate transient kindlin- and talin-dependent adhesion to the ECM, which is not reinforced by actomyosin engagement, we conducted SCFS experiments with interphase and mitotic^{STC} TKO, KKO and TKO+THD fibroblasts to the fibronectin fragment FNIII7-10. As reported previously^{25,26}, TKO or KKO fibroblasts in the interphase majorly reduce adhesion forces to FNIII7-10 for all contact times.

Similarly, mitotic^{STC} TKO and KKO fibroblasts establish lower adhesion forces to FNIII7-10 compared to mitotic^{STC} fibroblasts expressing kindlin or talin at all contact times. Importantly, mitotic^{STC} and interphase TKO or KKO fibroblasts establish similar adhesion forces to FNIII7-10 indicating that adhesion regulation during mitosis depends on both adaptor proteins.

The expression of THD in TKO fibroblasts fully recovered the adhesion defects of TKO fibroblasts to FNIII7-10 for contact times ≤ 120 s but was insufficient to restore the adhesion force at longer contact times to the same level as observed for fibroblasts expressing full-length talin. Mitotic^{STC} TKO+THD fibroblasts established lower adhesion forces to FNIII7-10 at contact times ≥ 60 s compared to interphase TKO+THD fibroblasts, thereby confirming the results of HeLa cells (**Fig. R7**). Importantly, the adhesion forces of mitotic^{STC} TKO+THD and wild-type fibroblasts were indistinguishable. These results verify that mitotic cells employ integrins to initiate adhesion, which depends on the presence of kindlin and talin. However, mitotic cells fail to couple newly bound integrins to actin by the talin rod domain. Hence, the initiated cell adhesion is not reinforced by the contractile actomyosin. We included this data into the revised version of our manuscript to strengthen our claim that integrins that bind to ECM ligand are not coupled to actin *via* talin, which results in a transient ligand binding. Please see the revised results section “**Mitotic cells do not connect newly ligated integrins to the actomyosin cortex**”, the revised **Discussion** and revised **Supplementary Fig. 8**.

Fig. R7. Kindlin and talin are essential for mitotic cell-ECM adhesion in fibroblasts. Cell-ECM adhesion forces of interphase (**left**) or mitotic^{STC} (**right**) TKO (**a**), KKO (**b**) or TKO+THD (**c**) fibroblasts to FNIII7-10 after given contact times. Adhesion forces of wild type interphase or mitotic^{STC} fibroblasts to FNIII7-10 are given as reference in grey (**Supplementary Fig. 2g**). Dots represent adhesion forces of single cells, red bars their median values and n (cells) the number of tested cells per condition. AS-values give the adhesion strengthening rate as the slope (\pm SE) of a linear fit through adhesion forces for all contact times. If only one row P-values are given they compare given and reference data. Otherwise, top row P-values compare

mitotic^{STC} adhesion forces with respective interphase adhesion forces and bottom row *P*-values compare given and reference adhesion forces. 'Mitotic^{STC}' indicates that mitotic cells were enriched by 2 μ M STC for 12 h prior to and incubated with STC throughout the experiments. *P*-values were calculated using two-tailed Mann-Whitney tests and *P*-values comparing *AS*-values were calculated by a two-tailed extra sum of squares F-test.

Reviewer 2: Molecular details of the findings are insufficient. [...]

Authors: The reviewer comments that the manuscript provides limited molecular details. We respectfully point out that we show novel molecular details of mitotic cell adhesion regulation, which include:

- 1) We show that like interphase cells mitotic cells initiate cell-ECM adhesion *via* integrins, which depends on talin and kindlin.
- 2) We show that less integrins reside in an extended conformation in mitotic cells compared to interphase cells.
- 3) We show that newly ligand-bound integrins in mitotic cells are not connected to the cytoskeleton *via* the talin rod domain or vinculin. This lacking connection does not allow the force-induced stabilization of the integrin ligand bound state.
- 4) We show that the missing actomyosin connection of integrins curbs adhesion strengthening, integrin clustering and prevents cell spreading.
- 5) We show that mitotic cells increase cell-cell adhesion forces and strengthening to other interphase or mitotic cells.
- 6) Our results show that adhesome components contributing to cell-ECM adhesion in interphase, participate in cell-cell adhesion during mitosis.

Moreover, by addressing the constructive feedback from all reviewers, we have included additional mechanistic insight in our revised manuscript.

Reviewer 2: [...] For example, how changing the adhesion forces affect the cell rounding is not explored. [...]

Authors: The reviewer is interested in how changing the adhesion forces affects cell rounding. In our experiments we cannot address how adhesion forces affect the rounding of mitotic cells, as our SCFS approach to quantify cell adhesion forces requires cantilever-attached cells that fully detach from the substrate during the retraction of the cantilever. However, the cantilever attachment of adherent cells transitioning into mitosis is insufficient for adhesion force quantification due to their higher adhesion forces to the substrate. Nonetheless, there is clear evidence in literature that the disassembly of focal adhesions and hence the reduction of adhesion force is critically important for cell rounding and successful mitosis. Explicitly, it was shown that cells unable to disassemble focal adhesions due artificial integrin activation by the expression of dominant active Rap1 or the presence of Mn^{2+} show rounding defects in mitosis and are often multinucleated. Our results provide mechanistic insight that in mitotic cells integrins only transiently bind a ligand which results in curbed adhesion strengthening and maturation and ultimately in loss of cell spreading.

Reviewer 2: [...] The role of Cadherins in cell-cell adhesion during mitosis through integrin is a pure assumption there is no experiment that tests this, other than cellular localization. Perhaps not N or E cadherin but other cadherin members play a role during mitosis.

Authors: We agree with the reviewer that the interaction between integrins and cadherins is an assumption and we do not intend to claim a direct interaction between integrins and cadherins in our manuscript. Hence, in the discussion of our manuscript we clearly state that whether the “contribution of $\beta 1$ integrins is accomplished through direct binding to E- and/or N-cadherin, as reported for the collagen-binding $\alpha 2\beta 1$ or $\alpha E\beta 7$ integrins, remains to be explored.” To identify specific cadherins and integrins that facilitate this potential interaction would require to engineer many more HeLa cell lines and to conduct many more SCFS experiments, which is beyond the scope of the manuscript. However, to further validate whether cadherins are involved in cell-cell adhesion during mitosis, we incubated interphase and mitotic^{STC} HeLa cells with EGTA to chelate Ca^{2+} -ions (Fig. R7). While the Ca^{2+} -chelation curbs the homophilic interaction of cadherins²⁷, it maintains the integrin function²⁸. Interphase/interphase cell-cell adhesion forces of HeLa cells in the presence of EGTA are neglectable for all contact times, thus indicating that cadherins are the main facilitator of interphase cell-cell adhesion. However, while the chelation of Ca^{2+} -ions also reduces the adhesion forces of mitotic^{STC} HeLa cells to interphase or to mitotic^{STC} HeLa cells, their adhesion forces are significantly higher than adhesion forces between two EGTA-treated interphase HeLa cells. Together with the $\beta 1$ integrin-blocking experiments (Fig. 8b), the results support a role of $\beta 1$ integrins for mitotic cell-cell adhesion. However, whether $\beta 1$ integrins interact with cadherins in a Ca^{2+} -independent manner cannot be concluded from our experiments. We have included this new data to the revised manuscript to deepen the mechanistic insights into how mitotic cells employ integrins to strengthen cell-cell adhesion to neighboring cells. Please see the revised results sections “Mitotic cells do not connect newly ligated integrins to the actomyosin cortex”, the revised Discussion and revised Fig. 8a.

Fig. R8. Cadherins differentially contribute to cell-cell adhesion in interphase and mitosis. Adhesion forces between two interphase (left), an interphase and a mitotic^{STC} cell (middle), or two mitotic^{STC} (right) HeLa cells after given contact times in the presence of the Ca^{2+} -chelator EGTA. Dots represent adhesion forces of single cells, red bars median values and n (cells) the number of tested cells per condition. AS-values give the adhesion strengthening rate as the slope (\pm SE) of a linear fit through adhesion forces for all contact times with the P -value comparing the AS-value to that of a reference data set. Adhesion forces of

untreated HeLa cells in the respective conditions are given in grey. *P*-values compare adhesion forces of given data with reference data. 'Mitotic^{STC}' indicates that mitotic cells were enriched by 2 μ M STC for 12 h prior to and incubated with STC throughout the experiments. *P*-values were calculated using two-tailed Mann-Whitney tests and *P*-values comparing *AS*-values were calculated by a two-tailed extra sum of squares F-test.

Reviewer 2: This reviewer thinks that orthogonal approaches and methodologies should be taken to support the hypothesis that during mitosis integrins lose actomyosin connection which reinforces cell-cell adhesion.

The following conclusions need experimental confirmation using orthogonal approaches in HeLa and also another epithelial cell line.

"Integrins lose actomyosin connection, which curbs cell-ECM adhesion strengthening, and reinforce cell-cell adhesion sites"

Authors: We agree that this sentence is misleading and changed it to "...*newly ligand-bound integrins are not coupled to actin and hence poorly strengthen adhesion to ECM, β 1 integrins reinforce the adhesion of mitotic cells to neighboring cells*". Using SCFS and genetically engineered cell lines we have shown that mitotic cells do not engage newly bound integrins to actin (**Fig. R7a**) and that in MCF7 cells that integrins participate in cell-cell adhesion strengthening (**Fig. R6c**). Further, MDCK cells showed a clear β 1 integrin localization to the lateral cell sides (**Fig. 8**). We have included these new experiments to the revised manuscript. Please see the revised results sections "***Mitotic cells do not connect newly ligated integrins to the actomyosin cortex***" and "***Mitotic cells employ integrins to strengthen cell-cell adhesion***", the revised **Discussion** and revised **Supplementary Figs. 8a and 10d**.

Reviewer 2: ... "Interestingly, the binding probability of mitotic STC HeLa cells to interphase HeLa was four-fold higher compared to binding probabilities between interphase cells (Fig. 2e), which indicates that mitotic cells decrease integrin and increase cadherin activity."

Authors: Our results show that less β 1 integrins are in an extended conformation and that the binding probability of mitotic^{STC} HeLa cells to Matrigel is lower compared to interphase cells (**Fig. 2c**). On the other hand, the binding probability of mitotic^{STC} HeLa cells to other HeLa cells is drastically increased (**Fig. 2e**). To our best knowledge there are no specific antibodies against cadherins that report their activity and hence a validation is not feasible by flow cytometry. However, we agree that the last part of the cited sentence makes a too strong conclusion that we did not intend to draw. We thus have revised the sentence in the results section "***Integrin-ECM binding diminishes, and cell-cell binding increases in mitosis***" to "...*indicating that mitotic cells slightly decrease adhesion initiation to the ECM but strongly increase adhesion initiation to other cells*".

Minor Points

Reviewer 2: Some important explanations are missing in the figure legends. To make the figures self-explanatory a revision of the legends is necessary. For example, the reference dataset is not clearly explained in the legends.

In Figure 1f and later, VN, Col1, fl-FN vs should be explained in the legend.

What is AS-values? abbreviations should be explained in the figure legend or in the text.

Authors: We thank the reviewer for pointing out this critical issue. We have now paid particular attention to make the figures and their legends self-explanatory. We specify the reference data set of every figure panel in the legend and the purified ECM proteins are not abbreviated. Furthermore, the AS-values are now explained in every figure legend by *“AS-values give the adhesion strengthening rate as the slope (\pm SE) of a linear fit through adhesion forces for all contact times with the P-value comparing the AS-value to that of the reference data set”*.

Reviewer 2: There are some mistakes in the text, should be corrected, for example: In lines 80-81 “we combined SCFS with fluorescence microscopy to quantified cell cycle-dependent adhesion forces of HeLa cells”.

Authors: We carefully revised the manuscript to, hopefully, eliminate all mistakes in the text.

Reviewer 2: According to Lock&Stromblad 2018, integrin b5 is the functional adhesion during mitosis. However, this study ignores integrin b5. Is there a reason for this? How do you interpret Lock, 2018 results? This paper should be mentioned in the discussion.

Authors: The publication mentioned by the reviewer describes a distinct class of adhesion sites, termed reticular adhesion, formed by α V β 5 integrins, facilitating mitotic cell adhesion to vitronectin. To test whether these reticular adhesions assemble during mitosis we included vitronectin as substrate in **Fig. 1f**. We find that the adhesion of mitotic^{STC} HeLa cells to vitronectin is significantly lower at 120 s contact time compared to interphase HeLa cells. We conducted additional SCFS experiments with contact times ranging from 5 s to 360 s, which confirmed the findings in **Fig. 1f (Fig. R9)**. We included the publication in the discussion of the original manuscript. However, we agree with the reviewer that we could discuss our results in more detail in light of Lock&Stromblad *et al.* 2018. Hence, we added the following sentence to the discussion of the manuscript: *“Additionally, from our experiments we conclude that reticular adhesions that anchor the mitotic cell body to vitronectin are established during cell rounding rather than being newly established during mitosis.”*

Fig. R9. Mitotic^{STC} HeLa (kyoto) cells reduce adhesion strengthening to vitronectin. Adhesion forces of interphase (a) or mitotic^{STC} (b) HeLa (Kyoto) cells to VN or BSA after given contact times. Dots represent adhesion forces of single cells, red bars median values and $n(\text{cells})$ the number of tested cells per condition. AS-values give the adhesion strengthening rate as the slope (\pm SE) of a linear fit through adhesion forces for all contact times with the P -value comparing the AS-value to that of a reference data set. Adhesion forces of interphase HeLa cells to Matrigel are given as reference in grey for comparison with mitotic cells. P -values comparing adhesion forces of interphase and mitotic^{STC} HeLa cells. 'Mitotic^{STC}' indicates that mitotic cells were enriched by 2 μM STC for 12 h prior to and incubated with STC throughout the experiments. P -values were calculated using two-tailed Mann-Whitney tests and P -values comparing AS-values were calculated by a two-tailed extra sum of squares F-test.

Reviewer 2: In Supp Figure 3. Although the integrins expression pattern is clear, VE-Cad and N-cad expression are not clear, there is no clear distinction between negative and positive controls. Perhaps it is due to antibodies but this should be figured out.

Authors: Although we agree with the reviewer that the difference between positive and negative control is not very big, the logarithmic difference is significant (**Fig. R10**).

Fig. R10. Cadherin expression in control HeLa cells. Flow cytometry analysis of HeLa cells for expression levels N- and VE-cadherin. Normalized histograms of fluorescence intensities for interphase or mitotic^{STC} HeLa cells stained with antibodies against the given integrin subunit or cadherin are shown. Negative controls were unstained cells. 50'000 cells were analyzed for each condition.

Reviewer 2: In line 150 9EG7 Antibody needs a reference.

Authors: We have added a reference for the 9EG7 antibody.

Reviewer 2: In Figure 5e the specificity of the immunostainings should be shown by knockdowns.

Authors: We acknowledge that the reviewer would like to see the specificity of the antibodies. However, the confocal microscopy image in **Fig. 8d** shows basal and lateral, but not apical localization of the anti- β 1 integrin antibody. To us this shows that the antibody does not show only unspecific binding. Further, AIB2 has been used as immunostaining AB in many other publications.

NCOMMS-21-27544-T

In mitosis integrins reduce adhesion to extracellular matrix and strengthen adhesion to adjacent cells

Point-by-point response to the comments of reviewer #3

Reviewer 3: In this manuscript, the authors compared the initiation and strengthening of cell-matrix and cell-cell adhesion using atomic force microscopy. They reported that whereas the initial formation cell-matrix adhesion is intact in mitotic cells, these adhesions fail to undergo mechanical strengthening. The loss of mechanical strengthening could at least partially be rescued by overexpression of a constitutive Rap1a GTPase, which has been shown to increase cell spreading of mitotic cells. As expected, cell-matrix adhesion initiation in mitotic cells relies upon beta 1 integrin, Talin and Kindlin. Furthermore, the authors reported that mitotic cells form stronger cell-cell adhesion with interphase cells, which is in part contributed by beta 1 integrin, Talin and Kindlin. The manuscript thus provides a detailed description of the change of cell adhesion force in mitotic cells. The experiments are carefully designed with good control experiments and solid statistics. However, the work is overall descriptive with limited mechanistic depth. I would recommend this manuscript with a few revisions that potentially strengthen the mechanistic part. These points are ordered according to their importance.

Authors: We thank the reviewer for the constructive and positive feedback. We have addressed several of the proposed experiments to strengthen the mechanistic insights of the manuscript. The details of these experiments are given further below.

Furthermore, we agree with the reviewers that the data presented in our manuscript was too densely packed. This dense packing of the original figures made it difficult for the reader to clearly discern all information. However, based on the constructive reviewer's comments, we also had to include new experimental data to revise our manuscript. Thus, to display the information at better clarity we have increased the number of main figures from six to nine, decreased the information per figure, and increased the font size of the figures to match the recommendations given in the journal guide lines.

Reviewer 3: It is very intuitive that mitotic cells would reduce their cell-ECM adhesion and it is expected that the remaining adhesions still depend on Talin and Kindlin as in interphase cells. Mitotic cells round up by channeling RhoA activation mediated by Ect2 to cell cortical actin assembly and stiffening of cell cortex. Such stiff cortical actin network may prevent integrin activation via ERM proteins that engage cortical actin and compete with Talin for integrin binding (DOI: 10.1038/nature14323). This manuscript is a good context that the authors also clarify whether Ect2 knockdown by siRNA (Dix et al., 2018, Developmental Cell 45, 132–145, the same paper that the authors refer to for Rap1 study), perturbation of F-actin by low dose Cytochalasin, or RhoA inhibition in mitotic cells influence their cell-ECM adhesion. These data would be more important than the Rap1 study which is largely separated from the rest of the paper.

Authors: Thank you for suggesting to change the properties of the cortical actin by cytochalasin D or RhoA inhibition and to investigate how this change effects the adhesion strengthening of mitotic^{STC} HeLa cells. We conducted SCFS experiments with mitotic^{STC} HeLa cells in the presence of increasing cytochalasin D concentrations (0.1 μ M, 0.5 μ M and 1 μ M). Thereto, detached mitotic^{STC} HeLa cells were incubated with the respective concentration of cytochalasin D 30 min prior and throughout the SCFS experiments (**Fig. R10a**). Inhibition of actin polymerization during mitosis at a very low cytochalasin D concentration (0.1 μ M) did not affect the cell adhesion forces to Matrigel for all contact times. However, with increasing concentration mitotic^{STC} HeLa cells establish lower adhesion forces. While mitotic^{STC} HeLa cells incubated with 0.5 μ M cytochalasin D only reduced their adhesion force at longer contact times (\geq 240 s), incubation with 1 μ M cytochalasin D reduced their adhesion force for all contact times, except for 60 s. Consequently, cytochalasin D also reduced the adhesion strengthening of mitotic^{STC} HeLa cells at concentrations of 0.5 μ M and 1 μ M. The results indicate that an intact actomyosin cortex is important for the adhesion initiation process and that mitotic cells having a destabilized actomyosin cortex cannot strengthen their adhesion.

Further, we have tested the effect of RhoA inhibition on the adhesion strengthening of interphase and mitotic^{STC} HeLa cells to Matrigel (**Fig. R10b,c**). We incubated interphase or mitotic^{STC} HeLa cells with 0.5 μ g ml⁻¹ RhoA inhibitor I for 4 h before detaching HeLa cells from the culture plate and throughout SCFS. While RhoA inhibition reduced the adhesion force of interphase HeLa cells at contact times \geq 120 s and thereby reduced their adhesion strengthening, it did not affect the adhesion of mitotic^{STC} HeLa cells within the first 360 s contact time.

Taken together, we agree with the hypothesis of the reviewer that talin may not be able to engage to the stiff cortical actin. However, the results do not indicate that the mild inhibition of actin polymerization or the inhibition of RhoA during mitosis increase the adhesion strengthening of mitotic cells. We have included the additional results to the new **Fig. 6c,d**, described them in the results section "***Mitotic cells do not connect integrins to the actomyosin cortex***", and discussed them in the revised **Discussion**.

Fig. R11. Actin perturbation but not RhoA inhibition reduces adhesion of mitotic HeLa cells. **a**, Adhesion forces of mitotic^{STC} HeLa cells in to Matrigel in the presence of 0.1 μM (left), 0.5 μM (middle) or 1.0 μM (right) cytochalasin D after given contact times. **b,c**, adhesion forces of interphase (**b**) or mitotic^{STC} HeLa cells in the presence of Rho inhibitor I after given contact times. **a-c**, Dots represent adhesion forces of single cells, red bars median values and *n* (cells) the number of tested cells per condition. AS-values give the adhesion strengthening rate as the slope (\pm SE) of a linear fit through adhesion forces for all contact times with the *P*-value comparing the AS-value to that of a reference data set. Adhesion forces of unperturbed HeLa cells to Matrigel are given as reference in grey for comparison with mitotic cells. *P*-values compare displayed and reference adhesion forces. *P*-values were calculated using two-tailed Mann-Whitney tests and *P*-values comparing AS-values were calculated by a two-tailed extra sum of squares F-test.

Reviewer 3: As both Talin and Kindlin are required for dynamic regulation of cell-ECM and cell-cell adhesion in mitosis and the authors have both proteins fluorescently tagged and stably transduced into the corresponding KO cells, it is absolutely necessary to provide time lapse z-stack imaging of spontaneous cell division events in epithelial sheet form by HeLa-Kyoto cells. From this experiment the author should analyze the dynamic re-localization (potentially to cell-cell adhesion) and expression of Talin-Ypet and Cherry-Kindlin. The data could be integrated into Figure 5.

Authors: We agree with the reviewer that time lapse, z-stack microscopy of kindlin and talin would be fantastic to characterize the transition of kindlin and talin from the cell-ECM interphase to cell-cell adhesion sites. However, the high cytosolic fluorescent signal of mCherry-Kindlin and Talin-YPet make it impossible to localize kindlin and talin

at the cell-cell region in interphase (**Fig. R12**).

Fig. R12. Talin1-YPet localization in a mitotic HeLa cell. Representative confocal microscopy image of a fluorescent mitotic TKO + talin1-YPet HeLa cell attached to a neighboring interphase TKO + talin1-YPet HeLa cell. Along the red line, the intensity profile shows high cytosolic localization of talin1-YPet in both the interphase and mitotic cell. Scale bar, 10 μm

Reviewer 3: The increase cell-cell adhesion in mitotic cells contributed by integrin system is very intriguing. It has been reported that artificially activated beta 1 integrins (by P4G11 or 12G10) concentrate at cell-cell junction (DOI: 10.1006/excr.2000.5099) and potentially reinforce cell-cell adhesion (DOI: 10.1091/mbc.E16-12-0852). The authors should test whether integrin-activating antibodies would accelerate or strengthen mitotic cell-cell adhesion. These data should be accompanied in Figure 5.

Authors: We agree that it is interesting to address whether artificially activated integrin accelerate or reinforce cell-cell adhesion in distinct cell cycle phases. Hence, we used 12G10 or Mn^{2+} -incubation to artificially induce integrin activation and quantified cell-cell adhesion forces (**Fig. R13**). To our surprise cell-cell adhesion forces were not affected by the integrin activation. This observation leads to two possible hypotheses: 1) the participation of integrins in cell-cell adhesion does not depend on the conformation of integrins or 2) integrin-activation is not the limiting factor during integrin-mediated cell-cell adhesion strengthening. We described these experiments in the revised manuscript to describe more in detail the nature of how integrins participate in mitotic cell-cell adhesion (see revised **Results** section "**Mitotic cells employ integrins to strengthen cell-cell adhesion**", **Supplementary Fig. 11a,b** and **Discussion**). However, to identify which of the two hypothesis is true would require a large set of additional cadherin- and integrin-deficient HeLa cells and a large set of additional SCFS experiments. Although we are currently pursuing some of these experiments, we believe that the manuscript is already very rich on experimental data and findings. Hence, we would believe that integrating another set of rather complex data would go beyond the scope of the manuscript.

Fig. R13. Artificial integrin activation does not affect mitotic cell-cell adhesion. a,b Adhesion forces between two interphase (left), an interphase and a mitotic^{STC} cell (middle), or two mitotic^{STC} (right) HeLa cells after given contact times in the presence of the β 1-integrin activating antibody 12G10 (a) or Mn^{2+} (b). Dots represent adhesion forces of single cells, red bars median values and $n(\text{cells})$ the number of tested cells per condition. AS-values give the adhesion strengthening rate as the slope (\pm SE) of a linear fit through adhesion forces for all contact times with the P -value comparing the AS-value to that of a reference data set. Adhesion forces of untreated HeLa cells in the respective conditions are given in grey. P -values compare adhesion forces of given data with reference data. ‘Mitotic^{STC}’ indicates that mitotic cells were enriched by 2 μ M STC for 12 h prior to and incubated with STC throughout the experiments. P -values were calculated using two-tailed Mann-Whitney tests and P -values comparing AS-values were calculated by a two-tailed extra sum of squares F-test.

Reviewer 3: The investigation of Rap1 GTPase is largely detached from the rest of the study. It is known that Rap1 could promote Talin activation by binding to both its rod domain and head domain. As is discussed by the authors, the action of Rap1 could be mediated by Talin or completely independent of Talin. Combining Rap1 binding deficient Talin mutant (R35E, R118E) with constitutive Rap1 could clarify this mechanistic linkage.

Authors: We did not intent a mechanistical link between Rap1 activation and mitotic cell adhesion. Other publications showed that artificially activated integrins by Mn^{2+} or by a dominant active version of Rap1 prevents complete mitotic rounding, which results in multinucleated cells. While we were surprised to find that Mn^{2+} activates integrins on mitotic^{STC} HeLa cells but that this activation does not increase the cell adhesion force, we used Rap1-DA as a tool to intracellularly activate integrins and

address whether this would affect cell adhesion strengthening in mitotic cells. Indeed, we find that expressing Rap1-DA mitotic^{STC} HeLa cells increases their adhesion strengthening, however, not to levels of interphase cells. Our intention was not to claim mechanistic insight into how Rap1 potentially activates integrins. Hence, we believe that the experiments suggested by the reviewer are surely very interesting and insightful in the molecular details how Rap1-DA activates mitotic adhesion, but not within the scope of the manuscript. Further, we believe that the workload for these experiments is too high. However, we revised the text to “*Although our experiments do not provide mechanistic insights into how Rap1-CA promotes integrin mediated ECM binding of mitotic cells, the experiments show that non-regulatable Rap1A majorly affects the adhesion of mitotic cells.*” (See revised manuscript, **Discussion**).

Minor Points

Reviewer 3: The cartoon in Figure 6 is too complicated to efficiently convey the key message. I would suggest using a more abstract and simplified graph.

Authors: We agree with the reviewer and simplified the figure (see new **Fig. 9**). We hope that the new figure is better to understand and conveys the most important points of the manuscript.

References

1. Dao, L., Gonnermann, C. & Franz, C. M. Investigating differential cell-matrix adhesion by directly comparative single-cell force spectroscopy. *J. Mol. Recognit.* 26, 578–589 (2013).
2. Taubenberger, A., Cisneros, D. A., Puech, P.-H., Müller, D. J. & Franz, C. M. Revealing early steps of $\alpha 2\beta 1$ integrin-mediated adhesion to collagen type I by using single-cell force spectroscopy. *Mol. Biol. Cell* 18, 1634–1644 (2007).
3. Dao, L. *et al.* Revealing non-genetic adhesive variations in clonal populations by comparative single-cell force spectroscopy. *Exp. Cell Res.* 318, 2155–2167 (2012).
4. Benito-Jardón, M. *et al.* αv -Class integrin binding to fibronectin is solely mediated by RGD and unaffected by an RGE mutation. *J. Cell Biol.* 219, 507 (2020).
5. Schubert, R. *et al.* Assay for characterizing the recovery of vertebrate cells for adhesion measurements by single-cell force spectroscopy. *FEBS Lett.* 588, 3639–3648 (2014).
6. Ramanathan, S. P. *et al.* Cdk1-dependent mitotic enrichment of cortical myosin II promotes cell rounding against confinement. *Nat. Cell Biol.* 17, 148–159 (2015).
7. Cattin, C. J. *et al.* Mechanical control of mitotic progression in single animal cells. *Proc. Natl. Acad. Sci. U.S.A.* 112, 11258–11263 (2015).
8. Hosseini, B. H. *et al.* Immune synapse formation determines interaction forces between T cells and antigen-presenting cells measured by atomic force microscopy. *Proc National Acad Sci* 106, 17852–17857 (2009).
9. Taubenberger, A. V., Quent, V. M., Thibaudeau, L., Clements, J. A. & Hutmacher, D. W. Delineating breast cancer cell interactions with engineered bone microenvironments. *J Bone Miner Res* 28, 1399–1411 (2013).
10. Weder, G. *et al.* Measuring cell adhesion forces during the cell cycle by force spectroscopy. *Biointerphases* 4, 27–34 (2009).
11. Weder, G. *et al.* The quantification of single cell adhesion on functionalized surfaces for cell sheet engineering. *Biomaterials* 31, 6436–6443 (2010).
12. Krieg, M. *et al.* Tensile forces govern germ-layer organization in zebrafish. *Nat. Cell Biol.* 10, 429–436 (2008).
13. Riet, J. T. *et al.* Dynamic coupling of ALCAM to the actin cortex strengthens cell adhesion to CD6. *J. Cell Sci.* 127, 1595–1606 (2014).
14. Strohmeyer, N., Bharadwaj, M., Costell, M., Fässler, R. & Müller, D. J. Fibronectin-bound $\alpha 5\beta 1$ integrins sense load and signal to reinforce adhesion in less than a second. *Nat. Mater.* 16, 1262–1270 (2017).
15. Viljoen, A. *et al.* Force spectroscopy of single cells using atomic force microscopy. *Nat Rev Methods Primers* 1, 1–24 (2021).

16. Yu, M., Strohmeyer, N., Wang, J., Müller, D. J. & Helenius, J. Increasing throughput of AFM-based single cell adhesion measurements through multisubstrate surfaces. *Beilstein J. Nanotechnol.* 6, 157–166 (2015).
17. Spoerri, P. M., Strohmeyer, N., Sun, Z., Fässler, R. & Müller, D. J. Protease-activated receptor signalling initiates $\alpha 5\beta 1$ -integrin-mediated adhesion in non-haematopoietic cells. *Nat. Mater.* 19, 218–226 (2020).
18. Bharadwaj, M. *et al.* αV -class integrins exert dual roles on $\alpha 5\beta 1$ integrins to strengthen adhesion to fibronectin. *Nat. Commun.* 8, 14348 (2017).
19. Friedrichs, J. *et al.* Contributions of galectin-3 and -9 to epithelial cell adhesion analyzed by single cell force spectroscopy. *J. Biol. Chem.* 282, 29375–29383 (2007).
20. Manninen, A., Müller, D. J. & Helenius, J. Galectin-3 regulates integrin $\alpha 2\beta 1$ -mediated adhesion to collagen-I and -IV. *J. Biol. Chem.* 283, 32264–32272 (2008).
21. Yu, M., Wang, J., Müller, D. J. & Helenius, J. In PC3 prostate cancer cells ephrin receptors crosstalk to $\beta 1$ -integrins to strengthen adhesion to collagen type I. *Scientific Reports* 5, 8206 (2015).
22. Benoit, M., Gabriel, D., Gerisch, G. & Gaub, H. E. Discrete interactions in cell adhesion measured by single-molecule force spectroscopy. *Nat. Cell Biol.* 2, 313–317 (2000).
23. Jones, M. C., Askari, J. A., Humphries, J. D. & Humphries, M. J. Cell adhesion is regulated by CDK1 during the cell cycle. *J. Cell Biol.* 217, 3203–3218 (2018).
24. Gough, R. E. *et al.* Talin mechanosensitivity is modulated by a direct interaction with cyclin-dependent kinase-1. *J Biol Chem* 297, 100837 (2021).
25. Böttcher, R. T. *et al.* Kindlin-2 recruits paxillin and Arp2/3 to promote membrane protrusions during initial cell spreading. *J. Cell Biol.* 216, 3785–3798 (2017).
26. Theodosiou, M. *et al.* Kindlin-2 cooperates with talin to activate integrins and induces cell spreading by directly binding paxillin. *Elife* 5, e10130 (2016).
27. Smith, A. L., Dohn, M. R., Brown, M. V. & Reynolds, A. B. Association of Rho-associated protein kinase 1 with E-cadherin complexes is mediated by p120-catenin. *Mol Biol Cell* 23, 99–110 (2012).
28. Leavesley, D. I., Schwartz, M. A., Rosenfeld, M. & Cheresch, D. A. Integrin beta 1- and beta 3-mediated endothelial cell migration is triggered through distinct signaling mechanisms. *J Cell Biology* 121, 163–170 (1993).

REVIEWER COMMENTS

Reviewer #1 (Remarks to the Author):

I thank the authors for the detailed response to my comments and the additional work they have put in to the manuscript. I have no concerns and believe this is an excellent and interesting article for the field.

Reviewer #2 (Remarks to the Author):

I thank the authors for addressing all the comments raised previously. If possible, one additional comment is about the model figure (Figure 9), it could be revised but not very crucial.

It looks complicated (especially c and d could be simplified) and typos in the figure should be corrected. RhoA part of the model is not clear. Perhaps actin and RhoA should also be shown in mitotic cells but not connected.

Reviewer #4 (Remarks to the Author):

The manuscript and Huber uses an impressive AFM-based method to follow adhesions forces developed between cells and complexes extracellular matrix or other cells. Numerous additional experiments were added and a clear effort was performed to reformat figures in order to simplify their reading. Indeed, this manuscript is highly dense and make it difficult to follow. Indeed, having all the details of the forces produced at each time point (5s, 20s, 60s... 360s) largely increase the amount of data while the AS value is the most efficient to give a synthetic idea of the consequence of the tested conditions. Moreover, the Fig.3 could be removed to supplementary figure since the confirmation that kindlin-talin are important for all steps of adhesion development and vinculin is important for adhesion reinforcement are two facts highly known in the field (and not different from mitotic cells).

My main concern is the interpretations about the difference between cell-cell contact in mitotic and inter phasic conditions. As matrigel, a reductionist approach will be extremely valuable to prove these assumptions. Purified extracellular domain of cadherin spread on PDMS should be used to quantify the level of forces developed by cadherin alone. Indeed, the cells used to mimic a cell-cell contact are spread for 20h on their surface and could produce large amount of ECM at their surrounding. This could explain why blocking antibody affect the test used to mimic cell-cell contacts at short term (only probes during 1 min while cell-cell contacts needs hours to be stabilize). There is not differential analysis of the effects of EGTA (assume to block only cadherin) vs the blocking antibody strategy (that could have also a certain level of non-specificity) that could help to understand the relative importance of both type of adhesions measured by AFM.

This is why time lapse z-stack of talin or kindlin activities were absolutely necessary. It will have been perhaps interesting to get FRET imaging of alpha-catenin vs talin or kindlin to overcome the expression issues and check their physical proximity (supported by their need to be associated with actin during cell-cell contact) in the specific context interin-cadherin synergy during cell-cell contact development in mitotic cells.

The part on the manuscript on Rap1 seems indeed an extra message since not fully developed. Indeed, previous work have show the importance of Rap1 to regulate endocytosis or promote the relocalization of NM-IIB in cell-cell contact (Balzac et al., 2005; Gomez et al., 2015). The ratio between NM-IIA and NM-IIB in inter phasic and mitotic cells could have been an interesting hypothesis to follow the data on Rap1 activation.

Other points

Fig.4a: I am quite surprised that vinculin KO has no effect on cell-cell contact since shown previously during reinforcement of forces developed in cell-cell contacts (Seddiki et al., 2014)

Fig.6C: I am quite confused to see that there is no significative difference of forces at 0,1mM of CytoD and a significative difference for the two higher concentrations since the distributions of the points are highly homologuous. Do the authors think that the level of resolution of this technic is around a AS values of 2pN/s?

Fig2b-Fig.5a: I am quite confuse that the mitotic cells does not present the same response to 9EG7 staining between both figures. Indeed, without Mn²⁺ we have an increase of 9EG7 on mitotic cells vs interphase (Fig.5b) and a decrease of 9EG7 staining between interphase and mitotic in Fig.2a. There is something not clear for me in the legend.

Fig.8: a nice ratiometric analysis between Cadherin and integrin staining should be performed in order to determine if the ratio changes between mitotic and non mitotic cells. This is really not clear on the images presented.

NCOMMS-21-27544A

In mitosis integrins reduce adhesion to extracellular matrix and strengthen adhesion to adjacent cells

Point-by-point response to the comments of reviewer #1

Reviewer 1: I thank the authors for the detailed response to my comments and the addition work they have put in to the manuscript. I have no concerns and believe this is an excellent and interesting article for the field.

Authors: Thank you for your critical and constructive comments, which helped to improve our manuscript. We are happy that we could address all reviewer comments satisfactorily.

NCOMMS-21-27544A

In mitosis integrins reduce adhesion to extracellular matrix and strengthen adhesion to adjacent cells

Point-by-point response to the comments of reviewer #2

Reviewer 2: I thank the authors for addressing all the comments raised previously. If possible, one additional comment is about the model figure (Figure 9), it could be revised but not very crucial.

It looks complicated (especially c and d could be simplified) and typos in the figure should be corrected. RhoA part of the model is not clear. Perhaps actin and RhoA should also be shown in mitotic cells but not connected.

Authors: Thank you for your comments. We have removed the typos and revised Fig. 9 as suggested.

NCOMMS-21-27544A

In mitosis integrins reduce adhesion to extracellular matrix and strengthen adhesion to adjacent cells

Point-by-point response to the comments of reviewer #4

Reviewer 4: The manuscript and Huber uses an impressive AFM-based method to follow adhesions forces developed between cells and complexes extracellular matrix or other cells. Numerous additional experiments were added and a clear effort was performed to reformat figures in order to simplify their reading. Indeed, this manuscript is highly dense and make it difficult to follow. Indeed, having all the details of the forces produced at each time point (5s, 20s, 60s... 360s) largely increase the amount of data while the AS value is the most efficient to give a synthetic idea of the consequence of the tested conditions.

Authors: We thank the reviewer for the feedback. We agree that the adhesion strengthening (AS)-value is well suited to quantify the adhesion strengthening process within the first 360 s of cell-ECM and cell-cell adhesion. However, to additionally provide the adhesion forces of individual cells it is essential to display the relative cell adhesion strengths over different contact times, which can be quite different from AS-values. To illustrate this need, we have generated three sets of *in silico* cell adhesion forces that have a very similar AS-value (**Fig. R1**). However, the data sets populate the adhesion forces quite differently. The blue cell population distributes homogeneously. The purple cell population shows two subpopulations, one of which showing significantly lower cell adhesion forces compared to the blue population, while the other subpopulation shows similar or higher cell adhesion forces. The orange cell population distributes similarly homogeneous as the blue population but shows significantly higher adhesion forces at all contact times. The differences in adhesion forces between the blue and the orange cell population are similar for all contact times. This exemplifies why we consider displaying adhesion forces as essential.

Fig. R1. Illustration of the necessity to display adhesion forces and AS-values. We generated three data sets of cell adhesion forces shown in blue, purple and orange. Left, dots represent single cell adhesion forces ($n=15$ per dataset) and the bar their median. P-values compare adhesion forces of the blue data set with the respective other data set and are calculated by the two-tailed Mann-Whitney test. Right, dots represent mean cell adhesion forces, error bars the SD, and the lines linear fits through adhesion forces at all contact times. AS-values represent the slopes of the linear fits. P-values, which compare whether the slopes of the fits are different, are calculated by two-tailed extra sum of squares F-tests.

the lines linear fits through adhesion forces at all contact times. AS-values represent the slopes of the linear fits. P-values, which compare whether the slopes of the fits are different, are calculated by two-tailed extra sum of squares F-tests.

Reviewer 4: Moreover, the Fig.3 could be removed to supplementary figure since the confirmation that kindlin-talin are important for all steps of adhesion development and vinculin is important for adhesion reinforcement are two facts highly known in the field (and not different from mitotic cells).

Authors: We agree with the reviewer that the function of the three adaptor proteins kindlin, talin and vinculin are well documented for the adhesion of interphase cells. However, we consider Fig. 3 to be important for the manuscript for the following two reasons: first, to our knowledge we demonstrate for the first time the importance and the quantitative contribution of talin and kindlin to establish cell adhesion to Matrigel, which is mostly composed of collagens and laminins. Second, the data presented in Fig. 3 show the differential contribution of all three adaptor proteins to the adhesion force (or strength) of mitotic cells. So far it has not been shown how mitotic cells initiate adhesion to the ECM. Particularly, we show for the first time that integrins of mitotic cells can only transiently bind ECM ligands in a kindlin and talin head domain dependent manner. Different to interphase cells, mitotic cells cannot strengthen adhesion to the ECM since integrins cannot engage to the actomyosin cortex. Hence, we think that Fig. 3 provides essential information to understand how cell-ECM adhesion is regulated in mitotic cells. We thus think it is important to keep this figure in the main manuscript.

Reviewer 4: My main concern is the interpretations about the difference between cell-cell contact in mitotic and inter phasic conditions. As matrigel, a reductionist approach will be extremely valuable to prove these assumptions. Purified extracellular domain of cadherin spread on PDMS should be used to quantify the level of forces developed by cadherin alone. Indeed, the cells used to mimic a cell-cell contact are spread for 20h on their surface and could produce large amount of ECM at their surrounding. This could explain why blocking antibody affect the test used to mimic cell-cell contacts at short term (only probes during 1 min while cell-cell contacts needs hours to be stabilize).

Authors: In our experiments we aimed to mimic physiological conditions as close as possible. Hence, we used Matrigel, which represents a complex mixture of ECM proteins, to investigate cell-ECM adhesion and adhesion between interphase cells and mitotic cells. The concern of the reviewer is that mitotic cells may be covered with ECM proteins, which leads to the participation of integrins in cell-cell adhesion. However, the reviewer assumes that the cells used in our cell-cell adhesion experiments are spread on the surface for 20 h. This is for most experiments not the case. For interphase/interphase and mitotic/interphase cell-cell adhesion experiments, we seeded freshly suspended interphase cells onto Matrigel-coated PDMS supports and allowed them to spread for 30 min up to 3 h. We have clarified the corresponding methods section mentioning that the cells spread maximally for 3 h on Matrigel (see revised Methods, section "Cell-cell SCFS"). Since the cells spread on Matrigel are prepared the same way, we can exclude that the cell preparation causes the different adhesion forces between mitotic cells and interphase cells vs. interphase cells and interphase cells. Further, by using β 1-blocking antibodies (A1B2) we show that integrins on the cantilever-bound mitotic cell contribute to the adhesion between mitotic cell and interphase cell (Fig. 8). Hence, we can exclude that the contribution of integrins arise from a potential ECM coverage of the cantilever-bound mitotic cell.

In mitotic cell / mitotic cell adhesion experiments, the cells are spread on their substrate (Matrigel) and incubated with STC for at least 12 h. However, it was recently shown that active $\beta 1$ integrins distribute non-homogenously in mitotic cells and localize mainly at the basal side^{1,2}. These recent studies could not detect active $\beta 1$ integrins on the apical side of mitotic cells. However, these would be required to immobilize ECM proteins at the apical side of mitotic cells, which we characterize in our cell-cell adhesion force experiments. Additionally, we show that integrins only transiently bind ECM proteins and hence cannot keep ECM ligands bound to the cell surface. Hence, we conclude that an ECM coverage of mitotic cells is very unlikely.

Altogether, we think that a reductionistic approach with cadherins adsorbed to PDMS is not required to prove the contributions of integrins in mediating mitotic cell-cell adhesion. We also conclude that ECM proteins are not immobilized on the cell surface in our experiments and hence do not contribute to the increased adhesion forces established by mitotic cells to other interphase or mitotic cells. Using a reductionist approach would not be beneficial to understand how mitotic cells regulate adhesion. In fact, using cadherins adsorbed to a support to investigate cell cycle dependent cell-cell adhesion regulation would reduce the complexity and possibly also the physiological relevance of our experiments. This is because cadherins extracted from the cell and adsorbed onto a surface are not regulated such as they are in their natural cellular environment. For all these reasons we decided not to conduct the reductionist cell adhesion experiments with truncated cadherins being spread on PDMS.

Reviewer 4: There is not differential analysis of the effects of EGTA (assume to block only cadherin) vs the blocking antibody strategy (that could have also a certain level of non-specificity) that could help to understand the relative importance of both type of adhesions measured by AFM.

Authors: Our experiments differentiate the effects of EGTA and the $\beta 1$ -blocking antibodies (AIB2) on cell adhesion mediated via integrins and cadherins. While the addition of EGTA to the medium in cell-cell adhesion experiments did not affect the integrin-mediated cell spreading on Matrigel-coated supports, it completely abolished the adhesion between two interphase HeLa cells (see Fig. 8). These experimental findings are in line with the literature showing that EGTA inhibits the binding of cadherins but not of integrins³⁻⁵. On the other hand, we performed cell adhesion experiments of $\beta 1$ -blocking antibody incubated HeLa cells to Matrigel. The antibodies completely abolished the cell adhesion, showing their high potency of blocking $\beta 1$ integrin. Further, incubation of interphase HeLa cells with AIB2 did not affect their adhesion forces to other interphase cells, showing that the AIB2 does not inhibit cadherin-mediated adhesion. Hence, we can exclude an effect of $\beta 1$ -blocking AIB2 on cadherins.

Reviewer 4: This is why time lapse z-stack of talin or kindlin activities were absolutely necessary. It will have been perhaps interesting to get FRET imaging of alpha-catenin vs talin or kindlin to overcome the expression issues and check their physical proximity (supported by their need to be associated with actin during cell-cell contact) in the specific context interin-cadherin synergy during cell-cell contact development in mitotic cells.

Authors: We agree with the reviewer that a z-stack time lapse would have been an excellent experiment to visualize how cells regulate their cell-ECM and cell-cell

adhesion while transitioning into mitosis. However, we faced the problem of too high cytosolic fluorescent signal, which made it impossible to localize adhesome proteins during mitosis. To overcome this limitation, the reviewer comments that it would have been perhaps interesting to perform FRET imaging. We agree with the reviewer that such experiments could be very helpful. Setting up such a completely new set of experiments would first require optimizing and verifying a large amount of genetically expressed FRET-pairs, which we currently do not have. Since the manuscript is already very dense and includes a considerable variety of different experimental methods and data sets (such as mentioned by the reviewer), we are afraid that successfully setting up and conducting such technically challenging experiments are out of the scope of our current manuscript.

Reviewer 4: The part on the manuscript on Rap1 seems indeed an extra message since not fully developed. Indeed, previous work have show the importance of Rap1 to regulate endocytosis or promote the relocalization of NM-IIB in cell-cell contact (Balzac et al., 2005; Gomez et al., 2015). The ratio between NM-IIA and NM-IIB in inter phasic and mitotic cells could have been an interesting hypothesis to follow the data on Rap1 activation.

Authors: The intention of the Rap1 experiments in our manuscript was to show that mitotic cell-ECM adhesion is, at least in part, rescuable by the overexpression of a constitutive active form of Rap1. While it is surely interesting to look at changes in the ratios between NM-IIA and NM-IIB in interphase and mitotic cells, this would deviate from the context of the manuscript. We are concerned that further enhancing the complexity and opening a new context of our paper potentially confuses the reader. We thus rather consider to follow this very interesting suggestion in a separate work.

Other points

Reviewer 4: Fig.4a: I am quite surprised that vinculin KO has no effect on cell-cell contact since shown previously during reinforcement of forces developed in cell-cell contacts (Seddiki et al., 2014)

Authors: We understand that the reviewer is confused that we do not observe an impact of vinculin in cell-cell adhesion forces between two interphase cells. Many publications describe the reinforcement of cell-cell adhesion by vinculin⁶⁻¹¹ We do not claim that vinculin is not involved in cell-cell adhesion. However, in our experiments we characterize how two interphase cells initiate and strengthen adhesion when brought into contact. Hence, our results show that in the adhesion formation between two interphase cells does not involve vinculin within the first 360 s of cell-cell contact. We further show that mitotic cells employ vinculin earlier in adhesion strengthening compared to interphase cells. To avoid confusion of the reader, we have revised our manuscript to more clearly describe the relevance and context of this finding (see revised **Discussion**).

Reviewer 4: Fig.6C: I am quite confused to see that there is no significative difference of forces at 0,1mM of CytoD and a significative difference for the two higher concentrations since the distributions of the points are highly homologuous. Do the authors think that the level of resolution of this technic is around a AS values of 2pN/s?

Authors: In Fig. 6c we show the effect of actin depolymerization on adhesion forces of mitotic^{STC} HeLa cells to Matrigel in the presence of different CytoD concentrations.

We use the two-tailed Mann-Whitney test to analyze differences in cell adhesion forces of treated and untreated HeLa cells (**Fig. R2a**). To determine AS-values, we use the slope of a linear fit through all adhesion forces recorded over all contact times in the respective condition (**Fig. R2b**). The 95% confidence interval of each fit is determined by the spread of the measured adhesion forces. This analysis allows us to provide the standard error of the slope, which provides the accuracy of the AS-value. We use a two-tailed extra sum of squares F-test to analyze whether two slopes (AS-values) are statistically different. Our statistical analysis using the cell adhesion forces shows that the lowest concentration (0.1 μM CytoD) does not affect the adhesion forces of mitotic^{STC} HeLa cells to Matrigel nor does it affect the AS-value. With increasing concentrations of CytoD an effect on the adhesion forces of mitotic^{STC} HeLa cells is observable. At 0.5 μM CytoD, we observe a reduction of adhesion forces at longer contact times (>120 s) and hence a reduced AS-value. At the highest concentration of 1 μM CytoD we observe reduced adhesion forces for all contact times and hence also reduced AS-values.

Fig. R2. Actin perturbation by cytochalasin D reduces the adhesion of mitotic HeLa cells to Matrigel in dose dependent manner. **a**, Adhesion forces of mitotic^{STC} HeLa cells to Matrigel in the presence of 0.1 μM (left), 0.5 μM (middle) or 1.0 μM (right) cytochalasin D (cytoD) after given contact times. Dots represent adhesion forces of single cells and red bars median values. Adhesion forces of untreated mitotic^{STC} HeLa cells are given as reference in grey. **b**, AS-values are the slopes of a linear fit through all adhesion forces for all contact times. Lines depict the linear fit with their 95% confidence intervals represented in shaded color. Dots represent the mean cell adhesion forces at the given contact time and the bars indicate the SD. The *P*-values compare the AS-value to that of untreated mitotic^{STC} HeLa cells. *P*-values were calculated using two-tailed Mann-Whitney tests and *P*-values comparing AS-values were calculated by a two-tailed extra sum of squares F-test.

Reviewer 4: Fig2b-Fig.5a: I am quite confuse that the mitotic cells does not present the same response to 9EG7 staining between both figures. Indeed, without Mn2+ we have an increase of 9EG7 on mitotic cells vs interphase (Fig.5b) and a decrease of 9EG7 staining between interphase and mitotic in Fig.2a. There is something not clear for me in the legend.

Authors: We thank the reviewer for spotting this mistake. While rearranging figures for clarity we mistakenly changed **Fig. 5a**. We have reverted the figure to the first submitted version of the manuscript (see revised **Fig. 5a**). Both sub-figures (here shown as **Fig. R3**) now show that less $\beta 1$ integrins on mitotic cells adopt a high affinity conformation, as shown by reduced levels of 9EG7 binding.

Fig. R3. Fewer $\beta 1$ integrins adopt a high affinity conformation in mitosis. Flow cytometry of interphase and mitotic^{STC} HeLa cells labeled for $\beta 1$ integrins in an extended conformation (clone 9EG7) as displayed in Fig. 2b (left) and revised Fig. 5a (right). Dots represent the median fluorescent intensities of 20'000 cells analyzed per sample normalized to the mean of median fluorescent intensity of interphase HeLa cell samples, bars the mean of all medians and error bars show the SEM. n(samples) indicate the number of biological

independent samples tested. P-values compare indicated conditions and were calculated using two-tailed Mann-Whitney tests.

Reviewer 4: Fig.8: a nice ratiometric analysis between Cadherin and integrin staining should be performed in order to determine if the ratio changes between mitotic and non mitotic cells. This is really not clear on the images presented.

Authors: Thank you for suggesting a ratiometric analysis of the fluorescent image showing the localization of E-cadherin GFP and AIB2-labelled $\beta 1$ integrins. We have conducted the analysis and identified the membrane signals for the ratio analysis by a colocalization analysis of the AIB2-labelled $\beta 1$ integrins and E-cadherin GFP channels (**Fig. R4a,b**, now included into revised **Fig. 8d-g**). The AIB2-labelled $\beta 1$ integrins and E-cadherin GFP signals are exclusively found in the cell membranes. We obtained the thresholds for segmentation automatically using the method by Costes et al.¹². We extracted the intensities from the AIB2-labelled $\beta 1$ integrins and E-cadherin GFP channels at the positions of significant colocalization and creating a new channel with the ratios AIB2-labelled $\beta 1$ integrins / E-cadherin GFP. We applied a small, 3x3x3 median filter to the ratio channel to suppress local extreme ratios due to noisy pixels. For visualization, all ratios across the analyzed images were pooled and binned using the 25th (ratio = 2.0), 50th (ratio = 4.0) and 75th (ratio = 8.0) percentile and color-coded accordingly. Non-colocalizing pixels were assigned the value of 0. Out-of-focus planes were omitted from the calculation. We find a large variation of ratios between $\beta 1$ integrins and E-cadherin GFP across different membrane regions of cells and between different cells independent of whether the connected cells are two interphase or an interphase and a mitotic cell (**Fig. R4c**). However, the ratiometric maps do not indicate a clear change in ratios between $\beta 1$ integrins and E-cadherin GFP in mitotic and non-mitotic cells.

Fig. R4, taken from revised Fig. 8. $\beta 1$ integrins and E-cadherins co-localize at cell-cell adhesions. a,b, Representative immunofluorescence image of MDCK cells seeded on a

Matrigel-coated substrate ($n=14$ images). Cells expressing E-Cadherin-GFP (green; **b**, left) were stained for $\beta 1$ integrin (magenta; **b**, right) and DNA (blue) using $\beta 1$ integrin-blocking antibodies (AIIB2) and SPY650-DNA, respectively (Methods). The dashed line in **a** indicates the section used for intensity plot (**a**, bottom) and orthogonal views (**b**, bottom). For line profiles, intensities were normalized to the highest intensity for the respective channel. **c**, The ratios of intensity signals of $\beta 1$ integrin and E-cadherin GFP are displayed in a false color image. Scale bar, 10 μm .

References

1. Petridou, N. I. & Skourides, P. A. A ligand-independent integrin $\beta 1$ mechanosensory complex guides spindle orientation. *Nat Commun* 7, 10899 (2016).
2. Anastasiou, O., Hadjisavva, R. & Skourides, P. A. Mitotic cell responses to substrate topological cues are independent of the molecular nature of adhesion. *Sci Signal* 13, (2020).
3. Smith, A. L., Dohn, M. R., Brown, M. V. & Reynolds, A. B. Association of Rho-associated protein kinase 1 with E-cadherin complexes is mediated by p120-catenin. *Mol Biol Cell* 23, 99–110 (2012).
4. Leavesley, D. I., Schwartz, M. A., Rosenfeld, M. & Cheresh, D. A. Integrin beta 1- and beta 3-mediated endothelial cell migration is triggered through distinct signaling mechanisms. *J Cell Biology* 121, 163–170 (1993).
5. Zuhorn, I. S., Kalicharan, D., Robillard, G. T. & Hoekstra, D. Adhesion receptors mediate efficient non-viral gene delivery. *Mol Ther* 15, 946–953 (2007).
6. Bays, J. L. & DeMali, K. A. Vinculin in cell–cell and cell–matrix adhesions. *Cell Mol Life Sci* 74, 2999–3009 (2017).
7. Duc, Q. le *et al.* Vinculin potentiates E-cadherin mechanosensing and is recruited to actin-anchored sites within adherens junctions in a myosin II–dependent manner. *J Cell Biol* 189, 1107–1115 (2010).
8. Huveneers, S. *et al.* Vinculin associates with endothelial VE-cadherin junctions to control force-dependent remodeling. *J Cell Biol* 196, 641–652 (2012).
9. Monster, J. L. *et al.* An asymmetric junctional mechanoresponse coordinates mitotic rounding with epithelial integrity. *J Cell Biol* 220, e202001042 (2021).
10. Huveneers, S. & Rooij, J. de. Mechanosensitive systems at the cadherin–F-actin interface. *J Cell Sci* 126, 403–413 (2013).
11. Seddiki, R. *et al.* Force-dependent binding of vinculin to α -catenin regulates cell–cell contacts stability and collective cell behavior. *Mol Biol Cell* 29, 380–388 (2018).
12. Costes, S. V. *et al.* Automatic and Quantitative Measurement of Protein-Protein Colocalization in Live Cells. *Biophys J* 86, 3993–4003 (2004).

REVIEWER COMMENTS

Reviewer #4 (Remarks to the Author):

I would thank the reviewers to have consider carefully my comments, especially statistical analysis (Fig.R2), figure rearrangement and ratiometric analysis (Fig.8G now).

Their comments about the difficulty to realize cadherin surface in a reasonable amount of time is also fair.

I am just puzzled by two of their explanations:

- the author claims in their rebuttal letter that «The antibodies completely abolished the cell adhesion, showing their high potency of blocking $\beta 1$ integrin. » . This is not so clear for me and I was just asking if EGTA treatment or blocking B1 had the same quantitative effect on blocking the levels of adhesion forces between mitotic/interphase and mitotic/mitotic (Fig.8A-B). The strength of this manuscript is the high level of quantitative data and I was simply wondering if EGTA had the same ability than blocking antibody against B1 to modulate these forces. This is important to determine the level of importance of both type of adhesion in the development of adhesion forces.

-the importance of ECM environment of mitotic cells is not a small detail. First, I would thank the reviewers for the few precisions given in the rebuttal letter about their methodology. I understand the following: cells are pre-cultured on matrigel for 20h, +/- STC (for 12h); then the cells are detached and spread on another layer of matrigel for a maximum of 3h before performing cell-cell SCFS. Is it correct? Thus, cells are pre-conditioned on matrigel and trypsin-detachment is made on matrigel, that could also induces partial degradation of matrigel that could bind more to mitotic cells than to inter phasic cells. These details are not trivial in order to compare the effects of blocking B1 on interphasic and mitotic adhesion forces. In order to test that simply, it will be possible for the reviewer to coat a cantilever with gelatin (binding highly fibronectin) or with a Pan-collagen antibody and probe a layer of inter phasic or mitotic cells in order to show if there is differences in the energy of adhesion between both cellular conditions.

At least, a comment on a possible impact of ECM specifically associated with mitotic cells on the effects on the blocking antibody treatment should be integrated in the discussion.

NCOMMS-21-27544B

In mitosis integrins reduce adhesion to extracellular matrix and strengthen adhesion to adjacent cells

Point-by-point response to the comments of reviewer #4

Reviewer 4: I would thank the reviewers to have consider carefully my comments, especially statistical analysis (Fig.R2), figure rearrangement and ratiometric analysis (Fig.8G now).

Their comments about the difficulty to realize cadherin surface in a reasonable amount of time is also fair.

Authors: We thank the reviewer for acknowledging our effort to carefully address all raised concerns and for understanding the difficulty of conducting the proposed experiments in a reasonable amount of time.

Reviewer 4: The author claims in their rebuttal letter that «The antibodies completely abolished the cell adhesion, showing their high potency of blocking $\beta 1$ integrin». This is not so clear for me [...]

Authors: We apologize if the rationale for our claim was not clear. We conclude our claim from SCFS experiments measuring the adhesion forces of HeLa cells to Matrigel in the presence of the integrin subunit $\beta 1$ blocking-antibody AIB2 (**Suppl. Fig. 2c** and **Fig. R1**). Blocking $\beta 1$ integrins by AIB2 reduced the adhesion force of HeLa cells to Matrigel to background levels. Therefore, these experiments show high potency to block $\beta 1$ integrin-mediated adhesion. In our Manuscript we describe these results in the results section ***"Mitotic cells poorly strengthen integrin-mediated cell-ECM adhesion"*** in which we write *"Negligible adhesion forces of interphase HeLa cells to BSA or in the presence of $\beta 1$ integrin blocking antibodies (AIB2) to Matrigel showed that the initiation and strengthening of adhesion occurred predominantly via $\beta 1$ integrins"*. To avoid confusion of the reader we have also revised our Manuscript in the results section ***"Mitotic cells employ integrins to strengthen cell-cell adhesion"*** to clearly describe the potent $\beta 1$ integrin blocking by AIB2.

Fig. R1, taken from Supplementary Fig. 2c. $\beta 1$ integrin-blocking antibody AIB2 has a high potency to block adhesion formation of $\beta 1$ integrins. Adhesion forces of interphase HeLa cells, incubated with integrin subunit $\beta 1$ blocking-antibody clone AIB2 to Matrigel after given contact times. Cells were incubated with clone AIB2 for 30 min prior to the experiments (dilution 1:100) and throughout the experiments (dilution 1:1'000). Adhesion forces of unperturbed HeLa cells (**Fig. 1a**) are given in grey as reference. Dots represent adhesion forces of single cells, red bars median values and (n) the number of tested cells per condition. AS-values give the adhesion strengthening rate as the slope (\pm SE) of a linear fit through

adhesion forces for all contact times with the P -value comparing the AS-value to that of the reference data set. Given P -values calculated using a two-tailed Mann-Whitney test compare displayed adhesion forces with reference data and comparing AS-values to given reference data were calculated by two-tailed extra sum of squares F-tests.

Reviewer 4: [...] and I was just asking if EGTA treatment or blocking B1 had the same quantitative effect on blocking the levels of adhesion forces between mitotic/interphase and mitotic/mitotic (Fig.8A-B). The strength of this manuscript is the high level of quantitative data and I was simply wondering if EGTA had the same ability than blocking antibody against B1 to modulate these forces. This is important to determine the level of importance of both type of adhesion in the development of adhesion forces.

Authors: Indeed, our SCFS experiments show that mitotic^{STC} HeLa cells treated with EGTA or AIB2 establish similar adhesion force, which they strengthen similarly to interphase cells and to mitotic cells (**Fig. R2, included in our revision as new Supplementary Fig. 10c**). We carried out an additional statistical analysis of the adhesion forces established between AIB2-treated mitotic^{STC} HeLa cells adhering to interphase HeLa cells and mitotic^{STC} HeLa cells adhering to interphase HeLa cells in the presence of EGTA. This analysis shows that EGTA-treatment reduces the adhesion force between mitotic^{STC} and interphase HeLa cells much more than AIB2-treatment at contact time ≤ 20 s. However, at longer contact times, the adhesion forces of AIB2- and EGTA-treated mitotic^{STC} HeLa cells become indistinguishable and are approximately 50% of the adhesion forces established by non-perturbed mitotic^{STC} HeLa cells to interphase HeLa cells. Furthermore, the adhesion strengthening of EGTA- or AIB2-treated mitotic^{STC} HeLa cells to interphase cells is similar. Therefore, we conclude that while $\beta 1$ integrins do not contribute to the adhesion of mitotic HeLa cells to interphase cells in the first 20 s, they contribute approximately 50% to the adhesion forces at longer contact times.

Furthermore, the adhesion force and adhesion strengthening of AIB2- or EGTA-treated mitotic^{STC} HeLa cells to other mitotic^{STC} HeLa cells are indistinguishable for all contact times. Hence, this analysis shows that $\beta 1$ integrins and cadherins contribute equally to the adhesion force between two mitotic HeLa cells. To more clearly describe this situation we have revised the results section **“Mitotic cells employ integrins to strengthen cell-cell adhesion”** and state that integrins and cadherins contribute approximately equally to the adhesion of mitotic^{STC} HeLa cells to other interphase or mitotic^{STC} cells.

Fig. R2, included as new Supplementary Fig. 10c. Integrins and cadherins contribute equally to adhesion forces of mitotic^{STC} HeLa cells to interphase or other mitotic^{STC} HeLa cells. Adhesion forces of mitotic^{STC} HeLa cells, incubated with integrin subunit $\beta 1$ blocking-antibody clone AIB2 (purple) or EGTA (blue) to interphase (left) or mitotic^{STC} (right) HeLa cells

after given contact times. Cells were incubated with EGTA or AIB2 for 30 min prior to the experiments (7.5 mM EGTA and AIB2 1:100 dilution) and in the case of EGTA throughout the experiments. Adhesion forces of unperturbed HeLa cells (**Fig. 1g**) are given in green as a reference. Dots represent adhesion forces of single cells, red bars median values, and (*n*) the number of tested cells per condition. AS-values give the adhesion strengthening rate as the slope (\pm SE) of a linear fit through adhesion forces for all contact times with the *P*-value comparing the two given AS-value. Given *P*-values calculated using a two-tailed Mann-Whitney tests compare AIB2- and EGTA-treated adhesion forces. AS-values were compared by two-tailed extra sum of squares F-tests.

Reviewer 4: The importance of ECM environment of mitotic cells is not a small detail. First, I would thank the reviewers for the few precisions given in the rebuttal letter about their methodology. I understand the following: cells are pre-cultured on matrigel for 20h, +/- STC (for 12h); then the cells are detached and spread on another layer of matrigel for a maximum of 3h before performing cell-cell SCFS. Is it correct?

Authors: We note that HeLa cells were not cultured on Matrigel prior to SCFS experiments. We cultured HeLa cells in uncoated tissue culture flasks and allowed them to grow on their own excreted matrix. Only in the case of SCFS experiments that quantified mitotic-mitotic cell-cell adhesion, mitotic HeLa cells were cultured for ~12 h on Matrigel-coated glass-bottom Petri dishes in the presence of STC prior to the experiments.

Reviewer 4: [...] Thus, cells are pre-conditioned on matrigel and trypsin-detachment is made on matrigel, that could also induces partial degradation of matrigel that could bind more to mitotic cells than to inter phasic cells. These details are not trivial in order to compare the effects of blocking B1 on interphasic and mitotic adhesion forces. In order to test that simply, it will be possible for the reviewer to coat a cantilever with gelatin (binding highly fibronectin) or with a Pan-collagen antibody and probe a layer of inter phasic or mitotic cells in order to show if there is differences in the energy of adhesion between both cellular conditions.

At least, a comment on a possible impact of ECM specifically associated with mitotic cells on the effects on the blocking antibody treatment should be integrated in the discussion.

Authors: We understand the concerns of the reviewer that ECM proteins or their fragments could potentially cover HeLa cells on the substrate, which could induce the participation of integrins on cantilever-bound mitotic^{STC} HeLa cells. For the suggested control experiments, a homogenous and complete cantilever coating with antibodies or gelatin is essential to minimize non-specific interactions between the cantilever and the layer of interphase or mitotic cells. Setting up such an experiment would require the systematic development of a protocol to achieve a homogenous cantilever coating of mixed components. Additionally using gelatin to recognize fibronectin would be challenging, as also mammalian cells establish strong integrin-mediated adhesion to gelatin^{1,2}. Taking into account these limitations for the proposed experiments, we decided to conduct an array of immunofluorescent experiments to investigate whether we can detect fibronectin, laminin, collagen I, or collagen IV on HeLa cells in the cantilever- and substrate-bound conditions. Importantly, we used antibodies that bind to mouse and human ECM proteins to detect ECM proteins in mouse-derived and excreted ECM proteins by HeLa cells. First, we seeded interphase HeLa cells on

Matrigel-coated supports and allowed them to spread for 2 h. Second, we seeded HeLa cells onto Matrigel and incubated them with STC for ~12 h to enrich mitotic^{STC} HeLa cells. Third, we seeded interphase HeLa cells on concanavalinA (ConA)-coated supports and allowed them to interact with the support for 1 h. Fourth, we incubated cultured HeLa cells for ~12 h with STC to enrich mitotic^{STC} HeLa cells, detached them, and seeded them on ConA-coated supports for 30 min. While the first two conditions mimic the substrate-bound cells in cell-cell SCFS experiments, the latter two mimic the cantilever bound cells. After fixation with paraformaldehyde (PFA), we labeled the cells for DNA (DAPI), F-actin (SirActin) and for fibronectin, laminin, collagen I or collagen IV (**Fig. R3, included in our revision as new Supplementary Fig. 11**). The results show that while we could detect some of the ECM proteins within the cell, we could not detect any extracellular collagen I, collagen IV, fibronectin, or laminin under any of the conditions. We have included the results in the manuscript section “**Mitotic cells employ integrins to strengthen cell-cell adhesion**” and in **new Suppl. Fig. 11**. Therefore, we conclude that integrins that participate in mitotic cell-cell adhesion are unlikely to bind to ECM proteins on the surface of substrate-bound cells. However, since we cannot completely exclude this, we decided to mention in the **Discussion** that “*We did not detect collagens, laminins, or fibronectin on the cell surface of interphase or mitotic cells in our experiments, indicating that integrins involved in cell-cell adhesion of mitotic cells are unlikely to bind to ECM proteins on the cell surface of other interphase or mitotic cells. However, we cannot fully exclude the participation of ECM proteins in mitotic cell-cell adhesion experiments.*”

Fig. R3, included as new Supplementary Fig. 11. ECM proteins are not detectable on the cell surface of interphase or mitotic cells. Representative orthogonal views ($n=5$) of interphase (top row) or mitotic^{STC} (second row) HeLa cells seeded on ConA for 30 min, interphase HeLa cells seeded on Matrigel for 2 h (third row), or HeLa cells seeded on Matrigel and arrested in mitosis by STC for 12 h (bottom row). Cells are labeled for collagen I (far left column), collagen IV (second column), fibronectin (third column) or laminin (right column) in green, actin in red and for DNA in blue (Methods). Scale bar, 10 μ m.

References

1. Chun, J.-S., Ha, M.-J. & Jacobson, B. S. Differential Translocation of Protein Kinase C ϵ during HeLa Cell Adhesion to a Gelatin Substratum. *J Biol Chem* 271, 13008–13012 (1996).

2. Davidenko, N. *et al.* Evaluation of cell binding to collagen and gelatin: a study of the effect of 2D and 3D architecture and surface chemistry. *J Mater Sci Mater Medicine* 27, 148 (2016).

REVIEWERS' COMMENTS

Reviewer #4 (Remarks to the Author):

I would thank again the reviewers to have consider some of my comments to strengthen their careful experiments, add some key comments of their experiments, refine some of their conclusions and aspects of their discussion. Indeed, this is especially important in the case of the experiments that try to show that there is no ECM at the cell surface of analyzed cells. Despite the comments of the authors, it is quite possible to detect positive signal for collagenI and laminin at the surface of interphasic cells on matrigel, even in low amount. Simply adding these precisions in the discussion will let the readers able to see all the aspects of these beautiful and challenging experiments. I am fully supporting the publication of this manuscript.

NCOMMS-21-27544B

In mitosis integrins reduce adhesion to extracellular matrix and strengthen adhesion to adjacent cells

Point-by-point response to the comments of reviewer #4

Reviewer 4: I would thank again the reviewers to have consider some of my comments to strengthen their careful experiments, add some key comments of their experiments, refine some of their conclusions and aspects of their discussion. Indeed, this is especially important in the case of the experiments that try to show that there is no ECM at the cell surface of analyzed cells. Despite the comments of the authors, it is quite possible to detect positive signal for collagenI and laminin at the surface of interphasic cells on matrigel, even in low amount. Simply adding these precisions in the discussion will let the readers able to see all the aspects of these beautiful and challenging experiments.

I am fully supporting the publication of this manuscript.

Authors: We thank the reviewer for acknowledging our effort to carefully address the concerns raised and for fully supporting the publication of our manuscript.

In our understanding of the immunofluorescence data that we provided in the last revision, we can detect ECM proteins within the cells and co-localized them with actin staining. Because we do not see ECM proteins outside of the actin signal, we conclude that the ECM proteins are not on the cell surface but within the cells. However, since we cannot fully exclude the possibility of ECM on the cell surface, we write in the discussion of our revised Manuscript: *'However, we cannot fully exclude the participation of ECM proteins in mitotic cell-cell adhesion experiments.'*